# Moniqua: Modulo Quantized Communication in Decentralized SGD

## Abstract

Decentralized stochastic gradient descent (SGD), where parallel workers are connected to form a graph and communicate adjacently, has shown promising results both theoretically and empirically. In this paper we propose Moniqua, a technique that allows decentralized SGD to use quantized communication. We prove in theory that Moniqua communicates a provably bounded number of bits per iteration, while converging at the same asymptotic rate as the original algorithm does with full-precision communication. Moniqua improves upon prior works in that it (1) requires no additional memory, (2) applies to non-convex objectives, and (3) supports biased or linear quantizers. We demonstrate empirically that Moniqua converges faster with respect to wall clock time than other quantized decentralized algorithms. We also show that Moniqua is robust to very low bit-budgets, allowing less than 4-bits-per-parameter communication without affecting convergence when training VGG16 on CIFAR10.

## 1 Introduction

Stochastic gradient descent (SGD), as a widely adopted optimization algorithm for machine learning, has shown promising performance when running at large scale (Zhang, 2004; Bottou, 2010; Dean et al., 2012; Goyal et al., 2017). However, the communication bottleneck among workers[1] when running distributed SGD presents a non-trivial challenge (Alistarh, 2018). State-of-the-art frameworks such as TensorFlow (Abadi et al., 2016), CNTK (Seide and Agarwal, 2016) and MXNet (Chen et al., 2015) are built in a centralized fashion, where workers exchange gradients either via a centralized parameter server (Li et al., 2014a;b) or the MPI AllReduce operation (Gropp et al., 1999). Such a design, however, puts heavy pressure on the central server and strict requirements on the underlying network. In other words, when the underlying network is poorly constructed, i.e. high latency or low bandwidth, it can easily cause degradation of training performance due to communication congestion in the central server or stragglers (slow workers) in the system.

There are two general approaches to deal with these problems: (1) decentralized training (Lian et al., 2017a;b; Tang et al., 2018a; Hendrikx et al., 2018) and (2) quantized communication[2] (Zhang et al., 2017; Alistarh et al., 2017; Wen et al., 2017). In decentralized training, all the workers are connected to form a graph and each worker communicates only with adjacent workers by averaging model parameters. This balances load and is robust to scenarios where workers can only be partially connected or the communication latency is high. On the other hand, quantized communication reduces the amount of data exchanged among workers, which leads to faster convergence with respect to wall clock time (Alistarh et al., 2017; Seide et al., 2014; Doan et al., 2018; Zhang et al., 2017; Wang et al., 2018). This is especially useful when the communication bandwidth is restricted.

At this point, a natural question is: *Can we apply quantized communication to decentralized training, and thus benefit from both of them?* Unfortunately, directly combining them together negatively affects the convergence rate (Tang et al., 2018b). This happens because existing quantization techniques are mostly designed for centralized SGD, where workers communicate via exchanging gradients (Alistarh et al., 2017; Seide et al., 2014; Wangni et al., 2018). Gradients are robust to quantization since they get smaller in magnitude near local optimum and in some sense carry less information, causing quantization error to approach zero (De Sa et al., 2018). In contrast, decentralized workers are communicating model parameters, which do not necessarily approach zero, and so quantization error does not diminish unless precision is explicitly increased (Tang et al., 2018c). Previous work

---

[1]A worker could refer to any computing unit that is capable of computing, communicating and has local memory such as CPU, GPU, or even a single thread, etc.

[2]These approaches include low-precision, sparsification, and compression techniques more generally.

solved this problem by adding an error tracker to compensate quantization errors (Tang et al., 2019) or adding replicas of neighboring models and focusing on quantizing model difference which does approach zero (Koloskova et al., 2019; Tang et al., 2018b). However, these methods suffer from trade-offs and limitations in that: (1) the extra replicas or error tracking incurs substantial memory overhead that is proportional to model size (more details in Section 2); and (2) these methods are statistically restricted, in the sense that they are either limited to convex problems (Koloskova et al., 2019) or require unbiased or non-linear quantizers (Koloskova et al., 2019; Tang et al., 2018b; 2019).

To address these problems, in this paper we propose Moniqua, an extra-memory-free (details in Section 2) method for decentralized training to use quantized communication. Moniqua supports both biased and linear quantizers, as well as non-convex objectives.

**Intuition behind Moniqua.** In a communication step of decentralized training, a worker $w_1$ updates its model parameter $m_1$ by averaging with a neighboring worker $w_2$'s model parameter $m_2$: $m_1 \leftarrow \frac{1}{2}(m_1 + m_2)$. Note that $\frac{1}{2}(m_1 + m_2) = m_1 + \frac{1}{2}(m_2 - m_1)$, so averaging is equivalent to letting $w_1$ obtain $m_2 - m_1$ (same logic for $w_2$). Since $m_1$ and $m_2$ will approach the same local optimum as the algorithm converges, we can expect the higher-order bits of $m_1$ and $m_2$ to get close. Then we can save communication by having $w_2$ not communicate those higher-order bits to $w_1$. More explicitly, if we know that $\|m_1 - m_2\|_\infty \leq \theta$ for some known parameter $\theta$ (later we will show it can be derived in theory), then instead of sending the entire model $m_2$ which might cause overhead, $w_2$ can just send its $j$-th coordinate $(m_2)_j$ as $(m_2)_j \bmod \theta$ ($\forall j \in [d]$). Note that given $\|m_1 - m_2\|_\infty \leq \theta$:

$$(m_2)_j \bmod \theta - (m_1)_j \bmod \theta = ((m_2)_j - (m_2)_j) \bmod \theta = (m_2)_j - (m_2)_j$$

so $w_1$ can obtain the $j$-th coordinate of $m_2 - m_1$ by locally computing $(m_2)_j \bmod \theta - (m_1)_j \bmod \theta$ with $(m_2)_j \bmod \theta$ received from $w_2$. Since $(m_2)_j \bmod \theta$ is generally a smaller number than $(m_2)_j$, $w_2$ can send fewer bits with the same level of absolute error.

In this paper, we make the following contributions.

- We show by example that directly quantizing communication in decentralized training, even with an unbiased quantizer, can fail to converge asymptotically. (Section 3)

- We propose **Moniqua**, a general algorithm that uses **mo**dular arithmetic for commu**ni**cation **qua**ntization in decentralized training. We prove applying Moniqua achieves the same asymptotic convergence rate as the baseline full-precision algorithm (D-PSGD) while requiring at most $O(\log \log n)$ number of bits per parameter communicated, where $n$ is the number of parallel workers. (Section 4)

- We apply Moniqua to decentralized algorithms with variance reduction and asynchronous communication ($D^2$ and AD-PSGD) and prove Moniqua enjoys the same asymptotic rate as with full-precision communication when applied to these cases. (Section 5)

- We empirically evaluate Moniqua and show it outperforms all the related algorithms given an identical quantizer. We also show Moniqua is scalable and robust to very low bit-budgets, and we introduce techniques we found empirically useful to run Moniqua even more efficiently. (Section 6)

## 2 RELATED WORK

**Decentralized Stochastic Gradient Descent (SGD)**   Decentralized algorithms (Mokhtari and Ribeiro, 2015; Sirb and Ye, 2016; Lan et al., 2017; Wu et al., 2018a) have been widely studied with consideration of communication efficiency, privacy and scalability. In the domain of large-scale machine learning, D-PSGD was the first Decentralized SGD algorithm that enjoys the same asymptotic convergence rate $O(1/\sqrt{Kn})$ (where $K$ is the number of total iterations and $n$ is the number of workers) as centralized algorithms (Lian et al., 2017a). After D-PSGD came $D^2$, which improves D-PSGD and is applicable to the case where workers are not sampling from identical data sources (Tang et al., 2018a). Another extension was AD-PSGD, which lets workers communicate *asynchronously* and has a convergence rate of $O(1/\sqrt{K})$ (Lian et al., 2017b). In this paper we prove that Moniqua is applicable to all of these three algorithms. Other relevant work includes: He et al. (2018), which investigates decentralized learning on linear models; Nazari et al. (2019), which introduces decentralized algorithms with online learning; Zhang and You (2019), which analyzes the case when workers cannot mutually communicate; and Assran et al. (2018), which investigates Decentralized SGD specifically for deep learning.

**Quantized Communication in Centralized SGD**   Prior research on quantized communication is often focused on centralized algorithms, such as randomized quantization (Doan et al., 2018; Suresh et al., 2017; Zhang et al., 2017) and randomized sparsification (Wangni et al., 2018; Stich et al., 2018; Wang et al., 2018; Alistarh et al., 2018). Many examples of prior work focus on studying quantization in the communication of deep learning tasks specifically (Han et al., 2015; Wen et al., 2017; Grubic et al., 2018). Alistarh et al. (2017) proposes QSGD, which uses an encoding-efficient scheme, and discusses its communication complexity. Another method, 1bitSGD, quantizes exchanged gradients with one bit and shows great empirical success on speech recognition (Seide et al., 2014). Other work discusses the convergence rate under sparsified or quantized communication (Jiang and Agrawal, 2018; Stich et al., 2018). Acharya et al. (2019) theoretically analyzes sublinear communication for distributed training.

**Quantized Communication in Decentralized SGD**   Quantized communication for decentralized algorithms is a rising topic in the optimization community. Previous work has proposed decentralized algorithms with quantized communication for strongly convex objectives (Reisizadeh et al., 2018; Koloskova et al., 2019). Following that, Tang et al. (2018b) proposes DCD/ECD-PSGD, which quantizes communication via estimating model difference. Furthermore, Tang et al. (2019) proposes DeepSqueeze, which applies an error-compensation method (Wu et al., 2018b) to decentralized setting. From a systems perspective, Koloskova et al. (2019) and Tang et al. (2018b) require $O(d \cdot l)$ and Tang et al. (2019) requires $O(d)$ extra memory compared to D-PSGD to implement quantized communication, where $d$ denotes the dimension of the model and $l$ denotes the number of connections in the network. In comparison, Moniqua is extra-memory-free.

## 3   SETTING AND NOTATION

In this section, we introduce our notation and the general assumptions we will make about the quantizers for our results to hold. Then we describe D-PSGD (Lian et al., 2017a), the basic algorithm for Decentralized SGD, and we show how naive quantization can fail in decentralized training.

**Quantizers.**   Throughout this paper, we assume that we use a quantizer $\mathcal{Q}_\delta$ that has bounded error

$$\|\mathcal{Q}_\delta(x) - x\|_\infty \leq \delta, \quad \forall x \in [-1, 1]^d \tag{1}$$

where $\delta$ is some constant. In general, a smaller $\delta$ denotes more fine-grained quantization requiring more bits. For example, a biased linear quantizer can achieve (1) by rounding $x$ to the nearest number in the set $\{2\delta n \mid n \in \mathbb{Z}\}$; this will require about $\delta^{-1}$ quantization points to cover the interval $[-1, 1]$, so such a linear quantizer can satisfy (1) using only $\left\lceil \log_2 \left( \frac{1}{\delta} + 1 \right) \right\rceil$ bits (Li et al., 2017; Gupta et al., 2015). Note that (1) can be satisfied (for appropriate values of $\delta$) by both linear (Gupta et al., 2015; De Sa et al., 2017) and non-linear (Stich, 2018; Alistarh et al., 2017) quantizers, and thus it is more general than assumptions used in previous works where only non-linear quantizers are considered (Koloskova et al., 2019; Tang et al., 2018c; 2019).

**Decentralized parallel SGD (D-PSGD).**   D-PSGD (Lian et al., 2017a) is the first and most basic Decentralized SGD algorithm. In D-PSGD, $n$ workers are connected to form a graph. Each worker $i$ stores a copy of model $x \in \mathbb{R}^d$ and a local dataset $\mathcal{D}_i$ and collaborates to optimize

$$\min_{x \in \mathbb{R}^d} f(x) = \frac{1}{n} \sum_{i=1}^n \underbrace{\mathbb{E}_{\xi \sim \mathcal{D}_i} f_i(x; \xi)}_{f_i(x)} \tag{2}$$

where $\xi$ is data sample from $\mathcal{D}_i$. In each iteration of D-PSGD, worker $i$ computes a local gradient sample using $\mathcal{D}_i$. Then it *averages* its model parameters with its neighbors according to a symmetric and doubly stochastic matrix $W$, where $W_{ij}$ denotes the ratio worker $j$ averages from worker $i$. Formally: Let $x_{k,i}$ and $\widetilde{g}_{k,i}$ denote local model and sampled gradient on worker $i$ at $k$-th iteration, respectively. Let $\alpha$ denote the step size. The update rule of D-PSGD can be expressed as:

$$x_{k+1,i} = \sum_{j=1}^n x_{k,j} W_{ji} - \alpha \widetilde{g}_{k,i} = x_{k,i} \underbrace{- \sum_{j=1}^n (x_{k,i} - x_{k,j}) W_{ji}}_{\text{communicate to reduce difference}} \underbrace{- \alpha \widetilde{g}_{k,i}}_{\text{gradient step}} \tag{3}$$

From (3) we can see the update of a single local model contains two parts: communication to reduce model difference and a gradient step. Lian et al. (2017a) shows that all local models in D-PSGD are able to reach the same stationary point.

**Failure with direct quantization.** Here, we illustrate why directly quantizing communication in decentralized training —naively quantizing the exchanged data—can fail to converge asymptotically even on a simple problem. This naive approach with quantizer $\mathcal{Q}_\delta$ can be represented by

$$x_{k+1,i} = x_{k,i}W_{ii} + \sum_{j \neq i} \mathcal{Q}_\delta(x_{k,j})W_{ji} - \alpha\widetilde{g}_{k,i} \qquad (4)$$

Based on Equation 4, we obtain the following theorem.

**Theorem 1** *For some constant $\delta$, suppose that we use an unbiased linear quantizer $\mathcal{Q}$ with representable points $\{\delta n \mid n \in \mathbb{Z}\}$ to learn on the quadratic objective function $f(x) = (x - \delta/2)^\top(x - \delta/2)/2$ with the direct quantization approach (4). Let $\phi$ denote the smallest value of a non-zero entry in $W$. Regardless of what step size we adopt, it will always hold for all iterations $k$ and local model indices $i$ that $\mathbb{E}\|\nabla f(x_{k,i})\|^2 \geq \frac{\phi^2\delta^2}{8(1+\phi^2)}$. That is, the local iterates will fail to asymptotically converge to a region of small gradient magnitude in expectation.*

## 4 MONIQUA

Theorem 1 shows that when directly quantizing communication in decentralized SGD, even with an unbiased quantizer, any local model can fail to converge on a simple quadratic objective. In this seciton, we propose a technique, Moniqua, that solves this problem. Moniqua works under the following common assumptions for analyzing decentralized optimization algorithms (Lian et al., 2017a; Tang et al., 2018b; Koloskova et al., 2019).

(A1) **Lipschitzian gradient.** All the functions $f_i$ have $L$-Lipschitzian gradients.
$$\|\nabla f_i(x) - \nabla f_i(y)\| \leq L\|x - y\|, \forall x, y \in \mathbb{R}^d$$

(A2) **Spectral gap.** The communication matrix $W$ is a symmetric doubly stochastic matrix and $\max\{|\lambda_2(W)|, |\lambda_n(W)|\} = \rho < 1$, where $\lambda_i(W)$ denotes the $i$th eigenvalue of $W$.

(A3) **Bounded variance.** There exist non-negative $\sigma$ and $\varsigma$ such that
$$\mathbb{E}_{\xi_i \sim \mathcal{D}_i}\left\|\nabla\widetilde{f}_i(x; \xi_i) - \nabla f_i(x)\right\|^2 \leq \sigma^2, \qquad \mathbb{E}_{i \sim \{1, \cdots, n\}}\|\nabla f_i(x) - \nabla f(x)\|^2 \leq \varsigma^2$$
where $\nabla\widetilde{f}_i(x; \xi_i)$ denotes gradient sample on worker $i$ computed via data sample $\xi_i$.

(A4) **Initialization.** All the local models are initialized by the same weight: $x_{0,i} = x_0$, for all $i$ and without loss of generality $x_0 = 0$.

(A5) **Bounded gradient magnitude.** The norm of a sampled gradient is bounded by $\|\widetilde{g}_{k,i}\|_\infty \leq G_\infty$, for all $i$ and $k$ with some constant $G_\infty$.

In Section 1, we described how a modulo operation can be used to avoid sending redundant bits if a bound $\theta$ on model difference is known. Here we outline how we can obtain such a bound. We do so by leveraging the following insight: in decentralized training, all the workers initialize local models at same point and average with each other periodically. The only difference among their models is caused by the sampled gradients (updated with the step size), and this difference is reduced each time they communicate. Since we have an upper bound on the magnitude of the gradients (A5) as well as a bound characterizing how quickly the communication process converges (A2), we can combine these to get an a priori bound $\theta$ on how much the models can differ. We can then pass this bound $\theta$ as a parameter to the algorithm, which can proceed to modulo-quantize the communication via the process described in Section 1. We formalize this approach as Moniqua (Algorithm 1).

---

**Algorithm 1** Pseudo-code of Moniqua on worker $i$

---

**Input:** initial point $x_{0,i} = x_0$, step size $\alpha$, the priori bound $\theta$, communication matrix $W$, number of iterations $K$, quantizer $\mathcal{Q}_\delta$, neighbor list $\mathcal{N}_i$
1: **for** $k = 0, 1, 2, \cdots, K - 1$ **do**
2:     Compute a local stochastic gradient $\widetilde{g}_{k,i}$ with data sample $\xi_{k,i}$ and current weight $x_{k,i}$
3:     Compute modulo-ed model: $q_{k,i} \leftarrow \theta \cdot \mathcal{Q}_\delta\left(\frac{x_{k,i}}{\theta} \mod 1\right)$ (element-wise division and mod)
4:     Average with neighboring workers: $x_{k+\frac{1}{2},i} \leftarrow x_{k,i} + \sum_{j \in \mathcal{N}_i}(q_{k,j} - q_{k,i})W_{ji}$
5:     Update the local weight with local gradient: $x_{k+1,i} \leftarrow x_{k+\frac{1}{2},i} - \alpha\widetilde{g}_{k,i}$
6: **end for**
**Output:** Averaged model $\overline{X}_K = \frac{1}{n}\sum_{i=1}^n x_{K,i}$

---

In line 3 we rescale each coordinate so that the number to be quantized falls in the region of $[-1, 1]$, which is required for (1) to apply. Note that with quantization, the priori bound $\theta$ could increase since local models may move further apart due to quantization error. However, with appropriately chosen $\delta$, we can still obtain a bound $\theta$ and apply modulo-quantized communication that allows Moniqua to converge. We present these parameter choices in Theorem 2, along with the resulting convergence rate for Moniqua.

**Theorem 2** *If we run Algorithm 1 in a setting where*

$$\theta = \frac{2\log(16n)\alpha G_\infty}{1-\rho}, \qquad \delta = \frac{1-\rho}{4\log(16n)}, \quad and \quad \alpha = \frac{1}{\varsigma^{2/3}K^{1/3} + \sigma\sqrt{K/n} + 2L},$$

*then the output of Algorithm 1 converges at the asymptotic rate*

$$\frac{1}{K}\sum_{k=0}^{K-1}\mathbb{E}\left\|\nabla f(\overline{X}_k)\right\|^2 \lesssim \frac{1}{K} + \frac{\sigma}{\sqrt{nK}} + \frac{\varsigma^{\frac{2}{3}}}{K^{\frac{2}{3}}} + \frac{\sigma^2 n}{\sigma^2 K + n} + \frac{G_\infty^2 dn}{\sigma^2 K + n}.$$

*where $\rho$, $f(0) - f^*$ and $L$ are omitted as constants.*

**Consistent with D-PSGD.** Note that D-PSGD converges at the asymptotic rate of $O(\sigma/\sqrt{nK} + \varsigma^{\frac{2}{3}}/K^{\frac{2}{3}} + n/K)$, and thus Moniqua has the same asymptotic rate as D-PSGD (Lian et al., 2017a). In other words, the asymptotic convergence rate is not negatively impacted by the quantization.

**Robust to large $d$.** In Assumptions (A3) and (A5), we use $l_2$-norm and $l_\infty$-norm to bound sample variance and gradient magnitude, respectively. Note that, when $d$ gets larger, the variance $\sigma^2$ will also grow proportionally. So, the last term will tend to remain $n/K$ asymptotically with large $d$.

**How many bits does Moniqua need?** The specific number of bits required by Moniqua depends on the underlying quantizer ($\mathcal{Q}_\delta$). If we use nearest rounding (Gupta et al., 2015) as $\mathcal{Q}_\delta$ in Theorem 2, it suffices to use at each step a number of bits $\mathcal{B}$ for each parameter sent, where

$$\mathcal{B} = \left\lceil\log_2\left(\frac{1}{\delta} + 1\right)\right\rceil = \left\lceil\log_2\left(\frac{4\log_2(16n)}{1-\rho} + 1\right)\right\rceil$$

Note that this bound is independent of model dimension $d$. When the system scales up, the number of required bits grows at a rate of $O(\log\log n)$.

## 5 SCALABLE MONIQUA

Previous work has extended D-PSGD to $D^2$ (Tang et al., 2018a) (to make Decentralized SGD applicable to workers sampling from different data sources) and AD-PSGD (Lian et al., 2017b) (an asynchronous version of D-PSGD). In this section, we theoretically prove Moniqua is applicable to both of these algorithms.

**Moniqua with Decentralized Data** Decentralized data refers to the case where all the local datasets $\mathcal{D}_i$ are not identically distributed (Tang et al., 2018a). More explicitly, the outer variance $\mathbb{E}_{i\sim\{1,\cdots,n\}}\left\|\nabla f_i(x) - \nabla f(x)\right\|^2$ is no longer bounded by $\varsigma^2$ as assumed in D-PSGD (Assumption (A3)). This update rule presented can be explicitly expressed in two steps[3]:

$$X_{k+\frac{1}{2}} = 2X_k - X_{k-1} - \alpha\widetilde{G}_k + \alpha\widetilde{G}_{k-1}$$
$$X_{k+1} = X_{k+\frac{1}{2}}W + (Q_k - X_{k+\frac{1}{2}})(W - I)$$

where $X_k$, $\widetilde{G}_k$ and $Q_k$ are matrix in the shape of $\mathbb{R}^{d\times n}$, where their $i$-th column are $x_{k,i}$, $\widetilde{g}_{k,i}$ and $q_{k,i}$ respectively. And $X_{-1}$ and $\widetilde{G}_{-1}$ are $0^{d\times n}$ by convention. Based on this, we obtain the following convergence theorem.

**Theorem 3** *If we run $D^2$ with Monqiua in a setting where*

$$\theta = (6D_1 n + 8)\alpha G_\infty, \qquad \delta = \frac{1}{6nD_2}, \quad and \quad \alpha = \frac{1}{\sigma\sqrt{K/n} + 2L},$$

*where $D_1$ and $D_2$ are two constants that only depend on the eigenvalues of $W$ (definition can be found in supplementary material), the output has the following asymptotic convergence rate:*

$$\frac{1}{K}\sum_{k=0}^{K-1}\mathbb{E}\left\|\nabla f(\overline{X}_k)\right\|^2 \lesssim \frac{1}{K} + \frac{\sigma}{\sqrt{nK}} + \frac{\sigma^2 n}{\sigma^2 K + n} + \frac{G_\infty^2 dn}{\sigma^2 K + n}.$$

---

[3]Detailed pseudo-code in the supplementary material.

Note that $D^2$ (Tang et al., 2018a) with full-precision communication has the asymptotic convergence rate of $O\left(1/K + \sigma/\sqrt{nK} + n/K\right)$, Moniqua on $D^2$ has the same asymptotic rate.

**Moniqua with Asychronous Communication.** Both D-PSGD and $D^2$ are synchronous algorithms as they require global synchronization at the end of each iteration, which can become a bottleneck when such synchronization is not cheap. Another algorithm, AD-PSGD, avoids this overhead by letting workers communicate asynchronously (Lian et al., 2017b). In the analysis of AD-PSGD, an iteration represents a *single* gradient update on *one* randomly-chosen worker, rather than a synchronous bulk update of all the workers. This single-worker-update analysis models the asynchronous nature of the algorithm. We apply Moniqua to AD-PSGD and obtain the following update rule[4]:

$$X_{k+1} = X_k W_k + (Q_k - X_k)(W_k - I) - \alpha \widetilde{G}_{k-\tau_k}$$

where $W_k$ describes the communication behaviour between the $k$th and $(k+1)$th gradient update, and $\tau_k$ denotes the delay (measured as a number of iterations) between when the gradient is computed and updated to the model. Note that unlike D-PSGD, here $W_k$ can be different at each update step and usually each individually has $\rho = 1$, so we can't expect to get a bound in terms of a bound on the spectral gap, as we did in Theorems 2 and 3. Instead, we require the following condition, which is inspired by the literature on Markov chain Monte Carlo methods: for some constant $t_{\mathrm{mix}}$,

$$\forall \mu \in \mathbb{R}^n, \ \forall k \in \mathbb{N}, \ \text{if } \mu_i \geq 0 \text{ and } \mathbb{1}^\top \mu = 1, \ \text{it must hold that } \left\|\left(\prod_{i=1}^{t_{\mathrm{mix}}} W_{k+i}\right)\mu - \tfrac{\mathbb{1}}{n}\right\|_1 \leq \tfrac{1}{2}.$$

We call this constant $t_{\mathrm{mix}}$ because it is effectively the *mixing time* of the time-inhomogeneous Markov chain with transition probability matrix $W_k$ at time $k$ (Levin and Peres, 2017). Note that this condition is more general than those used in previous work on AD-PSGD because it does not require that the $W_k$ are sampled independently or in an unbiased manner. Using this, we obtain the following convergence theorem.

**Theorem 4** *If we run AD-PSGD with Moniqua in a setting where*

$$\theta = 16 t_{\mathrm{mix}} \alpha G_\infty, \qquad \delta = \frac{1}{32 t_{\mathrm{mix}}}, \quad \text{and} \quad \alpha = \frac{n}{2L + \sqrt{K(\sigma^2 + 6\varsigma^2)}},$$

*the output has the following asymptotic convergence rate:*

$$\frac{1}{K}\sum_{k=0}^{K-1} \mathbb{E}\left\|\nabla f(\overline{X}_k)\right\|^2 \lesssim \frac{1}{K} + \frac{\sqrt{\sigma^2 + 6\varsigma^2}}{\sqrt{K}} + \frac{(\sigma^2 + 6\varsigma^2)t_{\mathrm{mix}}^2 n^2}{(\sigma^2 + 6\varsigma^2)K + 1} + \frac{n^2 t_{\mathrm{mix}}^2 G_\infty^2 d}{(\sigma^2 + 6\varsigma^2)K + 1}$$

Note that AD-PSGD (Lian et al., 2017b) with full-precision communication has the asymptotic convergence rate of $O\left(1/K + \sqrt{\sigma^2 + 6\varsigma^2}/\sqrt{K} + n^2/K\right)$, Moniqua converges at the same rate.

## 6 EXPERIMENTS

In this section, we evaluate Moniqua empirically. First, we compare Moniqua and other quantized decentralized training algorithms' convergence under different network configurations. Second, we evaluate Moniqua's scalability on $D^2$ and AD-PSGD. Third, we introduce two additional techniques to run Moniqua more efficiently and empirically investigate the limits of Moniqua.

**Configuration.** All the models and training scripts in this section are implemented in PyTorch and run on Google Cloud Platform. We launch an instance as one worker, each configured with a 2-core CPU with 4 GB memory and an NVIDIA Tesla P100 GPU. We use MPICH as the communication backend. All the instances are running Ubuntu 16.04, and latency and bandwidth on the underlying network are configured using the `tc` command in Linux. In all the experiments, we use the following hyperparameters by default: batch size $= 128$, weight decay $= 1e - 4$, and momentum $= 0.9$, which are default values adopted in previous works (Lian et al., 2017b; Grubic et al., 2018). We tune the step size from set $\{0.5, 0.1, 0.05, 0.01\}$ for each algorithm. Throughout our experiments, we adopt the commonly used (Gupta et al., 2015; Li et al., 2017) stochastic rounding[5] with quantization step $\delta$.

---

[4]Details in the supplementary material.

[5]Since several baselines are not applicable to biased quantizers, for fair comparison we consistently use stochastic rounding (unbiased). More experiments using different quantizers including biased and non-linear quantizers on Moniqua can be found in supplementary material.

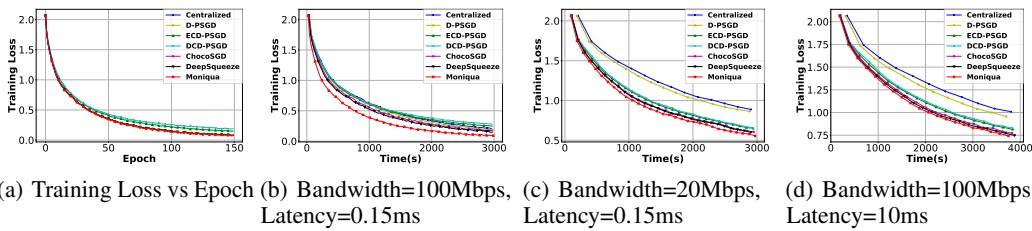

(a) Training Loss vs Epoch (b) Bandwidth=100Mbps, Latency=0.15ms (c) Bandwidth=20Mbps, Latency=0.15ms (d) Bandwidth=100Mbps, Latency=10ms

Figure 1: Performance of different algorithms under different network configurations

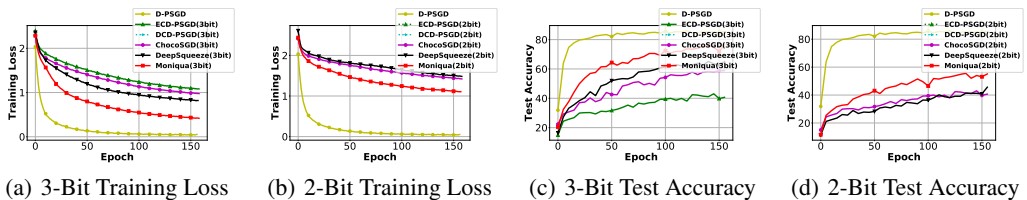

(a) 3-Bit Training Loss (b) 2-Bit Training Loss (c) 3-Bit Test Accuracy (d) 2-Bit Test Accuracy

Figure 2: Performance of Moniqua and other quantization algorithms under extreme bit-budget.

**Wall-clock Time Evaluation.** We start by evaluating the performance of Moniqua and other baseline algorithms under different network configurations. We launch 8 workers connected in a ring topology and train a ResNet110 (He et al., 2016) model on CIFAR10 (Krizhevsky et al., 2014). We compare Moniqua with the following baselines:[6] Centralized (implemented as a standard AllReduce operation), D-PSGD (Lian et al., 2017a) with full-precision communication, DCD/ECD-PSGD (Tang et al., 2018b), ChocoSGD (Koloskova et al., 2019) and DeepSqueeze (Tang et al., 2019). We set $\delta = 0.01$ for stochastic rounding across all algorithms that use quantization. To prevent overflow, we use 16-bit integers[7] `torch.int16` as the floored output on the sender side. For Moniqua, we set $\theta = 3.0$.

We plot our results in Figure 1. As can be seen in Figure 1(a), with respect to epochs, All the algorithms have similar convergence curve while DCD/ECD-PSGD have slightly slower convergence curves. We can see from Figures 1(b) and 1(c) that when the network bandwidth decreases, the curves begin to separate. AllReduce and full-precision D-PSGD suffer the most, since they require a large volume of high-precision exchanged data. And from Figure 1(b) to Figure 1(d), when the network latency increases, we observe similar behavior. On the other hand, from Figure 1(b) to Figure 1(c) and Figure 1(d), curves of all the quantized baselines (DCD/ECD-PSGD, ChocoSGD and DeepSqueeze) are getting closer to Moniqua. This is because, as shown in Figure 1(b), the extra updating of the replicas in DCD/ECD-PSGD and ChocoSGD as well as the error tracking in DeepSqueeze counteract the benefits from accelerated communication. However, when network bandwidth decreases or latency increases, communication becomes the bottleneck and allow these algorithms obtain acceleration compared to centralized SGD and D-PSGD. Delay between Moniqua and quantized baselines does not vary with the network since that only depends on the their extra local computation (error tracking and replica update). We observe that compared to Moniqua, DCD/ECD-PSGD is approximately 13 seconds slower while ChocoSGD and DeepSqueeze being 10 and 8 seconds slower repectively. From Figure 1 we can see that Moniqua outperforms all these other algorithms.

**Aggressive Quantization.** Now we investigate how Moniqua and baselines behave under aggressive quantization. We enforce two strict bit-budget: 2bit and 3bit (per parameter). We plot the results in Figure 2. We can see that DCD-PSGD fails to converge in both cases and ECD-PSGD fails to converge with 2bit. This is consistent with results in previous work (Tang et al., 2018c; 2019). On the other hand, Moniqua converges faster than any other baselines. We observe at the end of 150 epoch,

---

[6]Other algorithms are not applicable to non-convex DNN problems, so we are not comparing them here.

[7]Since we are measuring the system performance, the specific number of bits is not the focus here. In later section we will discuss statistical performance with small number of bits.

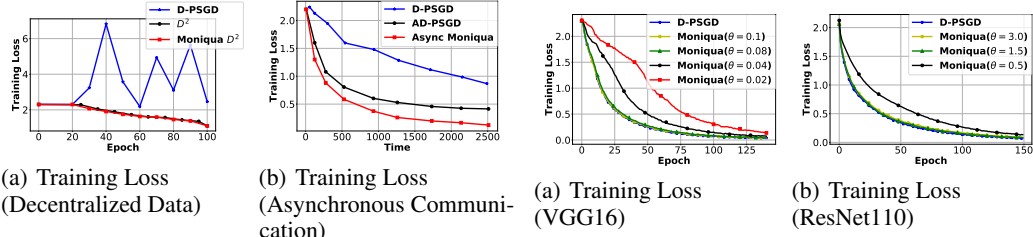

(a) Training Loss (Decentralized Data)

(b) Training Loss (Asynchronous Communication)

(a) Training Loss (VGG16)

(b) Training Loss (ResNet110)

Figure 3: Performance of applying Moniqua on $D^2$ and AD-PSGD

Figure 4: Performance of Moniqua on VGG16 and ResNet110 under different $\theta$

with 3-bit communication Moniqua achieves $85\%$ training accuracy while other baselines are below $70\%$ (full precision achieves $97\%$). Compared to the theoretical results in Section 4, we show that Moniqua is much more robust to low-bits budget in practice.

**Scalability of Moniqua.** We evaluate how Moniqua can be applied to $D^2$ (Tang et al., 2018a) and AD-PSGD (Lian et al., 2017b). First, we demonstrate how applying Moniqua to $D^2$ can handle decentralized data. We launch 10 workers, collaborating to train a VGG16 (Simonyan and Zisserman, 2014) model on CIFAR10. Similar to the setting of $D^2$ (Tang et al., 2018a), we let each worker have exclusive access to 1 labels (of the 10 labels total in CIFAR10). In this way, the data variance among workers is maximized. We plot the results in Figure 3(a). We observe that applying Moniqua on $D^2$ does not affect the convergence rate while D-PSGD can no longer converge because of the outer variance. Here we omit the wall clock time comparison since the communication volume is the same in comparison of Moniqua and Centralized algorithm in Figure 1.

Next, we evaluate Moniqua on AD-PSGD. We launch 6 workers organized in a ring topology, collaborating to train a ResNet110 model on CIFAR10. We set the network bandwidth to be 20Mbps and latency to be 0.15ms. We plot the results in Figure 3(b). We can see that both AD-PSGD and asynchronous Moniqua outperform D-PSGD. Besides, Moniqua outperforms AD-PSGD in that communication is reduced, which is aligned with the intuition and theory.

**Efficient Moniqua.** There are two techniques we have observed to improve the performance of Moniqua when using stochastic rounding: $\mathcal{Q}_\delta(x) = \delta \lfloor \frac{x}{\delta} + u \rfloor$ (where $u$ is uniformly sampled from $[0, 1]$), $\forall x \in \mathbb{R}^d$. The **first** is to use *shared randomness*, in which the same random seed is used for stochastic rounding on all the workers. That is, if two workers are exchanging tensors $x$ and $y$ respectively, then the floored tensors $\lfloor \frac{x}{\delta} + u \rfloor$ and $\lfloor \frac{y}{\delta} + u \rfloor$ they send use the *same* randomly sampled value $u$. This provably reduces the error due to quantization (more details are in the supplementary material). The **second** technique is to use a standard entropy compressor like bzip to further compress the communicated tensors. This can help further reduce the number of bits because the modulo operation in Moniqua can introduce some redundancy in the higher-order bits, which a traditional compression algorithm can easily remove.

To evaluate these methods, we train both ResNet110 and VGG16 on CIFAR10 using 8 ring-connected workers. We plot the training loss under different $\theta$ in Figure 4 (with $\delta = 0.01$ for stochastic rounding). Note that for VGG16, it can tolerate small $\theta = 0.08$ while still preserving the convergence rate. On the other hand, for ResNet110, it begins to diverge when $\theta$ decreases to $0.5$. This is because VGG16 has more fully connected layers than ResNet110, and these layers are less sensitive to quantization, as claimed in (Grubic et al., 2018). We observed that the fewest number of bits per number needed to communicate by Moniqua for VGG16 and ResNet110 to guarantee convergence (accuracy loss $< 0.3\%$, criterion adopted by (Grubic et al., 2018)) are $3.64$ and $5.67$, respectively (details in the supplementary material).

## 7 CONCLUSIONS

In this paper we propose Moniqua, a simple unified method of quantizing the communication in decentralized training algorithms. Theoretically, Moniqua supports biased quantizer and non-convex problems, while enjoying the same asymptotic convergence rate as full-precision-communication algorithms without incurring storage or computation overhead. Empirically, we observe Moniqua

converges faster than other related algorithms with respect to wall clock time. Additionally, Moniqua is robust to very low bits-budget.

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

# Supplementary Material

## A  OVERVIEW

This supplementary material contains proofs of all the theoretical results and extra experimental results of Moniqua. It is organized as follows: In Section B, we provably explain why using shared randomness in communication with stochastic rounding can improve performance (theoretical explanation for technique 1 in Experiment *Efficient Moniqua*). Then we demonstrate more experimental results in Section C. In Section D, we illustrate why naively quantizing communication in D-PSGD fails to converge asymptotically, as a proof to Theorem 1. In Section E, we introduce some useful tools of modeling communication as a Markov Chain for the rest of the proof (part of the intuition is illustrated in the paper). We recommend to go through this before getting into Section F to H. Finally we will provide proof to Theorem 2, 3 and 4 from Section F to H, with corollaries contained in the corresponding sections. Detailed algorithm statements for applying Moniqua on $D^2$ and AD-PSGD can be found in Section G Algorithm 2 and Section H Algorithm 3, respectively.

## B  SHARED RANDOMNESS (EXPERIMENT OF *Efficient Moniqua*)

In this section, we provide a theoretical explanation why using shared randomness in the stochastic rounding is able to improve the performance. Without the loss of generality, in the following analysis, we let the quantization step associated with stochastic rounding quantizer $\mathcal{Q}$ be $\delta = 1$. For any $z$ quantized using $\mathcal{Q}$, let $z_f = z - \lfloor z \rfloor$, the variance of quantization error can be expressed as

$$\mathbb{E}\left\| \mathcal{Q}(z) - z \right\|^2 = (1 - z_f)(-z_f)^2 + z_f(1 - z_f)^2 = z_f(1 - z_f) \tag{5}$$

Note that in Moniqua, the term asssociate with quantization error is

$$\mathbb{E}\left\| (q_{k,j} - x_{k,j}) - (q_{k,i} - x_{k,i}) \right\|^2$$

We now show for $\forall x, y \in \mathbb{R}^d$

$$\mathbb{E}\left\| (\mathcal{Q}(x) - x) - (\mathcal{Q}(y) - y) \right\|^2 = \mathbb{E}\left\| \mathcal{Q}(y - x) - (y - x) \right\|^2$$

With out the loss of generality, let $x - \lfloor x \rfloor \le y - \lfloor y \rfloor$. Let $x_f = x - \lfloor x \rfloor$ and $y_f = y - \lfloor y \rfloor$, then

$$\lfloor x + u \rfloor = \lfloor x \rfloor \quad \text{and} \quad \lfloor y + u \rfloor = \lfloor y \rfloor, \text{with probability} \quad \lceil y \rceil - y$$
$$\lfloor x + u \rfloor = \lceil x \rceil \quad \text{and} \quad \lfloor y + u \rfloor = \lceil y \rceil, \text{with probability} \quad x - \lfloor x \rfloor$$
$$\lfloor x + u \rfloor = \lfloor x \rfloor \quad \text{and} \quad \lfloor y + u \rfloor = \lceil y \rceil, \text{with probability} \quad (\lceil x \rceil - x) - (\lceil y \rceil - y)$$

Then we have

$$\mathbb{E}\left\| (\mathcal{Q}(x) - x) - (\mathcal{Q}(y) - y) \right\|^2$$
$$= \mathbb{E}\left\| \left( \delta \left\lfloor \frac{x}{\delta} + u \right\rfloor - x \right) - \left( \delta \left\lfloor \frac{y}{\delta} + u \right\rfloor - y \right) \right\|^2$$
$$= (\lceil y \rceil - y)((\lfloor x \rfloor - x) - (\lfloor y \rfloor - y))^2 + (x - \lfloor x \rfloor)((\lceil x \rceil - x) - (\lceil y \rceil - y))^2$$
$$+ ((\lceil x \rceil - x) - (\lceil y \rceil - y))((\lfloor x \rfloor - x) - (\lceil y \rceil - y))^2$$
$$= (1 - y_f)(x_f - y_f)^2 + (x_f)(x_f - y_f) + (y_f - x_f)(y_f - x_f - 1)^2$$
$$= (1 - y_f + x_f)(y_f - x_f)^2 + (y_f - x_f)(y_f - x_f - 1)^2$$
$$= (1 - y_f + x_f)(y_f - x_f)$$
$$= \mathbb{E}\left\| \mathcal{Q}(y - x) - (y - x) \right\|^2$$

The last equality holds due to equation 5. Next, let

$$\Delta = y - x$$
$$r = \mathcal{Q}(\Delta) - \Delta$$

And let $r_h$ denote $h$-th entry of $r$, let $\Delta_h$ denote $h$-th entry of $\Delta$. We obtain

$$r_h = \mathcal{Q}(\Delta_h) - \Delta_h$$

$$=\delta \begin{cases} -\frac{\Delta_h}{\delta} + \left\lfloor \frac{\Delta_h}{\delta} \right\rfloor + 1, & p_t \leq \frac{\Delta_h}{\delta} - \left\lfloor \frac{\Delta_h}{\delta} \right\rfloor \\ -\frac{\Delta_h}{\delta} + \left\lfloor \frac{\Delta_h}{\delta} \right\rfloor, & \text{otherwise} \end{cases}$$

$$=\delta \begin{cases} -q + 1, & p_t \leq q \\ -q, & \text{otherwise} \end{cases}$$

where

$$q = \frac{\Delta_h}{\delta} - \left\lfloor \frac{\Delta_h}{\delta} \right\rfloor, q \in [0, 1]$$

Based on that, we have

$$\mathbb{E}\left[r_h^2\right] \leq \delta^2((-q+1)^2 q + (-q)^2(1-q))$$
$$= \delta^2 q(1-q)$$
$$\leq \delta^2 \min\{q, 1-q\}$$

Since $\min\{q, 1-q\} \leq \left|\frac{x_h}{\delta}\right|$, we have

$$\mathbb{E}\left[r_h^2\right] \leq \delta^2 \left|\frac{\Delta_h}{\delta}\right| \leq \delta |\Delta_h|$$

Summing over the index $h$ yields,

$$\mathbb{E}\|r\|_2^2 \leq \delta \mathbb{E}\|\Delta\|_1 \leq \sqrt{d}\delta \mathbb{E}\|\Delta\|_2$$

Pushing back $x$ and $r$, we have

$$\mathbb{E}\|\mathcal{Q}(y-x) - (y-x)\|^2 \leq \sqrt{d}\delta \mathbb{E}\|y-x\| = \sqrt{d}\delta \mathbb{E}\|x-y\|$$

Putting it back we have

$$\mathbb{E}\|(\mathcal{Q}(x) - x) - (\mathcal{Q}(y) - y)\|^2 \leq \sqrt{d}\delta \mathbb{E}\|x-y\|$$

Now we can see that the error term is bounded by the distance of two quantized tensor, which, in decentralized training, refers to the distance between two models on adjacent workers. In such a way, the error bound can be reduced since the workers are getting close to each other.

## C    MORE EXPERIMENTAL RESULTS

### C.1    COMPUTE NUMBER OF BITS

In Experiment of *Efficient Moniqua*, we calculate the number of bits in the following way: First, we calculate the total number of bits each worker send out, sum them up and divided by number of epochs, and we get the average bandwidth consumption $\overline{BW}$ of the whole system in each epoch. Then we compute the number of bits required for each number in the following way (note that every worker has 2 neighbors in a ring topology):

$$\#bits = \frac{\overline{BW}}{\#\text{neighbors} \cdot \#\text{workers} \cdot \#\text{params of model}}$$

In our experiments, #neighbors=2, #workers=8. For VGG16, #params of model= 15,245,130 while for ResNet110, #params of model=1,146,842. We formalize the results in Table C.1 [8].

### C.2    VARIOUS QUANTIZERS

In this section, we will verify Moniqua is applicable to other quantizers aside from linear quantizer as shown in the paper. We test it on two more quantizers:

1. **Nearest Rounding (Biased)**

$$\mathcal{Q}(x) = \delta \left\lfloor \frac{x}{\delta} + 0.5 \right\rfloor$$

where $\delta$ is the quantization step as defined in the linear quantizer. In this experiment, we set $\delta = 0.01$, the same value as we used in the paper with stochastic rounding.

---

[8]Note that we only put results that's 'close to the limit of Moniqua' here

Table 1: Wall Clock Time consumption (Seconds)/Epoch in average under different network in Experiment of Evaluation of Moniqua.

|  | 100mbps/0.15ms | 20mbps/0.15ms | 100mbps/10ms | Extra Memory |
|---|---|---|---|---|
| Centralized | 38.92 | 206.14 | 343.28 | N/A |
| D-PSGD | 36.25 | 189.48 | 310.98 | N/A |
| DCD-PSGD | 32.99 | 105.40 | 202.42 | 20.4 MB |
| ECD-PSGD | 31.96 | 105.26 | 202.04 | 20.4 MB |
| ChocoSGD | 32.03 | 105.18 | 201.18 | 20.4 MB |
| DeepSqueeze | 30.01 | 103.67 | 193.92 | 13.6 MB |
| Moniqua | 22.42 | 95.08 | 184.86 | 0 B |

Table 2: Bandwidth consumption under different $\theta$ and $\delta$ when applying linear quantizer in Moniqua

| MODEL | MOD PARAM $\theta$ | QUANT STEP | BYTES/EPOCH | AVG BITS |
|---|---|---|---|---|
| VGG16 | NONE | NONE | 45594MB | 32 |
|  | 1.0 | 0.01 | 5206MB | 3.65 |
|  | 0.08 | 0.01 | 5192MB | 3.64 |
| RESNET110 | NONE | NONE | 3430MB | 32 |
|  | 2.0 | 0.01 | 609MB | 5.67 |
|  | 1.3 | 0.01 | 608MB | 5.67 |

2. **Randomized Gossip (Non-linear)**

$$\mathcal{Q}(x) = \begin{cases} x, & \text{with probability} \quad p \\ 0, & \text{with probability} \quad 1-p \end{cases}$$

In this experiment, we set $p = 0.7$.

We train ResNet110 on CIFAR10, and plot the results in Figure 5(c). We can see that the training curves of using three quantizers are all aligned with D-PSGD with full-precision communication. Note that in the paper we show that previous work cannot perserve the aligned curve even with stochastic rounding (unbiased), thus we are not comparing them here.

C.3 MORE RESULTS ON DIFFERENT HYPERPARAMETERS

In this experiment, we plot more result of training ResNet110 and VGG16 on CIFAR10 under different $\delta$ and $\theta$ in the experiment of aggressive quantization. And we plot the results in Figure 5(a) and Figure 5(b).

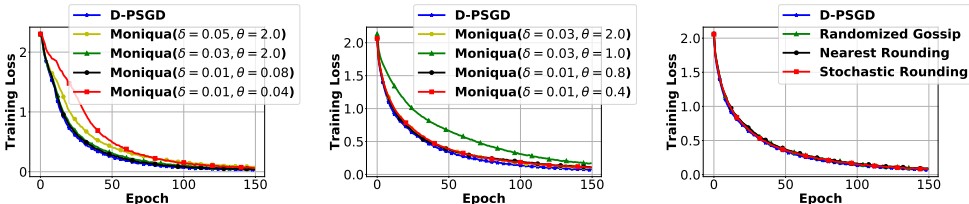

(a) Performance of Moniqua on VGG16 under different $\theta$ and $\delta$

(b) Performance of Moniqua on ResNet110 under different $\theta$ and $\delta$

(c) Performance of algorithms under different quantizer

## C.4 PERFORMANCE ON THE TESTSET UNDER AGGRESSIVE QUANTIZATION

We report the results in experiment of "Aggressive Quantization" and report the test error and test accuracy in the Figure 5 and Figure 6.

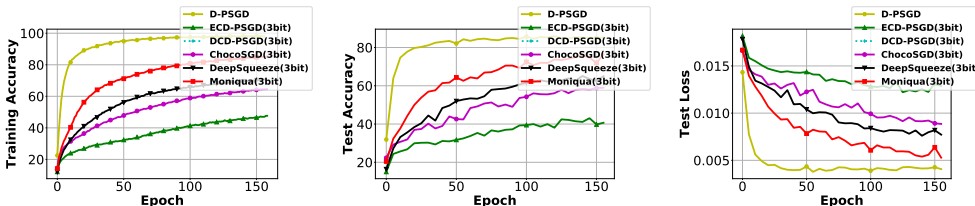

(d) Training Accuracy under differ-(e) Test Accuracy under different (f) Test Loss under different algo-ent algorithms with 3-bit communi-algorithms with 3-bit communica-rithms with 3-bit communication cation tion

Figure 5: More statistics from Experiment of Aggressive Quantization under 3-bit communication

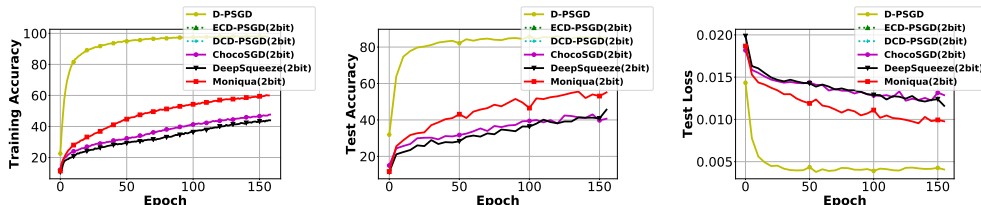

(a) Training Accuracy under differ-(b) Test Accuracy under different (c) Test Loss under different algo-ent algorithms with 2-bit communi-algorithms with 2-bit communica-rithms with 2-bit communication cation tion

Figure 6: More statistics from Experiment of Aggressive Quantization under 2-bit communication

## C.5 DECREASING STEP SIZE AND CONSENSUS ERROR

In this subsection, we provide more experimental results with decreasing step size. We also provide and discuss results on consensus error in this experiment. We run the experiments in the following setting:

**Models, Datasets and Hyperparameters.** We launch 8 workers connected using a ring network. We train ResNet110 and ResNet18 on CIFAR10. The hyperparameters of Moniqua are: ResNet110 (Initial step size $= 0.05$, $\theta = 3.0$, batch size $= 128$, weight decay $= 3e - 4$, and momentum $= 0.9$) and ResNet18 (Initial step size $= 0.1$, $\theta = 2.5$, batch size $= 128$, weight decay $= 1e - 4$, and momentum $= 0.9$). Step size is decreased (times a 0.1 factor) every 30 epochs. To be consistent with the original paper, we use the stochastic rounding to quantize each number.

**Results of Decreasing Step Size** We plot the results of test accuracy in Figure 7. We can see from Figure 7(a) that Moniqua requires at least 6 bits to achieve the comparable (accuracy drop $< 0.3\%$) test accuracy as the baseline (D-PSGD with 32 bits). Once the numeber bits decrease to 5, there is a accuracy gap between Moniqua and D-PSGD. On the other hand, other baselines including DeepSqueeze, ChocoSGD and DCD/ECD-PSGD are not able to achieve comparable test accuracy with 6 bits. Similarily, we can see from Figure 7(b) that when training ResNet18 with 4 bits, Moniqua is able to achieve comparable test accuracy after 120 epochs, while other baselines suffer a certain accuracy gap.

We also plot the test accuracy of different algorithms under different bit-level communication in Figure 8 (ResNet110). We can see that compared to the baselines, Moniqua is generally robust to low

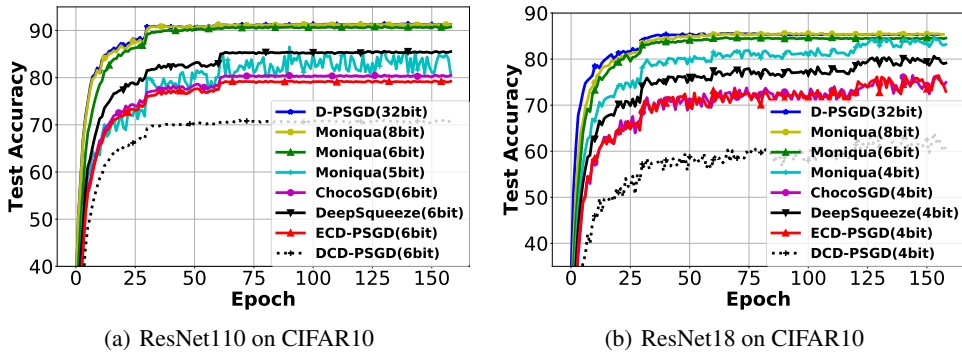

(a) ResNet110 on CIFAR10        (b) ResNet18 on CIFAR10

Figure 7: Test Accuracy and Consensus Error of Moniqua under decreasing step size.

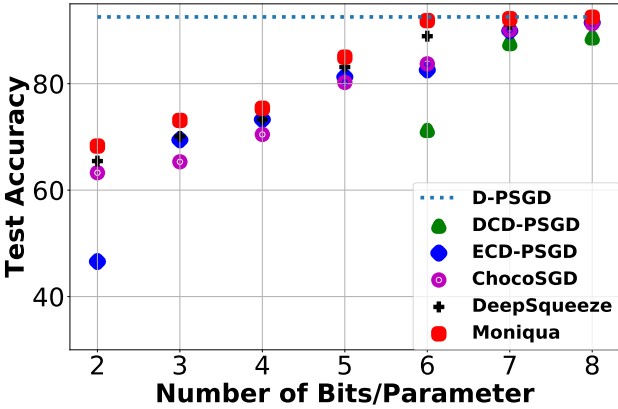

Figure 8: Test Accuracy of different algorithms on training ResNet110 on CIFAR10.

bits-budget. (Some of the dots are missing for some algorithms, that means they do not converge under the corresponding bits-budget.)

**Results of Consensus Error.** To better measure the behaviour of workers reaching consensus, we define the consensus error at iteration $k$: $\mathcal{C}_k$ as follows (Notations are the same as in the original paper):

$$\mathcal{C}_k = \frac{1}{nk} \sum_{t=0}^{k-1} \sum_{i=1}^{n} \left\| \frac{1}{n} \sum_{j=1}^{n} x_{t,j} - x_{t,i} \right\|^2 \tag{6}$$

Note that $\mathcal{C}_k$ is essentially the running average of distance among workers and the averaged model. Trivially, a decreasing $\mathcal{C}_k$ indicates the workers are reaching consensus. We measure the consensus error in three aspects. We first provide the results of the original paper in Figure 9(a), where constant step size is adopted. We can see that even with extremely small number of bits as used in "Aggressive Quantization", all the workers are able to reach consensus. We further plot consensus error under the setting where decreasing step size is adopted as defined in this section. We can see in Figure 9(b) and Figure 9(c) that all the workers are able to reach consensus.

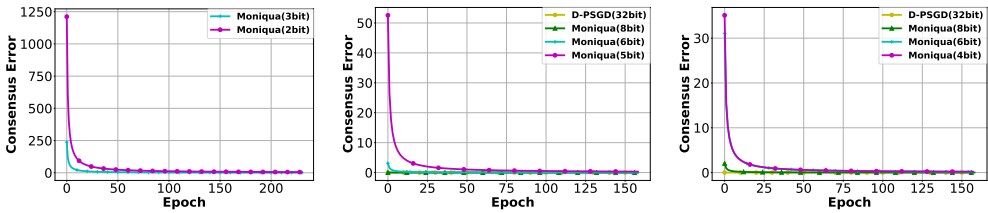

(a) Consensus Error on ResNet110 (b) Consensus Error on ResNet110 (c) Consensus Error on ResNet18
with constant step size with decreasing step size with decreasing step size

Figure 9: Consensus Error of Moniqua with different number of bits.

## D  WHY NAIVE QUANTIZATION FAILS IN D-PSGD (PROOF TO THEOREM 1)

The update rule of naive quantization on D-PSGD is

$$x_{k+1,i} = x_{k,i}W_{ii} + \sum_{j=1,j\neq i}^{n} \mathcal{Q}(x_{k,j})W_{ji} - \alpha_k\widetilde{g}_{k,i} = x_{k,i} + \sum_{j=1,j\neq i}^{n} (\mathcal{Q}(x_{k,j}) - x_{k,i})W_{ji} - \alpha_k\widetilde{g}_{k,i}$$

where $\alpha_k$ is allowed to vary with any policy. Let

$$X_k = [x_{k,1}, \cdots, x_{k,n}] \in \mathbb{R}^{d\times n}$$

$$\Omega_k = \left[\sum_{j\neq 1} W_{j1}\left(\mathcal{Q}(x_{k,j}) - x_{k,1}\right), \cdots, \sum_{j\neq n} W_{jn}\left(\mathcal{Q}(x_{k,j}) - x_{k,n}\right)\right] \in \mathbb{R}^{d\times n}$$

$$\widetilde{G}_k = [\widetilde{g}_{k,1}, \cdots, \widetilde{g}_{k,n}] \in \mathbb{R}^{d\times n}$$

by rewritting the update rule, we obtain

$$X_{k+1} = X_k + \Omega_k - \alpha_k\widetilde{G}_k$$

Let $Y_k = X_k - x^*\mathbb{1}_n^\top$, and considering the fact that $\nabla f(x) = x - \delta/2 = x - x^*$, we can rewrite the update rule as

$$Y_{k+1}e_i = Y_ke_i + \Omega_ke_i - \alpha_kY_ke_i + \alpha_k\left(\widetilde{G}_k - G_k\right)e_i$$

where $\left(\widetilde{G}_k - G_k\right)$ denotes variance in the gradient sampling.

Suppose that by using the update rule of naive quantization, worker $i$ converges to $x^*$. Then there must exist a $K$ such that $\forall k \geq K$,

$$\mathbb{E}\left\|Y_{k+1}e_i\right\|^2 \leq \mathbb{E}\left\|Y_ke_i\right\|^2 < \frac{\phi^2\delta^2}{8(1+\phi^2)} \tag{7}$$

Next we show that this assumption lets us derive a contradiction. Firstly, considering the property of linear quantizer,

$$\frac{\delta^2}{4} \leq \mathbb{E}\left\|\mathcal{Q}(x_{k,i}) - x^*\right\|^2 \leq 2\mathbb{E}\left\|\mathcal{Q}(x_{k,i}) - x_{k,i}\right\|^2 + 2\mathbb{E}\left\|x_{k,i} - x^*\right\|^2$$

As a result

$$\mathbb{E}\left\|\mathcal{Q}(x_{k,i}) - x_{k,i}\right\|^2 \geq \frac{\delta^2}{8} - \frac{\phi^2\delta^2}{8(1+\phi^2)} = \frac{\delta^2}{8(1+\phi^2)}$$

Since $\mathcal{Q}$ is unbiased, that means $\mathbb{E}[\mathcal{Q}(x) - x] = 0$, then we have

$$\mathbb{E}\left\|\Omega_ke_i\right\|^2$$

$$
\begin{aligned}
=&\mathbb{E}\left\|\sum_{j\neq i}W_{ji}\left(\mathcal{Q}(x_{k,j})-x_{k,i}\right)\right\|^2\\
=&\sum_{j\in\mathcal{N}_i}W_{ji}^2\mathbb{E}\left\|\left(\mathcal{Q}(x_{k,j})-x_{k,i}\right)\right\|^2+\sum_{m\neq n\neq i}\mathbb{E}\left\langle\left(\mathcal{Q}(x_{k,m})-x_{k,i}\right)W_{mi},\left(\mathcal{Q}(x_{k,n})-x_{k,i}\right)W_{ni}\right\rangle\\
\geq&\phi^2\sum_{j\in\mathcal{N}_i}\mathbb{E}\left\|\left(\mathcal{Q}(x_{k,j})-x_{k,i}\right)\right\|^2+\sum_{m\neq n\neq i}\mathbb{E}\left\langle\left(\mathcal{Q}(x_{k,m})-x_{k,i}\right)W_{mi},\left(\mathcal{Q}(x_{k,n})-x_{k,i}\right)W_{ni}\right\rangle\\
\overset{(*)}{=}&\phi^2\sum_{j\in\mathcal{N}_i}\mathbb{E}\left\|\mathcal{Q}(x_{k,j})-x_{k,i}\right\|^2\\
\geq&\frac{\phi^2\delta^2}{8(1+\phi^2)}
\end{aligned}
$$

where step $(*)$ holds due to unbiased quantizer. Putting it back to the update rule, we obtain

$$
\begin{aligned}
\mathbb{E}\left\|Y_{k+1}e_i\right\|^2=&\mathbb{E}\left\|\left(Y_k+\Omega_k-\alpha_kY_k+\alpha_k\left(\widetilde{G}_k-G_k\right)\right)e_i\right\|^2\\
\overset{(*)}{=}&\mathbb{E}\left\|(1-\alpha_k)Y_ke_i\right\|^2+\mathbb{E}\left\|\Omega_ke_i\right\|^2+\mathbb{E}\left\|\alpha_k\left(\widetilde{G}_k-G_k\right)e_i\right\|^2\\
\geq&\mathbb{E}\left\|\Omega_ke_i\right\|^2\\
\geq&\frac{\phi^2\delta^2}{8(1+\phi^2)}
\end{aligned}
$$

where cross terms in the $(*)$ step are all 0 due to the unbiased quantizer and unbiased sampling of the gradient. Her we obtain the contradictory that $\frac{\phi^2\delta^2}{8(1+\phi^2)}\leq\mathbb{E}\left\|x_{k+1}-x^*\right\|^2<\frac{\phi^2\delta^2}{8(1+\phi^2)}$. That being said, for $\forall k,i$

$$
\mathbb{E}\left\|x_{k,i}-x^*\right\|^2=\mathbb{E}\left\|\nabla f(x_{k,i})\right\|^2\geq\frac{\phi^2\delta^2}{8(1+\phi^2)}
$$

Thus we complete the proof.

# E    A MARKOV CHAIN ANALYSIS ON THE COMMUNICATION

To better understand how the parallel workers reach consensus over a communication matrix, in this section we use theory from the analysis of Markov Chains to obtain some useful lemmas for proof of Moniqua on D-PSGD and AD-PSGD.

Since the communication matrix $W$ is doubly stochastic (each row and column sum to 1), it has the same structure as the transition matrix of a Markov Chain with $\frac{\mathbb{1}_n}{n}$ as its the stationary distribution $\left(W\frac{\mathbb{1}_n}{n}=\frac{\mathbb{1}_n}{n}\right)$. Now let $t_{\text{mix}}$ and $d(t)$ denote the mixing time and maximal distance between initial state and stationary distribution as defined in Markov Chain theory.[9]

## E.1    D-PSGD

In D-PSGD, the communication matrix is fixed during the training. That makes it perfectly aligned with the structure of a Markov Chain. As a result, we obtain the following lemma:

**Lemma 1**

$$
\left\|W^t\left(I-\frac{\mathbb{1}_n\mathbb{1}_n^\top}{n}\right)\right\|_1\leq 2\cdot 2^{-\left\lfloor\frac{t}{t_{\text{mix}}}\right\rfloor}
$$

**Proof** *For $\forall x\in\mathbb{R}^d$, let $u\in\mathbb{R}^d$ be such a vector that every entry of $u$ is the positive entry of $x$ and 0 otherwise. Let $v\in\mathbb{R}^d$ be such a vector that every entry of $v$ is the absolute value of negative entry of $x$ and 0 otherwise. The setting above means $x=u-v$. For example,*

$$
x=[2,-1]^\top
$$

---

[9]Here we are using notation from Chapter 4.5 of *Markov Chains and Mixing Times* (Levin 2009), available at https://pages.uoregon.edu/dlevin/MARKOV/markovmixing.pdf

$$u = [2, 0]^\top$$
$$v = [0, 1]^\top$$

*And we have*

$$\left\| W^t \left( I - \frac{\mathbb{1}_n \mathbb{1}_n^\top}{n} \right) x \right\|_1$$

$$= \left\| W^t \left( I - \frac{\mathbb{1}_n \mathbb{1}_n^\top}{n} \right) (u - v) \right\|_1$$

$$\leq \left\| W^t \left( I - \frac{\mathbb{1}_n \mathbb{1}_n^\top}{n} \right) u \right\|_1 + \left\| W^t \left( I - \frac{\mathbb{1}_n \mathbb{1}_n^\top}{n} \right) v \right\|_1$$

$$= \mathbb{1}_n^\top u \left\| W^t \frac{u}{\mathbb{1}_n^\top u} - \frac{\mathbb{1}_n}{n} \right\|_1 + \mathbb{1}_n^\top v \left\| W^t \frac{v}{\mathbb{1}_n^\top v} - \frac{\mathbb{1}_n}{n} \right\|_1$$

$$\leq 2(\mathbb{1}_n^\top u + \mathbb{1}_n^\top v) d(t)$$

$$\leq 2d(t) \left\| x \right\|_1$$

*Considering the definition of L1-norm, we have*

$$\left\| W^t \left( I - \frac{\mathbb{1}_n \mathbb{1}_n^\top}{n} \right) \right\|_1 = \max \frac{\left\| W^t \left( I - \frac{\mathbb{1}_n \mathbb{1}_n^\top}{n} \right) x \right\|_1}{\left\| x \right\|_1} \leq 2d(t)$$

*According to a well-known results on the theory of Markov Chains,[10] $d(lt_{\mathrm{mix}}) \leq 2^{-l}$ holds for any non-negative integer l, so we have*

$$\left\| W^t \left( I - \frac{\mathbb{1}_n \mathbb{1}_n^\top}{n} \right) \right\|_1 \leq 2d(t) \leq 2d \left( \frac{t}{t_{\mathrm{mix}}} \cdot t_{\mathrm{mix}} \right) \leq 2d \left( \left\lfloor \frac{t}{t_{\mathrm{mix}}} \right\rfloor t_{\mathrm{mix}} \right) \leq 2 \cdot 2^{-\left\lfloor \frac{t}{t_{\mathrm{mix}}} \right\rfloor}$$

*That completes the proof.*

Additionally, based on standard results in the theory of reversible Markov Chains, we also have[11]

$$t_{\mathrm{mix}} \leq \log \left( \frac{1}{\frac{1}{4} \cdot \frac{1}{n}} \right) \frac{1}{1 - \rho} \leq \frac{\log(4n)}{1 - \rho}.$$

### E.2   AD-PSGD

Note that unlike D-PSGD, here $W_k$ can be different at each update step and usually each individually have spectral radius $\rho = 1$, so we can't expect to get a bound in terms of a bound on the spectral gap as we did in Theorems 2 and 3. Instead, we require the following condition, which is inspired by the literature on Markov chain Monte Carlo methods: for some constant $t_{\mathrm{mix}}$ (here $t_{\mathrm{mix}}$ is the same as $t_{\mathrm{mix}}$ in the paper) and for any $k$ and any non-negative vector $\mu \in \mathbb{R}^d$ such that $\mathbb{1}_n^\top \mu = 1$, it must hold that

$$\left\| \left( \prod_{i=1}^{t_{\mathrm{mix}}} W_{k+i} \right) \mu - \frac{\mathbb{1}_n}{n} \right\|_1 \leq \frac{1}{2}.$$

We call this constant $t_{\mathrm{mix}}$ because it is effectively the *mixing time* of the time-inhomogeneous Markov chain with transition probability matrix $W_k$ at time $k$. Note that this condition is more general than those used in previous work on AD-PSGD because it does not require that the $W_k$ are sampled independently or in an unbiased manner. Based on the above analysis, we can prove the following lemma, which is analogous to the lemma used in the synchronous case.

**Lemma 2** *For any $k \geq 0$ and for any $b \geq a \geq 0$, there exists $t_{\mathrm{mix}}$ such that*

$$\left\| \prod_{q=a}^{b} W_q \left( I - \frac{\mathbb{1}_n \mathbb{1}_n^\top}{n} \right) \right\|_1 \leq 2 \cdot 2^{-\left\lfloor \frac{b-a+1}{t_{\mathrm{mix}}} \right\rfloor}$$

---

[10] Again, see *Markov Chains and Mixing Times* for more details.

[11] Detailed analysis and proofs of this result can be found in chapter 12.2 of *Markov Chains and Mixing Times*.

**Proof** *Note that for any $x \in \mathbb{R}^d$, and let $u$ and $v$ be two vectors having same definition as in Lemma 1 with respect to $x$, then we have for any $k$*

$$
\begin{aligned}
\left\| \prod_{q=1}^{t_{\mathrm{mix}}} W_{q+k} \left( I - \frac{\mathbb{1}_n \mathbb{1}_n^\top}{n} \right) x \right\|_1 &= \left\| \prod_{q=1}^{t_{\mathrm{mix}}} W_{q+k} \left( I - \frac{\mathbb{1}_n \mathbb{1}_n^\top}{n} \right) (u - v) \right\|_1 \\
&\leq \left\| \prod_{q=1}^{t_{\mathrm{mix}}} W_{q+k} \left( I - \frac{\mathbb{1}_n \mathbb{1}_n^\top}{n} \right) u \right\|_1 + \left\| \prod_{q=1}^{t_{\mathrm{mix}}} W_{q+k} \left( I - \frac{\mathbb{1}_n \mathbb{1}_n^\top}{n} \right) v \right\|_1 \\
&= \mathbb{1}_n^\top u \left\| \prod_{q=1}^{t_{\mathrm{mix}}} W_{q+k} \frac{u}{\mathbb{1}_n^\top u} - \frac{\mathbb{1}_n}{n} \right\|_1 + \mathbb{1}_n^\top v \left\| \prod_{q=1}^{t_{\mathrm{mix}}} W_{q+k} \frac{v}{\mathbb{1}_n^\top v} - \frac{\mathbb{1}_n}{n} \right\|_1 \\
&\leq \frac{1}{2} (\mathbb{1}_n^\top u + \mathbb{1}_n^\top v) \\
&\leq \frac{1}{2} \|x\|_1
\end{aligned}
$$

*Considering the definition of the induced $\ell_1$ operator norm, we have*

$$
\left\| \prod_{q=1}^{t_{\mathrm{mix}}} W_{q+k} \left( I - \frac{\mathbb{1}_n \mathbb{1}_n^\top}{n} \right) \right\|_1 = \max_x \frac{\left\| \prod_{q=1}^{t_{\mathrm{mix}}} W_{q+k} \left( I - \frac{\mathbb{1}_n \mathbb{1}_n^\top}{n} \right) x \right\|_1}{\|x\|_1} \leq \frac{1}{2}
$$

*As a result, from the submultiplicativity of the matrix induced norm, we obtain*

$$
\begin{aligned}
&\left\| \prod_{q=a}^{b} W_q \left( I - \frac{\mathbb{1}_n \mathbb{1}_n^\top}{n} \right) \right\|_1 \\
&\leq \left\| \prod_{q=1}^{t_{\mathrm{mix}}} W_{a-1+q} \left( I - \frac{\mathbb{1}_n \mathbb{1}_n^\top}{n} \right) \right\|_1 \cdots \left\| \prod_{q=1}^{t_{\mathrm{mix}}} W_{\cdots+q} \left( I - \frac{\mathbb{1}_n \mathbb{1}_n^\top}{n} \right) \right\|_1 \cdot \left\| \prod_{q=1}^{t_r} W_{\cdots+q} \left( I - \frac{\mathbb{1}_n \mathbb{1}_n^\top}{n} \right) \right\|_1 \\
&\leq 2^{-\left\lfloor \frac{b-a+1}{t_{\mathrm{mix}}} \right\rfloor} \left\| \prod_{q=1}^{t_r} W_{\cdots+q} \left( I - \frac{\mathbb{1}_n \mathbb{1}_n^\top}{n} \right) \right\|_1
\end{aligned}
$$

*where $t_r = (b - a + 1) \bmod t_{\mathrm{mix}}$. Note that*

$$
\left\| \prod_{q=1}^{t_r} W_q \left( I - \frac{\mathbb{1}_n \mathbb{1}_n^\top}{n} \right) \right\|_1 \leq 1 - \frac{1}{n} + (n-1)\frac{1}{n} = 2 - \frac{2}{n} \leq 2
$$

*Putting it back we obtain*

$$
\left\| \prod_{q=a}^{b} W_{\cdots+q} \left( I - \frac{\mathbb{1}_n \mathbb{1}_n^\top}{n} \right) \right\|_1 \leq 2 \cdot 2^{-\left\lfloor \frac{b-a+1}{t_{\mathrm{mix}}} \right\rfloor}
$$

*That completes the proof.*

Note that in the analysis of Moniqua on AD-PSGD (Section H), we will use this lemma as an assumption.

## F  MONIQUA ON D-PSGD (PROOF TO THEOREM 2)

**Consistent with linear and non-linear quantizer**  Here we briefly explain why using $\theta \cdot \mathcal{Q}_\delta \left( \frac{x}{\theta} \bmod 1 \right)$ instead of $\mathcal{Q}_\delta (x \bmod \theta)$ for theoretical analysis and how it covers both linear and non-linear quantizers. Note that typically, a linear quantizer has:

$$
\| \mathcal{Q}_\delta(x) - x \|_\infty \leq \delta, \quad \forall x \in \mathbb{R}^d
$$

while a non-linear quantizer has

$$\|\mathcal{Q}_\delta(x) - x\|_\infty \le \delta\|x\|_\infty, \quad \forall x \in \mathbb{R}^d$$

so that for linear quantizer, with a given $x \le \theta$:

$$\left\|\theta \cdot \mathcal{Q}_\delta\left(\frac{x}{\theta} \bmod 1\right) - x\right\|_\infty = \theta\left\|\mathcal{Q}_\delta\left(\frac{x}{\theta} \bmod 1\right) - \left(\frac{x}{\theta} \bmod 1\right)\right\|_\infty \le \theta\delta = \theta\delta$$

And for non-linear quantizer, with a given $x \le \theta$:

$$\left\|\theta \cdot \mathcal{Q}_\delta\left(\frac{x}{\theta} \bmod 1\right) - x\right\|_\infty = \theta\left\|\mathcal{Q}_\delta\left(\frac{x}{\theta} \bmod 1\right) - \left(\frac{x}{\theta} \bmod 1\right)\right\|_\infty \le \theta\delta \cdot 1 = \theta\delta$$

As a result, we can use the same bound $\theta\delta$ for quantizers with both of the properties, which we will show in the rest of the proof.

### F.1 PROOF TO THEOREM 2

**Proof** *For convenience, we define the following notation*

$$X_k = [x_{k,1}, \cdots, x_{k,n}], \qquad Q_k = [q_{k,1}, \cdots, q_{k,n}]$$
$$\widetilde{G}_k = [\widetilde{g}_{k,1}, \cdots, \widetilde{g}_{k,n}], \qquad G_k = [g_{k,1}, \cdots, g_{k,n}]$$
$$\overline{X} = X\frac{\mathbb{1}_n}{n}, \forall X \in \mathbb{R}^{d \times n}, \qquad \Omega_k = (Q_k - X_k)(W - I)$$

*where $g_{k,i}$ denotes gradient computed via the whole dataset $\mathcal{D}_i$ and $x_{k,i}$*

*From a local view, the update rule of Algorithm 1 on worker $i$ at iteration $k$ can be written as*

$$x_{k+1,i} \leftarrow x_{k,i} + \sum_{j \in \mathcal{N}_i} (q_{k,j} - q_{k,i}) W_{ji} - \alpha\widetilde{g}_{k,i}$$

*which is equivalent to*

$$x_{k+1,i} = x_{k,i} + \sum_{j=1}^n (x_{k,j} - x_{k,i}) W_{ji} - \alpha\widetilde{g}_{k,i} + \sum_{j=1}^n ((q_{k,j} - x_{k,j}) - (q_{k,i} - x_{k,i})) W_{ji} \quad (8)$$

*From a global view, the update rule can be written as*

$$X_{k+1} = X_k + Q_k(W - I) - \alpha\widetilde{G}_k = X_kW - \alpha\widetilde{G}_k + (Q_k - X_k)(W - I) \quad (9)$$

*From Lemma 5 we have*

$$\frac{1}{K}\sum_{k=0}^{K-1} \mathbb{E}\left\|\nabla f(\overline{X}_k)\right\|^2 \le \frac{4(f(0) - f^*)}{\alpha K} + \frac{2\alpha L}{n}\sigma^2 + \frac{8\alpha^2 L^2\left(\sigma^2 + 3\varsigma^2\right)}{(1-\rho)^2}$$

$$+ \frac{8L^2}{nK(1-\rho)^2}\sum_{k=1}^{K-1} \mathbb{E}\|\Omega_k\|_F^2$$

*Note that*

$$\sum_{k=0}^{K-1} \mathbb{E}\|\Omega_k\|_F^2 = \sum_{k=0}^{K-1}\sum_{i=1}^n \mathbb{E}\left\|\sum_{j=1}^n ((q_{k,j} - x_{k,j}) - (q_{k,i} - x_{k,i})) W_{ji}\right\|^2$$

$$\overset{Lemma\ 3}{\le} 4\sum_{k=0}^{K-1}\sum_{i=1}^n \delta^2\theta^2 d \le \alpha^2 G_\infty^2 dnK$$

*The last step holds because $\delta\theta = \frac{1}{2}\alpha G_\infty$. Pushing it back we obtain*

$$\frac{1}{K}\sum_{k=0}^{K-1} \mathbb{E}\left\|\nabla f(\overline{X}_k)\right\|^2 \le \frac{4(f(0) - f^*)}{\alpha K} + \frac{2\alpha L}{n}\sigma^2 + \frac{8\alpha^2 L^2\left(\sigma^2 + 3\varsigma^2\right)}{(1-\rho)^2} + \frac{8\alpha^2 G_\infty^2 dL^2}{(1-\rho)^2}$$

*By setting $\alpha = \frac{1}{\varsigma^{\frac{2}{3}} K^{\frac{1}{3}} + \sigma\sqrt{\frac{K}{n}} + 2L}$, we have*

$$\frac{1}{K} \sum_{k=0}^{K-1} \mathbb{E} \left\| \nabla f(\overline{X}_k) \right\|^2 \leq \frac{8(f(0) - f^*)L}{K} + \frac{4\sigma(f(0) - f^* + L/2)}{\sqrt{nK}} + \frac{4\varsigma^{\frac{2}{3}}(f(0) - f^*)}{K^{\frac{2}{3}}}$$

$$+ \frac{8L^2\sigma^2 n}{(1-\rho)^2(\sigma^2 K + 4nL^2)} + \frac{24L^2\varsigma^{\frac{2}{3}}}{(1-\rho)^2 K^{\frac{2}{3}}} + \frac{8L^2 G_\infty^2 dn}{(1-\rho)^2(\sigma^2 K + 4nL^2)}$$

$$\lesssim \frac{1}{K} + \frac{\sigma}{\sqrt{nK}} + \frac{\varsigma^{\frac{2}{3}}}{K^{\frac{2}{3}}} + \frac{\sigma^2 n}{\sigma^2 K + n} + \frac{G_\infty^2 dn}{\sigma^2 K + n}$$

*That completes the proof of Theorem 2.*

### F.2 Lemma for Moniqua on D-PSGD

**Lemma 3** *If $\|x_{t,i} - x_{t,j}\|_\infty \leq \theta, \forall i,j$ holds at iteration $t$, then*

$$\left\| \sum_{j=1}^n \left( (q_{t,j} - x_{t,j}) - (q_{t,i} - x_{t,i}) \right) W_{ji} \right\|_\infty \leq 2\delta\theta$$

**Proof**

$$\left\| \sum_{j=1}^n \left( (q_{t,j} - x_{t,j}) - (q_{t,i} - x_{t,i}) \right) W_{ji} \right\|_\infty$$

$$\leq \sum_{j=1}^n W_{ji} \left\| (q_{t,j} - x_{t,j}) - (q_{t,i} - x_{t,i}) \right\|_\infty$$

$$= \sum_{j=1}^n W_{ji} \left\| \theta \mathcal{Q}\left( \frac{x_{t,j}}{\theta} \bmod 1 \right) - \theta \mathcal{Q}\left( \frac{x_{t,i}}{\theta} \bmod 1 \right) - (x_{t,j} - x_{t,i}) \right\|_\infty$$

$$= \sum_{j=1}^n W_{ji} \left\| \theta \mathcal{Q}\left( \frac{x_{t,j}}{\theta} \bmod 1 \right) - \theta \mathcal{Q}\left( \frac{x_{t,j}}{\theta} \bmod 1 \right) - \theta \left( \frac{x_{t,j} - x_{t,i}}{\theta} \bmod 1 \right) \right\|_\infty$$

$$= \sum_{j=1}^n W_{ji} \left\| \theta \mathcal{Q}\left( \frac{x_{t,j}}{\theta} \bmod 1 \right) - \theta \left( \frac{x_{t,j}}{\theta} \bmod 1 \right) - \left( \theta \mathcal{Q}\left( \frac{x_{t,i}}{\theta} \bmod 1 \right) - \theta \left( \frac{x_{t,i}}{\theta} \bmod 1 \right) \right) \right\|_\infty$$

$$\leq \sum_{j=1}^n W_{ji} \left\| \theta \mathcal{Q}\left( \frac{x_{t,j}}{\theta} \bmod 1 \right) - \theta \left( \frac{x_{t,j}}{\theta} \bmod 1 \right) \right\|_\infty$$

$$+ \sum_{j=1}^n W_{ji} \left\| \theta \mathcal{Q}\left( \frac{x_{t,i}}{\theta} \bmod 1 \right) - \theta \left( \frac{x_{t,i}}{\theta} \bmod 1 \right) \right\|_\infty$$

$$\leq 2\delta\theta$$

**Lemma 4** *In any iteration $k \geq 0$, and for any two worker $i$ and $j$, we have:*

$$\|X_k(e_i - e_j)\|_\infty \leq \theta = \frac{2\log(16n)}{1 - \rho} \alpha G_\infty$$

**Proof** *We use mathematical induction to prove this:*

*I. When $k = 0$, $\|X_0(e_i - e_j)\|_\infty = 0 \leq \theta, \forall i,j$*

*II. Suppose for $\|X_k(e_i - e_j)\|_\infty \leq \theta, k \geq 0, \forall i,j$, we have*

$$\|X_{k+1}(e_i - e_j)\|_\infty = \left\| \left( X_k W - \alpha \widetilde{G}_k + \Omega_k \right)(e_i - e_j) \right\|_\infty$$

$$\overset{X_0 = 0}{=} \left\| \sum_{t=0}^k \left( -\alpha \widetilde{G}_t + \Omega_t \right) W^{k-t}(e_i - e_j) \right\|_\infty$$

$$\leq \sum_{t=0}^{k} \left\| \left( -\alpha \widetilde{G}_t + \Omega_t \right) W^{k-t} (e_i - e_j) \right\|_{\infty}$$

$$\leq \sum_{t=0}^{k} \left\| -\alpha \widetilde{G}_t + \Omega_t \right\|_{1,\infty} \left\| W^{k-t}(e_i - e_j) \right\|_1$$

$$\leq \sum_{t=0}^{k} \left( \alpha \left\| \widetilde{G}_t \right\|_{1,\infty} + \|\Omega_t\|_{1,\infty} \right) \left\| W^{k-t}(e_i - e_j) \right\|_1$$

$$\overset{induction\ hypothesis}{\leq} (\alpha G_\infty + 2\delta\theta) \sum_{t=0}^{k} \left\| W^{k-t}(e_i - e_j) \right\|_1$$

$$\leq (\alpha G_\infty + 2\delta\theta) \sum_{t=0}^{\infty} \left\| W^{t}(e_i - e_j) \right\|_1$$

*For any $t \geq 0$, on one hand*

$$\left\| W^t(e_i - e_j) \right\|_1 \leq \sqrt{n} \left\| W^t(e_i - e_j) \right\|_2 \leq \sqrt{n} \left\| W^t e_i - \frac{\mathbb{1}_n}{n} \right\| + \sqrt{n} \left\| W^t e_j - \frac{\mathbb{1}_n}{n} \right\| \leq 2\sqrt{n}\rho^t$$

*where the last step holds due to the diagonalizability of $W$. On the other hand,*

$$\left\| W^t(e_i - e_j) \right\|_1 \leq \mathbb{1}_n^\top W^t e_i + \mathbb{1}_n^\top W^t e_i = \mathbb{1}_n^\top e_i + \mathbb{1}_n^\top e_j = 2$$

*So*

$$\left\| W^t(e_i - e_j) \right\|_1 \leq \min\{2\sqrt{n}\rho^t, 2\}$$

*Let $T_0 = \left\lceil \frac{-\log(\sqrt{n})}{\log(\rho)} \right\rceil$, so that $n\rho^{T_0} \leq 1$, then we have*

$$\sum_{t=0}^{\infty} \left\| W^t(e_i - e_j) \right\|_1 = \sum_{t=0}^{T_0-1} \left\| W^t(e_i - e_j) \right\|_1 + \sum_{t=T_0}^{\infty} \left\| W^t(e_i - e_j) \right\|_1$$

$$\leq \sum_{t=0}^{T_0-1} 2 + \sum_{t=0}^{\infty} 2\sqrt{n}\rho^{t+T_0}$$

$$\leq 2 \left\lceil \frac{-\log(\sqrt{n})}{\log(\rho)} \right\rceil + \sum_{t=0}^{\infty} 2 \left( \sqrt{n}\rho^{T_0} \right) \rho^t$$

$$\leq \frac{2\log(\sqrt{n})}{1-\rho} + 2 + \frac{2}{1-\rho}$$

$$\leq \frac{\log(16n)}{1-\rho}$$

*As a result, we have*

$$\|X_{k+1}(e_i - e_j)\|_\infty \leq (\alpha G_\infty + 2\delta\theta) \frac{\log(16n)}{1-\rho}$$

*Since $\delta = \frac{1-\rho}{4\log(16n)}$, we have*

$$\|X_{k+1}(e_i - e_j)\|_\infty \leq (\alpha G_\infty + 2\delta\theta) \frac{\log(16n)}{1-\rho} \leq \frac{2\log(16n)}{1-\rho} \alpha G_\infty = \theta$$

*Combining I and II, we complete the proof.*

**Lemma 5** *The output of Algorithm 1 has the following bound:*

$$\frac{1}{K} \sum_{k=0}^{K-1} \mathbb{E} \left\| \nabla f(\overline{X}_k) \right\|^2 \leq \frac{4(f(0) - f^*)}{\alpha K} + \frac{2\alpha L}{n} \sigma^2 + \frac{8\alpha^2 L^2 \left( \sigma^2 + 3\varsigma^2 \right)}{(1-\rho)^2}$$

$$+ \frac{8L^2}{nK(1-\rho)^2} \sum_{k=1}^{K-1} \mathbb{E} \left\| \Omega_k \right\|_F^2$$

**Proof** *From Lemma 8, we have*

$$\frac{1-\alpha L}{K}\sum_{k=0}^{K-1}\mathbb{E}\left\|\overline{G}_k\right\|^2 + \frac{1}{K}\sum_{k=0}^{K-1}\mathbb{E}\left\|\nabla f(\overline{X}_k)\right\|^2$$

$$\leq \frac{2(f(0)-f^*)}{\alpha K} + \frac{\alpha L}{n}\sigma^2 + \frac{L^2}{nK}\sum_{k=0}^{K-1}\sum_{i=1}^{n}\mathbb{E}\left\|\overline{X}_k - x_{k,i}\right\|^2$$

$$\overset{Lemma\ 6}{\leq} \frac{2(f(0)-f^*)}{\alpha K} + \frac{\alpha L}{n}\sigma^2 + \frac{2\alpha^2 L^2}{M_1(1-\rho)^2}\left(\sigma^2 + 3\varsigma^2 + \frac{3}{K}\sum_{k=0}^{K-1}\mathbb{E}\left\|\nabla f(\overline{X}_k)\right\|^2\right)$$

$$+ \frac{2L^2}{M_1 nK(1-\rho)^2}\sum_{k=1}^{K-1}\mathbb{E}\left\|\Omega_k\right\|_F^2$$

*where*

$$M_1 = 1 - \frac{6\alpha^2 L^2}{(1-\rho)^2}$$

*Rearrange the terms, we get*

$$\frac{1-\alpha L}{K}\sum_{k=0}^{K-1}\mathbb{E}\left\|\overline{G}_k\right\|^2 + \left(1 - \frac{6\alpha^2 L^2}{M_1(1-\rho)^2}\right)\frac{1}{K}\sum_{k=0}^{K-1}\mathbb{E}\left\|\nabla f(\overline{X}_k)\right\|^2$$

$$\leq \frac{2(f(0)-f^*)}{\alpha K} + \frac{\alpha L}{n}\sigma^2 + \frac{2\alpha^2 L^2\left(\sigma^2 + 3\varsigma^2\right)}{M_1(1-\rho)^2} + \frac{2L^2}{M_1 nK(1-\rho)^2}\sum_{k=1}^{K-1}\mathbb{E}\left\|\Omega_k\right\|_F^2$$

*Let*

$$M_2 = 1 - \frac{6\alpha^2 L^2}{M_1(1-\rho)^2}$$

*we get*

$$\frac{1-\alpha L}{K}\sum_{k=0}^{K-1}\mathbb{E}\left\|\overline{G}_k\right\|^2 + \frac{M_2}{K}\sum_{k=0}^{K-1}\mathbb{E}\left\|\nabla f(\overline{X}_k)\right\|^2$$

$$\leq \frac{2(f(0)-f^*)}{\alpha K} + \frac{\alpha L}{n}\sigma^2 + \frac{2\alpha^2 L^2\left(\sigma^2 + 3\varsigma^2\right)}{M_1(1-\rho)^2} + \frac{2L^2}{M_1 nK(1-\rho)^2}\sum_{k=1}^{K-1}\mathbb{E}\left\|\Omega_k\right\|_F^2$$

*Let $M_1, M_2 \geq \frac{1}{2}$ and rearrange the terms, we have*

$$\frac{1}{K}\sum_{k=0}^{K-1}\mathbb{E}\left\|\nabla f(\overline{X}_k)\right\|^2 \leq \frac{4(f(0)-f^*)}{\alpha K} + \frac{2\alpha L}{n}\sigma^2 + \frac{8\alpha^2 L^2\left(\sigma^2 + 3\varsigma^2\right)}{(1-\rho)^2}$$

$$+ \frac{8L^2}{nK(1-\rho)^2}\sum_{k=1}^{K-1}\mathbb{E}\left\|\Omega_k\right\|_F^2$$

*and that completes the proof*

**Lemma 6** *Let $M_1 = 1 - \frac{6\alpha^2 L^2}{(1-\rho)^2} > 0$, we have*

$$\frac{L^2}{nK}\sum_{k=0}^{K-1}\sum_{i=1}^{n}\mathbb{E}\left\|\overline{X}_k - x_{k,i}\right\|^2 \leq \frac{2\alpha^2 L^2}{M_1(1-\rho)^2}\left(\sigma^2 + 3\varsigma^2 + \frac{3}{K}\sum_{k=0}^{K-1}\mathbb{E}\left\|\nabla f(\overline{X}_k)\right\|^2\right)$$

$$+ \frac{2L^2}{M_1 nK(1-\rho)^2}\sum_{k=1}^{K-1}\mathbb{E}\left\|\Omega_k\right\|_F^2$$

**Proof**

$$\sum_{k=0}^{K-1}\sum_{i=1}^{n}\mathbb{E}\left\|\overline{X}_k - x_{k,i}\right\|^2$$

$$= \sum_{k=1}^{K-1} \sum_{i=1}^{n} \mathbb{E} \left\| X_k \left( \frac{\mathbb{1}_n}{n} - e_i \right) \right\|^2$$

$$= \sum_{k=1}^{K-1} \sum_{i=1}^{n} \mathbb{E} \left\| \left( X_{k-1}W - \alpha \widetilde{G}_{k-1} + \Omega_{k-1} \right) \left( \frac{\mathbb{1}_n}{n} - e_i \right) \right\|^2$$

$$\overset{x_{0,i}=0}{=} \sum_{k=1}^{K-1} \sum_{i=1}^{n} \mathbb{E} \left\| \sum_{t=0}^{k-1} \left( -\alpha \widetilde{G}_t + \Omega_t \right) \left( \frac{\mathbb{1}_n}{n} - W^{k-t-1} e_i \right) \right\|^2$$

$$\leq 2\alpha^2 \sum_{k=1}^{K-1} \sum_{i=1}^{n} \mathbb{E} \left\| \sum_{t=0}^{k-1} \widetilde{G}_t \left( \frac{\mathbb{1}_n}{n} - W^{k-t-1} e_i \right) \right\|^2$$

$$+ 2 \sum_{k=1}^{K-1} \sum_{i=1}^{n} \mathbb{E} \left\| \sum_{t=0}^{k-1} \Omega_t \left( \frac{\mathbb{1}_n}{n} - W^{k-t-1} e_i \right) \right\|^2$$

$$= 2\alpha^2 \sum_{k=1}^{K-1} \mathbb{E} \left\| \sum_{t=0}^{k-1} \widetilde{G}_t \left( \frac{\mathbb{1}_n \mathbb{1}_n^\top}{n} - W^{k-t-1} \right) \right\|_F^2 + 2 \sum_{k=1}^{K-1} \mathbb{E} \left\| \sum_{t=0}^{k-1} \Omega_t \left( \frac{\mathbb{1}_n \mathbb{1}_n^\top}{n} - W^{k-t-1} \right) \right\|_F^2$$

$$\overset{Lemma\ 10}{\leq} 2\alpha^2 \sum_{k=1}^{K-1} \mathbb{E} \left( \sum_{t=0}^{k-1} \rho^{k-t-1} \left\| \widetilde{G}_t \right\|_F \right)^2 + 2 \sum_{k=1}^{K-1} \mathbb{E} \left( \sum_{t=0}^{k-1} \rho^{k-t-1} \left\| \Omega_t \right\|_F \right)^2$$

$$\overset{Lemma\ 9}{\leq} \frac{2\alpha^2}{(1-\rho)^2} \sum_{k=1}^{K-1} \mathbb{E} \left\| \widetilde{G}_k \right\|_F^2 + \frac{2}{(1-\rho)^2} \sum_{k=1}^{K-1} \mathbb{E} \left\| \Omega_k \right\|_F^2$$

$$\overset{Lemma\ 7}{\leq} \frac{2\alpha^2}{(1-\rho)^2} \left( n\sigma^2 K + 3L^2 \sum_{k=0}^{K-1} \sum_{i=1}^{n} \mathbb{E} \left\| \overline{X}_k - x_{k,i} \right\|^2 + 3n\varsigma^2 K + 3n \sum_{k=0}^{K-1} \mathbb{E} \left\| \nabla f(\overline{X}_k) \right\|^2 \right)$$

$$+ \frac{2}{(1-\rho)^2} \sum_{k=1}^{K-1} \mathbb{E} \left\| \Omega_k \right\|_F^2$$

*Rearrange the terms, we have*

$$\left( 1 - \frac{6\alpha^2 L^2}{(1-\rho)^2} \right) \sum_{k=0}^{K-1} \sum_{i=1}^{n} \mathbb{E} \left\| \overline{X}_k - x_{k,i} \right\|^2$$

$$\leq \frac{2\alpha^2}{(1-\rho)^2} \left( n\sigma^2 K + 3n\varsigma^2 K + 3n \sum_{k=0}^{K-1} \mathbb{E} \left\| \nabla f(\overline{X}_k) \right\|^2 \right) + \frac{2}{(1-\rho)^2} \sum_{k=1}^{K-1} \mathbb{E} \left\| \Omega_k \right\|_F^2$$

*Let $M_1 = 1 - \frac{6\alpha^2 L^2}{(1-\rho)^2} > 0$, we have*

$$\frac{L^2}{nK} \sum_{k=0}^{K-1} \sum_{i=1}^{n} \mathbb{E} \left\| \overline{X}_k - x_{k,i} \right\|^2 \leq \frac{2\alpha^2 L^2}{M_1 (1-\rho)^2} \left( \sigma^2 + 3\varsigma^2 + \frac{3}{K} \sum_{k=0}^{K-1} \mathbb{E} \left\| \nabla f(\overline{X}_k) \right\|^2 \right)$$

$$+ \frac{2L^2}{M_1 nK (1-\rho)^2} \sum_{k=1}^{K-1} \mathbb{E} \left\| \Omega_k \right\|_F^2$$

**Lemma 7**

$$\sum_{k=0}^{K-1} \mathbb{E} \left\| \widetilde{G}_k \right\|_F^2 \leq n\sigma^2 K + 3L^2 \sum_{k=0}^{K-1} \sum_{i=1}^{n} \mathbb{E} \left\| \overline{X}_k - x_{k,i} \right\|^2 + 3n\varsigma^2 K + 3n \sum_{k=0}^{K-1} \mathbb{E} \left\| \nabla f(\overline{X}_k) \right\|^2$$

**Proof** *From the property of Frobenius norm, we have*

$$\mathbb{E} \left\| \widetilde{G}_k \right\|_F^2 = \sum_{i=1}^{n} \mathbb{E} \left\| \widetilde{g}_{k,i} \right\|^2$$

*Next, we derive the upper bound of $\mathbb{E} \left\| \widetilde{g}_{k,i} \right\|^2$*

$$\mathbb{E} \left\| \widetilde{g}_{k,i} \right\|^2$$
$$=\mathbb{E} \left\| \widetilde{g}_{k,i} - g_{k,i} + g_{k,i} \right\|^2$$
$$=\mathbb{E} \left\| \widetilde{g}_{k,i} - g_{k,i} \right\|^2 + \mathbb{E} \left\| g_{k,i} \right\|^2 + 2\mathbb{E} \left\langle \widetilde{g}_{k,i} - g_{k,i}, g_{k,i} \right\rangle$$
$$=\mathbb{E} \left\| \widetilde{g}_{k,i} - g_{k,i} \right\|^2 + \mathbb{E} \left\| g_{k,i} \right\|^2$$
$$\leq \sigma^2 + 3\mathbb{E} \left\| g_{k,i} - \nabla f_i(\overline{X}_k) \right\|^2 + 3\mathbb{E} \left\| \nabla f_i(\overline{X}_k) - \nabla f(\overline{X}_k) \right\|^2 + 3\mathbb{E} \left\| \nabla f(\overline{X}_k) \right\|^2$$
$$\leq \sigma^2 + 3L^2 \mathbb{E} \left\| \overline{X}_k - x_{k,i} \right\|^2 + 3\varsigma^2 + 3\mathbb{E} \left\| \nabla f(\overline{X}_k) \right\|^2$$

*Summing from $k = 0$ to $K - 1$, we obtain*

$$\sum_{k=0}^{K-1} \mathbb{E} \left\| \widetilde{G}_k \right\|_F^2$$
$$= \sum_{k=0}^{K-1} \sum_{i=1}^{n} \mathbb{E} \left\| \widetilde{g}_{k,i} \right\|^2$$
$$\leq \sum_{k=0}^{K-1} \sum_{i=1}^{n} \sigma^2 + 3L^2 \sum_{k=0}^{K-1} \sum_{i=1}^{n} \mathbb{E} \left\| \overline{X}_k - x_{k,i} \right\|^2 + 3 \sum_{k=0}^{K-1} \sum_{i=1}^{n} \varsigma^2 + 3 \sum_{k=0}^{K-1} \sum_{i=1}^{n} \mathbb{E} \left\| \nabla f(\overline{X}_k) \right\|^2$$
$$= n\sigma^2 K + 3L^2 \sum_{k=0}^{K-1} \sum_{i=1}^{n} \mathbb{E} \left\| \overline{X}_k - x_{k,i} \right\|^2 + 3n\varsigma^2 K + 3n \sum_{k=0}^{K-1} \mathbb{E} \left\| \nabla f(\overline{X}_k) \right\|^2$$

*That completes the proof*

**Lemma 8**

$$\frac{1 - \alpha L}{K} \sum_{k=0}^{K-1} \mathbb{E} \left\| \overline{G}_k \right\|^2 + \frac{1}{K} \sum_{k=0}^{K-1} \mathbb{E} \left\| \nabla f(\overline{X}_k) \right\|^2$$
$$\leq \frac{2(f(0) - f^*)}{\alpha K} + \frac{\alpha L}{n} \sigma^2 + \frac{L^2}{nK} \sum_{k=0}^{K-1} \sum_{i=1}^{n} \mathbb{E} \left\| \overline{X}_k - x_{k,i} \right\|^2$$

**Proof** *Let $\mathbb{1}_n$ denote a n-dimensional vector with all the entries be 1. And we have*

$$\overline{X}_{k+1} = (X_k W - \alpha \widetilde{G}_k + \Omega_k) \frac{\mathbb{1}_n}{n} = \overline{X}_k - \alpha \overline{\widetilde{G}}_k + (Q_k - X_k)(W - I) \frac{\mathbb{1}_n}{n} = \overline{X}_k - \alpha \overline{\widetilde{G}}_k$$

*And by Taylor Expansion, we have*

$$\mathbb{E} f(\overline{X}_{k+1}) = \mathbb{E} f \left( \frac{(X_k W - \alpha \widetilde{G}_k + \Omega_k) \mathbb{1}_n}{n} \right)$$
$$= \mathbb{E} f \left( \overline{X}_k - \alpha \overline{\widetilde{G}}_k \right)$$
$$\leq \mathbb{E} f(\overline{X}_k) - \alpha \mathbb{E} \langle \nabla f(\overline{X}_k), \overline{\widetilde{G}}_k \rangle + \frac{\alpha^2 L}{2} \mathbb{E} \left\| \overline{\widetilde{G}}_k \right\|^2$$

*And for the last term, we have*

$$\mathbb{E} \left\| \overline{\widetilde{G}}_k \right\|^2 = \mathbb{E} \left\| \frac{\sum_{i=1}^{n} \widetilde{g}_{k,i}}{n} \right\|^2$$
$$= \mathbb{E} \left\| \frac{\sum_{i=1}^{n} \widetilde{g}_{k,i} - \sum_{i=1}^{n} g_{k,i}}{n} + \frac{\sum_{i=1}^{n} g_{k,i}}{n} \right\|^2$$
$$= \mathbb{E} \left\| \frac{\sum_{i=1}^{n} \widetilde{g}_{k,i} - \sum_{i=1}^{n} g_{k,i}}{n} \right\|^2 + \mathbb{E} \left\| \frac{\sum_{i=1}^{n} g_{k,i}}{n} \right\|^2$$

$$+\mathbb{E}\left\langle \frac{\sum_{i=1}^{n} \widetilde{g}_{k,i} - \sum_{i=1}^{n} g_{k,i}}{n} + \frac{\sum_{i=1}^{n} g_{k,i}}{n} \right\rangle$$

$$= \mathbb{E}\left\| \frac{\sum_{i=1}^{n} \widetilde{g}_{k,i} - \sum_{i=1}^{n} g_{k,i}}{n} \right\|^2 + \mathbb{E}\left\| \frac{\sum_{i=1}^{n} g_{k,i}}{n} \right\|^2$$

$$\overset{Assumption\ (A3)}{=} \frac{1}{n^2} \sum_{i=1}^{n} \mathbb{E}\left\| \widetilde{g}_{k,i} - g_{k,i} \right\|^2 + \mathbb{E}\left\| \frac{\sum_{i=1}^{n} g_{k,i}}{n} \right\|^2$$

$$\overset{Assumption\ (A3)}{\leq} \frac{\sigma^2}{n} + \mathbb{E}\left\| \frac{\sum_{i=1}^{n} g_{k,i}}{n} \right\|^2$$

*Putting it back, we obtain*

$$\mathbb{E}f(\overline{X}_{k+1}) \leq \mathbb{E}f(\overline{X}_k) - \alpha \mathbb{E}\langle \nabla f(\overline{X}_k), \overline{\widetilde{G}}_k \rangle + \frac{\alpha^2 L}{2n}\sigma^2 + \frac{\alpha^2 L}{2}\mathbb{E}\left\| \frac{\sum_{i=1}^{n} g_{k,i}}{n} \right\|^2$$

$$= \mathbb{E}f(\overline{X}_k) - \frac{\alpha - \alpha^2 L}{2}\mathbb{E}\left\| \overline{G}_k \right\|^2 - \frac{\alpha}{2}\mathbb{E}\left\| \nabla f(\overline{X}_k) \right\|^2 + \frac{\alpha^2 L}{2n}\sigma^2$$

$$+ \frac{\alpha}{2}\mathbb{E}\left\| \nabla f(\overline{X}_k) - \overline{G}_k \right\|^2$$

*where the last step comes from $2\langle a, b\rangle = \|a\|^2 + \|b\|^2 = \|a - b\|^2$ And*

$$\mathbb{E}\left\| \nabla f(\overline{X}_k) - \overline{G}_k \right\|^2 \leq \frac{1}{n}\sum_{i=1}^{n} \mathbb{E}\left\| \nabla f_i\left( \frac{\sum_{i'=1}^{n} x_{k,i'}}{n} \right) - \nabla f_i(x_{k,i}) \right\|^2$$

$$\overset{Assumption\ (A1)}{\leq} \frac{L^2}{n}\sum_{i=1}^{n} \mathbb{E}\left\| \frac{\sum_{i'=1}^{n} x_{k,i'}}{n} - x_{k,i} \right\|^2$$

$$= \frac{L^2}{n}\sum_{i=1}^{n} \mathbb{E}\left\| \overline{X}_k - x_{k,i} \right\|^2$$

*putting it back, we have*

$$\frac{\alpha - \alpha^2 L}{2}\mathbb{E}\left\| \overline{G}_k \right\|^2 + \frac{\alpha}{2}\mathbb{E}\left\| \nabla f(\overline{X}_k) \right\|^2 \leq \mathbb{E}f(\overline{X}_k) - \mathbb{E}f(\overline{X}_{k+1}) + \frac{\alpha^2 L}{2n}\sigma^2 + \frac{\alpha L^2}{2n}\sum_{i=1}^{n} \mathbb{E}\left\| \overline{X}_k - x_{k,i} \right\|^2$$

*summing over from $k = 0$ to $K - 1$ on both sides, we have*

$$\frac{1 - \alpha L}{K}\sum_{k=0}^{K-1} \mathbb{E}\left\| \overline{G}_k \right\|^2 + \frac{1}{K}\sum_{k=0}^{K-1} \mathbb{E}\left\| \nabla f(\overline{X}_k) \right\|^2$$

$$\leq \frac{2(f(0) - f^*)}{\alpha K} + \frac{\alpha L}{n}\sigma^2 + \frac{L^2}{nK}\sum_{k=0}^{K-1}\sum_{i=1}^{n} \mathbb{E}\left\| \overline{X}_k - x_{k,i} \right\|^2$$

*That completes the proof.*

**Lemma 9** *Given two non-negative sequences $\{a_t\}_{t=1}^{\infty}$ and $\{b_t\}_{t=1}^{\infty}$ that satisfying*

$$a_t = \sum_{s=1}^{t} \rho^{t-s} b_s$$

*with $0 \leq \rho < 1$, we have*

$$S_k = \sum_{t=1}^{k} a_t \leq \frac{1}{1 - \rho}\sum_{s=1}^{k} b_s$$

$$D_k = \sum_{t=1}^{k} a_t^2 \leq \frac{1}{(1 - \rho)^2}\sum_{s=1}^{k} b_s^2$$

**Proof**

$$S_k = \sum_{t=1}^{k} a_t = \sum_{t=1}^{k} \sum_{s=1}^{t} \rho^{t-s} b_s = \sum_{s=1}^{k} \sum_{t=s}^{k} \rho^{t-s} b_s = \sum_{s=1}^{k} \sum_{t=0}^{k-s} \rho^{t} b_s \le \frac{1}{1-\rho} \sum_{s=1}^{k} b_s$$

$$D_k = \sum_{t=1}^{k} a_t = \sum_{t=1}^{k} \sum_{s=1}^{t} \rho^{t-s} b_s \sum_{r=1}^{t} \rho^{t-r} b_r = \sum_{s=1}^{k} \sum_{t=s}^{k} \rho^{t-s} b_s = \sum_{t=1}^{k} \sum_{s=1}^{t} \sum_{r=1}^{t} \rho^{2t-s-r} b_s b_r$$

$$\le \sum_{t=1}^{k} \sum_{s=1}^{t} \sum_{r=1}^{t} \rho^{2t-s-r} \frac{b_s^2 + b_r^2}{2} = \sum_{t=1}^{k} \sum_{s=1}^{t} \sum_{r=1}^{t} \rho^{2t-s-r} b_s^2$$

$$\le \frac{1}{1-\rho} \sum_{t=1}^{k} \sum_{s=1}^{t} \rho^{t-s} b_s^2 \le \frac{1}{(1-\rho)^2} \sum_{s=1}^{k} b_s^2$$

**Lemma 10** *For any $X_t \in \mathbb{R}^{d \times n}$, we have*

$$\left\| \sum_{t=0}^{k-1} X_t \left( \frac{\mathbb{1}_n \mathbb{1}_n^\top}{n} - W^{k-t-1} \right) \right\|_F^2 \le \left( \sum_{t=0}^{k-1} \rho^{k-t-1} \|X_t\|_F \right)^2$$

**Proof**

$$\left\| \sum_{t=0}^{k-1} X_t \left( \frac{\mathbb{1}_n \mathbb{1}_n^\top}{n} - W^{k-t-1} \right) \right\|_F^2 = \left( \left\| \sum_{t=0}^{k-1} X_t \left( \frac{\mathbb{1}_n \mathbb{1}_n^\top}{n} - W^{k-t-1} \right) \right\|_F \right)^2$$

$$\le \left( \sum_{t=0}^{k-1} \left\| X_t \left( \frac{\mathbb{1}_n \mathbb{1}_n^\top}{n} - W^{k-t-1} \right) \right\|_F \right)^2$$

$$\le \left( \sum_{t=0}^{k-1} \|X_t\|_F \left\| \frac{\mathbb{1}_n \mathbb{1}_n^\top}{n} - W^{k-t-1} \right\| \right)^2$$

$$\le \left( \sum_{t=0}^{k-1} \rho^{k-t-1} \|X_t\|_F \right)^2$$

*That completes the proof.*

## G  MONIQUA ON $D^2$ (PROOF TO THEOREM 3)

### G.1  ALGORITHM

---

**Algorithm 2** Moniqua with Variance Reduction on worker $i$

---

**Input:** initial point $x_{0,i} = x_0$, step size $\alpha$, the discrepency bound $\theta$, communication matrix $W$, number of iterations $K$, neighbor list of worker $i$: $\mathcal{N}_i$

1: **for** $k = 0, 1, 2, \cdots, K-1$ **do**
2:     Randomly sample data $\xi_{k,i}$ from local memory
3:     Compute a local stochastic gradient based on $\xi_{k,i}$ and current weight $x_{k,i}$: $\widetilde{g}_{k,i}$
4:     **if** $k = 0$ **then**
5:         Update local weight: $x_{k+\frac{1}{2},i} \leftarrow x_{k,i} - \alpha \widetilde{g}_{k,i}$
6:     **else**
7:         Update local weight: $x_{k+\frac{1}{2},i} \leftarrow 2x_{k,i} - x_{k-1,i} - \alpha \widetilde{g}_{k,i} + \alpha \widetilde{g}_{k-1,i}$
8:     **end if**
9:     Compute modulo-ed model: $q_{k,i} \leftarrow \theta \cdot \mathcal{Q}_\delta \left( \frac{x_{k+\frac{1}{2},i}}{\theta} \bmod 1 \right)$ (element-wise division and mod)
10:     Average with neighboring workers:: $x_{k+1,i} \leftarrow x_{k+\frac{1}{2},i} + \sum_{j \in \mathcal{N}_i} (q_{k,j} - q_{k,i}) W_{ji}$
11: **end for**
**Output:** $\overline{X}_K = \frac{1}{n} \sum_{i=1}^{n} x_{K,i}$

---

## G.2 Assumptions

$D^2$ makes the following assumptions (1-4), and we add the additional assumption (5):

1. **Lipschitzian Gradient**: All the function $f_i$ have L-Lipschitzian gradients.

2. **Communication Matrix**: Communication matrix $W$ is a symmetric doubly stochastic matrix. Let the eigenvalues of $W \in \mathbb{R}^{n \times n}$ be $\lambda_1 \geq \cdots \geq \lambda_n$. We assume $\lambda_2 < 1, \lambda_n > -\frac{1}{3}$.

3. **Bounded Variance**:

$$\mathbb{E}_{\xi_i \sim \mathcal{D}_i} \left\| \nabla \widetilde{f}_i(x_i; \xi_i) - \nabla f_i(x) \right\|^2 \leq \sigma^2, \forall i$$

where $\nabla \widetilde{f}_i(x; \xi_i)$ denotes gradient sample on worker $i$ computed via data sample $\xi_i$.

4. **Initialization**: All the models are initialized by the same parameters: $x_{0,i} = x_0, \forall i$ and with out the loss of generality $x_0 = 0$.

5. **Gradient magnitude**: The norm of a sampled gradient is bounded by $\|\widetilde{g}_{k,i}\|_\infty \leq G_\infty$ for some constant $G_\infty$.

## G.3 Proof to Theorem 3

**Proof** *From a local view, define $x_{-1} = \widetilde{g}_{-1} = 0$, the update rule of Moniqua on $D^2$ on worker $i$ in iteration $k$ can be written as*

$$x_{k+\frac{1}{2},i} = 2x_{k,i} - x_{k-1,i} - \alpha \widetilde{g}_{k,i} + \alpha \widetilde{g}_{k-1,i}$$

$$x_{k+1,i} = \sum_{j=1}^{n} x_{k+\frac{1}{2},j} W_{ji} + \sum_{j=1}^{n} \left( (q_{k,j} - x_{k+\frac{1}{2},j}) - (q_{k,i} - x_{k+\frac{1}{2},i}) \right) W_{ji}$$

*From a global view, the update rule can be written as*

$$X_{k+\frac{1}{2}} = 2X_k - X_{k-1} - \alpha \widetilde{G}_k + \alpha \widetilde{G}_{k-1}$$
$$X_{k+1} = X_{k+\frac{1}{2}} W + (Q_k - X_{k+\frac{1}{2}})(W - I)$$

*Define*

$$\Omega_k = (Q_k - X_{k+\frac{1}{2}})(W - I)$$

*Since $W$ is symmetric, it can be diagonalized as $W = P\Lambda P^\top$, where the i-th column of $P$ and $\Lambda$ are $W$'s i-th eigenvector and eigenvalue, respectively. And we obtain*

$$X_{k+1} = 2X_k P\Lambda P^\top - X_{k-1} P\Lambda P^\top - \alpha \widetilde{G}_k P\Lambda P^\top + \alpha \widetilde{G}_{k-1} P\Lambda P^\top + \Omega_k$$

*and*

$$X_{k+1}P = 2X_k P\Lambda - X_{k-1}P\Lambda - \alpha \widetilde{G}_k P\Lambda + \alpha \widetilde{G}_{k-1}P\Lambda + \Omega_k P$$

*Denote $Y_k = X_k P$, $H(X_k; \xi_k) = \widetilde{G}_k P$, and denote $y_{k,i}$, $h_{k,i}$ and $r_{k,i}$ as the i-th column of $Y_k$, $H_k$ and $\Omega_k P$, respectively. Then we have*

$$y_{k+1,i} = \lambda_i (2y_{k,i} - y_{k-1,i} - \alpha h_{k,i} + \alpha h_{k-1,i}) + r_{k,i}$$

*From Lemma 15 (Constants $C_1$, $C_2$, $C_3$ andn $C_4$ are defined in the Lemma 11. Constants $D_1$ and $D_2$ are defined in Lemma 15) we get*

$$\left( 1 - \frac{3C_1 \alpha^2 L^2}{C_4} \right) \mathbb{E} \left\| \nabla f(0) \right\| + \left( 1 - \alpha L - 3\frac{C_2}{C_4} \alpha^4 L^4 \right) \frac{1}{K} \sum_{k=1}^{K-1} \mathbb{E} \left\| \overline{G}_k \right\|^2 + \frac{1}{K} \sum_{k=0}^{K-1} \mathbb{E} \left\| \nabla f(\overline{X}_k) \right\|^2$$

$$\leq \frac{2(f(0) - f^*)}{\alpha K} + \frac{\alpha L}{n} \sigma^2 + \frac{3C_1 \alpha^2 L^2 (\sigma^2 + \varsigma_0^2)}{C_4 K} + 6\frac{C_2}{C_4} \alpha^2 \sigma^2 L^2 + 3\frac{C_2}{nC_4} \alpha^4 \sigma^2 L^4$$

$$+ \frac{C_3 L^2}{C_4} \left( \frac{3D_1 n + 4}{3D_2 n} \right)^2 \alpha^2 G_\infty^2 d$$

*Let $\alpha = \frac{1}{\sigma \sqrt{K/n + 2L}}$, we have*

$$\frac{1}{K} \sum_{k=0}^{K-1} \mathbb{E} \left\| \nabla f(\overline{X}_k) \right\|^2$$

$$\leq \frac{2(f(0) - f^*)}{\alpha K} + \frac{\alpha L}{n} \sigma^2 + \frac{3C_1 \alpha^2 L^2 (\sigma^2 + \varsigma_0^2)}{C_4 K} + 6\frac{C_2}{C_4} \alpha^2 \sigma^2 L^2 + 3\frac{C_2}{nC_4} \alpha^4 \sigma^2 L^4$$

$$+ \left( \frac{3D_1 n + 4}{3D_2 n} \right)^2 \frac{C_3 L^2}{C_4} G_\infty^2 d\alpha^2$$

$$\leq \frac{4(f(0) - f^*)L}{K} + \frac{2\sigma(f(0) - f^* + L/2)}{\sqrt{nK}} + \frac{3C_1 L^2 (\sigma^2 + \varsigma_0^2)n}{C_4 (\sigma^2 K^2 + 4nL^2 K)} + \frac{6C_2 L^2 \sigma^2 n}{C_4 (\sigma^2 K + 4nL^2)}$$

$$+ \frac{3C_2 n \sigma^2 L^2}{C_4 (\sigma^4 K^2 + 16n^2 L^4)} + \left( \frac{3D_1 n + 4}{3D_2 n} \right)^2 \frac{C_3 G_\infty^2 dL^2 n}{C_4 (\sigma^2 K + 4nL^2)}$$

$$\lesssim \frac{1}{K} + \frac{\sigma}{\sqrt{nK}} + \frac{(\sigma^2 + \varsigma_0^2)n}{\sigma^2 K^2 + nK} + \frac{\sigma^2 n}{\sigma^2 K + n} + \frac{\sigma^2 n}{\sigma^4 K^2 + n^2} + \frac{G_\infty^2 dn}{\sigma^2 K + n}$$

$$\lesssim \frac{1}{K} + \frac{\sigma}{\sqrt{nK}} + \frac{\sigma^2 n}{\sigma^2 K + n} + \frac{G_\infty^2 dn}{\sigma^2 K + n}$$

*That completes the proof.*

### G.4  LEMMA FOR $D^2$

**Lemma 11** *Define*

$$D_1 = \max \left\{ |v_n| + \frac{2|\lambda_n|}{1 - |v_n|}, \sqrt{\frac{\lambda_2}{1 - \lambda_2}} + \frac{2\lambda_2}{1 - \lambda_2} \right\}$$

$$D_2 = \max \left\{ \frac{2}{1 - |v_n|}, \frac{2}{\sqrt{1 - \lambda_2}} \right\}$$

$$v_n = \lambda_n - \sqrt{\lambda_n^2 - \lambda_n}$$

*Let $\delta = \frac{1}{6nD_2}$, and we have for $\forall i, j$*

$$\left\| x_{k+\frac{1}{2}} (e_i - e_j) \right\|_\infty \leq \theta = (6D_1 n + 8) \alpha G_\infty$$

**Proof** *We use mathematical induction to prove this:*

*I. When $k = 0$,*

$$\left\| X_{0+\frac{1}{2}} (e_i - e_j) \right\|_\infty = \left\| -\alpha \widetilde{G}_0 (e_i - e_j) \right\|_\infty \leq \alpha \left\| \widetilde{G}_0 \right\|_{1,\infty} \|e_i - e_j\|_1 \leq 2\alpha G_\infty \leq (6D_1 n + 8) \alpha G_\infty$$

*II. Suppose for $k \geq 0$, $\forall t \leq k$, we have $\left\| X_{t+\frac{1}{2}} (e_i - e_j) \right\| \leq (6D_1 n + 8) \alpha G_\infty$, then for $\forall i, j$*

$$\|X_{k+1} (e_i - e_j)\|_\infty$$

$$\leq \left\| X_{k+1} \left( \frac{\mathbb{1}_n}{n} - e_i \right) \right\|_\infty + \left\| X_{k+1} \left( \frac{\mathbb{1}_n}{n} - e_j \right) \right\|_\infty$$

$$= \left\| X_{k+1} PP^\top e_i - X_{k+1} P \begin{bmatrix} 1 & 0 & 0 & \dots & 0 \\ 0 & 0 & 0 & \dots & 0 \\ 0 & 0 & 0 & \dots & 0 \\ \vdots & \vdots & \vdots & \ddots & \vdots \\ 0 & 0 & 0 & \dots & 0 \end{bmatrix} P^\top e_i \right\|_\infty$$

$$+ \left\| X_{k+1}PP^\top e_j - X_{k+1}P \begin{bmatrix} 1 & 0 & 0 & \dots & 0 \\ 0 & 0 & 0 & \dots & 0 \\ 0 & 0 & 0 & \dots & 0 \\ \vdots & \vdots & \vdots & \ddots & \vdots \\ 0 & 0 & 0 & \dots & 0 \end{bmatrix} P^\top e_j \right\|_\infty$$

$$\leq \left\| X_{k+1}P \begin{bmatrix} 0 & 0 & 0 & \dots & 0 \\ 0 & 1 & 0 & \dots & 0 \\ 0 & 0 & 1 & \dots & 0 \\ \vdots & \vdots & \vdots & \ddots & \vdots \\ 0 & 0 & 0 & \dots & 1 \end{bmatrix} \right\|_{1,\infty} \left\| P^\top e_i \right\|_1 + \left\| X_{k+1}P \begin{bmatrix} 0 & 0 & 0 & \dots & 0 \\ 0 & 1 & 0 & \dots & 0 \\ 0 & 0 & 1 & \dots & 0 \\ \vdots & \vdots & \vdots & \ddots & \vdots \\ 0 & 0 & 0 & \dots & 1 \end{bmatrix} \right\|_{1,\infty} \left\| P^\top e_j \right\|_1$$

$$\leq 2\sqrt{n} \left\| X_{k+1}P \begin{bmatrix} 0 & 0 & 0 & \dots & 0 \\ 0 & 1 & 0 & \dots & 0 \\ 0 & 0 & 1 & \dots & 0 \\ \vdots & \vdots & \vdots & \ddots & \vdots \\ 0 & 0 & 0 & \dots & 1 \end{bmatrix} \right\|_{1,\infty}$$

*From the update rule, we have*

$$y_{k+1,i} = \lambda_i(2y_{k,i} - y_{k-1,i} - \alpha h_{k,i} + \alpha h_{k-1,i}) + r_{k,i} = \lambda_i(2y_{k,i} - y_{k-1,i}) + \lambda_i\beta_{k,i} + r_{k,i}$$

*where* $\beta_{k,i} = -\alpha h_{k,i} + \alpha h_{k-1,i}$, *for all* $y_i$ *with* $-\frac{1}{3} < \lambda_i < 0$, *from Lemma 13 we have*

$$y_{k+1,i} = y_{1,i}\left(\frac{u_i^{k+1} - v_i^{k+1}}{u_i - v_i}\right) + \sum_{s=1}^{k}(\lambda_i\beta_{s,i} + r_{s,i})\frac{u_i^{k-s+1} - v_i^{k-s+1}}{u_i - v_i}$$

*where* $u_i = \lambda_i + \sqrt{\lambda_i^2 - \lambda_i}$ *and* $v_i = \lambda_i - \sqrt{\lambda_i^2 - \lambda_i}$, *we obtain*

$$\|y_{k+1,i}\|_\infty \leq \|y_{1,i}\|_\infty \left|\frac{u_i^{k+1} - v_i^{k+1}}{u_i - v_i}\right| + |\lambda_i| \sum_{s=1}^{k} \|\beta_{s,i}\|_\infty \left|\frac{u_i^{k-s+1} - v_i^{k-s+1}}{u_i - v_i}\right|$$

$$+ \sum_{s=1}^{k} \|r_{s,i}\|_\infty \left|\frac{u_i^{k-s+1} - v_i^{k-s+1}}{u_i - v_i}\right|$$

*Since*

$$\left|\frac{u_i^{n+1} - v_i^{n+1}}{u_i - v_i}\right| \leq |v_i|^n \left|\frac{u_i\left(\frac{u_i}{v_i}\right)^n - v_i}{u_i - v_i}\right| \leq |v_i|^n$$

*We obtain*

$$\|y_{k+1,i}\|_\infty \leq \|y_{1,i}\|_\infty |v_i|^k + |\lambda_i| \sum_{s=1}^{k} \|\beta_{s,i}\|_\infty |v_i|^{k-s} + \sum_{s=1}^{k} \|r_{s,i}\|_\infty |v_i|^{k-s}$$

*For* $\beta_{s,i}$, *we have*

$$\|\beta_{s,i}\|_\infty = \|-\alpha h_{k,i} + \alpha h_{k-1,i}\|_\infty \leq 2\alpha(\|h_{k,i}\|_\infty + \|h_{k-1,i}\|_\infty)$$
$$\leq 2\alpha(\|G_k\|_{1,\infty}\|Pe_i\|_1 + \|G_{k-1}\|_{1,\infty}\|Pe_i\|_1)$$
$$\leq 2\alpha\sqrt{n}G_\infty$$

*For* $r_{s,i}$, *we have*

$$\|r_{k,i}\|_\infty = \|\Omega_k Pe_i\|_\infty \leq \|\Omega_k\|_{1,\infty}\|Pe_i\|_1 \leq 2\sqrt{n}\delta\theta$$

*when* $\lambda_i < 0$, *we have*

$$\|y_{k+1,i}\|_\infty \leq \|y_{1,i}\|_\infty |v_i|^k + |\lambda_i| \sum_{s=1}^{k} \|\beta_{s,i}\|_\infty |v_i|^{k-s} + \sum_{s=1}^{k} \|r_{s,i}\|_\infty |v_i|^{k-s}$$

$$\leq \|y_{1,i}\|_\infty |v_n|^k + |\lambda_n| \sum_{s=1}^k \|\beta_{s,i}\|_\infty |v_n|^{k-s} + \sum_{s=1}^k \|r_{s,i}\|_\infty |v_n|^{k-s}$$

$$\leq \alpha\sqrt{n}G_\infty |v_n|^k + 2\alpha\sqrt{n}G_\infty |\lambda_n| \sum_{s=1}^\infty |v_n|^{k-s} + 2\sqrt{n}\delta\theta \sum_{s=1}^\infty |v_n|^{k-s}$$

$$\leq \alpha\sqrt{n}G_\infty |v_n| + \frac{2\alpha\sqrt{n}G_\infty |\lambda_n|}{1-|v_n|} + \frac{2\sqrt{n}\delta\theta}{1-|v_n|}$$

*where $v_n = \lambda_n - \sqrt{\lambda_n^2 - \lambda_n}$.*

*On the other hand, when $0 \leq \lambda_i < 1$, from Lemma 13 we have*

$$y_{k+1,i}\sin\theta_i = y_{1,i}\lambda_i^{\frac{k}{2}} \sin[(t+1)\theta_i] + \lambda_i \sum_{s=1}^k \beta_{s,i}\lambda_i^{\frac{k-s}{2}} \sin[(k+1-s)\theta_i]$$

$$+ \sum_{s=1}^k r_{s,i}\lambda_i^{\frac{k-s}{2}} \sin[(k+1-s)\theta_i]$$

*By taking norm, we get*

$$\|y_{k+1,i}\|_\infty |\sin\theta_i| = \|y_{1,i}\|_\infty \lambda_i^{\frac{k}{2}} |\sin[(t+1)\theta_i]| + \lambda_i \sum_{s=1}^k \|\beta_{s,i}\|_\infty |\lambda_i^{\frac{k-s}{2}}| |\sin[(k+1-s)\theta_i]|$$

$$+ \sum_{s=1}^k \|r_{s,i}\|_\infty |\lambda_i^{\frac{k-s}{2}}| |\sin[(k+1-s)\theta_i]|$$

$$\leq \|y_{1,i}\|_\infty \lambda_2^{\frac{k}{2}} + 2\alpha\sqrt{n}G_\infty\lambda_2 \sum_{s=1}^\infty \lambda_2^{\frac{s}{2}} + 2\sqrt{n}\delta\theta \sum_{s=1}^\infty \lambda_2^{\frac{s}{2}}$$

$$\leq \alpha\sqrt{n}G_\infty\sqrt{\lambda_2} + \frac{2\alpha\sqrt{n}G_\infty\lambda_2 + 2\sqrt{n}\delta\theta}{\sqrt{1-\lambda_2}}$$

*Since $|\sin\theta_i| \geq \sqrt{1-\lambda_2}$, putting it back, we get*

$$\|y_{k+1,i}\| \leq \alpha\sqrt{n}G_\infty\sqrt{\frac{\lambda_2}{1-\lambda_2}} + \frac{2\alpha\sqrt{n}G_\infty\lambda_2 + 2\sqrt{n}\delta\theta}{1-\lambda_2}$$

*So there exists $D_1, D_2$*

$$D_1 = \max\left\{ |v_n| + \frac{2|\lambda_n|}{1-|v_n|}, \sqrt{\frac{\lambda_2}{1-\lambda_2}} + \frac{2\lambda_2}{1-\lambda_2} \right\}$$

$$D_2 = \max\left\{ \frac{2}{1-|v_n|}, \frac{2}{\sqrt{1-\lambda_2}} \right\}$$

*such that*

$$\|y_{k+1,i}\|_\infty \leq D_1\alpha\sqrt{n}G_\infty + D_2\sqrt{n}\delta\theta$$

*Putting it back we have $\forall i, j$*

$$\|X_{k+1}(e_i - e_j)\|_\infty \leq D_1\alpha n G_\infty + D_2 n\delta\theta$$

*As a result*

$$\left\|X_{k+\frac{1}{2}}(e_i - e_j)\right\|_\infty$$
$$= \left\|(2X_k - X_{k-1} - \alpha\widetilde{G}_k + \alpha\widetilde{G}_{k-1})(e_i - e_j)\right\|_\infty$$
$$\leq 2\|X_k(e_i - e_j)\|_\infty + \|X_{k-1}(e_i - e_j)\|_\infty + \alpha\left\|\widetilde{G}_k\right\|_{1,\infty} \|e_i - e_j\|_1 + \alpha\left\|\widetilde{G}_{k-1}\right\|_{1,\infty} \|e_i - e_j\|_1$$

$$\leq 3(D_1 \alpha n G_\infty + D_2 n \delta \theta) + 4\alpha G_\infty$$
$$\leq (6D_1 n + 8)\alpha G_\infty$$

*The last step is because $\delta = \frac{1}{6nD_2}$*

*Combining I and II we complete the proof.*

**Lemma 12**

$$(1 - 12C_2\alpha^2 L^2) \sum_{i=1}^{n} \sum_{k=1}^{K} \mathbb{E} \left\| \overline{X}_k - x_{k,i} \right\|^2$$
$$\leq 3C_1\alpha^2 n\sigma^2 + 3C_1\alpha^2 n\varsigma_0^2 + 3C_1\alpha^2 n\mathbb{E} \left\| \nabla f(0) \right\| + 6C_2\alpha^2 n\sigma^2 K + 3C_2\alpha^4 \sigma^2 L^2 K$$
$$+ 3C_2\alpha^4 nL^2 \sum_{k=1}^{K-1} \mathbb{E} \left\| \overline{G}_k \right\|^2 + C_3 \sum_{k=1}^{K-1} \mathbb{E} \left\| \Omega_k \right\|_F^2$$

**Proof**

$$\sum_{i=1}^{n} \left\| \overline{X}_k - x_{k,i} \right\|^2 = \sum_{i=1}^{n} \left\| X_k \left( e_i - \frac{\mathbb{1}_n}{n} \right) \right\|^2$$
$$= \left\| X_k \left( I - \frac{\mathbb{1}_n \mathbb{1}_n^\top}{n} \right) \right\|_F^2$$
$$= \left\| X_k P P^\top - X_k v_1 v_1^\top \right\|_F^2$$
$$\overset{\text{Lemma 14}}{=} \left\| X_k P \begin{bmatrix} 0 & 0 & 0 & \dots & 0 \\ 0 & 1 & 0 & \dots & 0 \\ 0 & 0 & 1 & \dots & 0 \\ \vdots & \vdots & \vdots & \ddots & \vdots \\ 0 & 0 & 0 & \dots & 1 \end{bmatrix} \right\|_F^2$$
$$= \sum_{i=2}^{n} \left\| y_{k,i} \right\|^2$$

*From the update rule, we obtain,*

$$y_{k+1,i} = \lambda_i(2y_{k,i} - y_{k-1,i} - \alpha h_{k,i} + \alpha h_{k-1,i}) + r_{k,i} = \lambda_i(2y_{k,i} - y_{k-1,i}) + \lambda_i \beta_{k,i} + r_{k,i}$$

*where$\beta_{k,i} = -\alpha h_{k,i} + \alpha h_{k-1,i}$, for all $y_i$ with $-\frac{1}{3} < \lambda_i < 0$, from Lemma 13 we have*

$$y_{k+1,i} = y_{1,i} \left( \frac{u_i^{k+1} - v_i^{k+1}}{u_i - v_i} \right) + \sum_{s=1}^{k} (\lambda_i \beta_{s,i} + r_{k,i}) \frac{u_i^{k-s+1} - v_i^{k-s+1}}{u_i - v_i}$$

*where $u_i = \lambda_i + \sqrt{\lambda_i^2 - \lambda_i}$ and $v_i = \lambda_i - \sqrt{\lambda_i^2 - \lambda_i}$, we obtain*

$$\left\| y_{k+1,i} \right\|^2 \leq 3 \left\| y_{1,i} \right\|^2 \left( \frac{u_i^{k+1} - v_i^{k+1}}{u_i - v_i} \right)^2 + 3\lambda_i^2 \left( \sum_{s=1}^{k} \left\| \beta_{s,i} \right\| \left| \frac{u_i^{k-s+1} - v_i^{k-s+1}}{u_i - v_i} \right| \right)^2$$
$$+ 3 \left( \sum_{s=1}^{k} \left\| r_{s,i} \right\| \left| \frac{u_i^{k-s+1} - v_i^{k-s+1}}{u_i - v_i} \right| \right)^2$$

*Since*

$$\left| \frac{u_i^{n+1} - v_i^{n+1}}{u_i - v_i} \right| \leq |v_i|^n \left| \frac{u_i \left( \frac{u_i}{v_i} \right)^n - v_i}{u_i - v_i} \right| \leq |v_i|^n$$

*We obtain*

$$\left\| y_{k+1,i} \right\|^2 \leq 3 \left\| y_{1,i} \right\|^2 |v_i|^{2t} + 3\lambda_i^2 \left( \sum_{s=1}^{k} \left\| \beta_{s,i} \right\| |v_i|^{k-s} \right)^2 + 3 \left( \sum_{s=1}^{k} \left\| r_{s,i} \right\| |v_i|^{k-s} \right)^2$$

*Summing over from $k = 0$ to $t = K - 1$, we obtain*

$$\sum_{k=0}^{K-1} \|y_{k+1,i}\|^2 = \sum_{k=1}^{K} \|y_{k,i}\|^2$$

$$\leq 3 \|y_{1,i}\|^2 \sum_{k=0}^{K-1} |v_i|^{2k} + 3\lambda_i^2 \sum_{k=1}^{K-1} \left( \sum_{s=1}^{k} \|\beta_{s,i}\| \, |v_i|^{k-s} \right)^2 + 3 \sum_{k=1}^{K-1} \left( \sum_{s=1}^{k} \|r_{s,i}\| \, |v_i|^{k-s} \right)^2$$

$$\leq \frac{3 \|y_{1,i}\|^2}{1 - |v_i|^2} + \frac{3\lambda_i^2}{(1 - |v_i|)^2} \sum_{k=1}^{K-1} \|\beta_{k,i}\|^2 + \frac{3}{(1 - |v_i|)^2} \sum_{k=1}^{K-1} \|r_{k,i}\|^2$$

$$\leq \frac{3 \|y_{1,i}\|^2}{1 - |v_n|^2} + \frac{3\lambda_n^2}{(1 - |v_n|)^2} \sum_{k=1}^{K-1} \|\beta_{k,i}\|^2 + \frac{3}{(1 - |v_n|)^2} \sum_{k=1}^{K-1} \|r_{k,i}\|^2$$

*where $v_n = \lambda_n - \sqrt{\lambda_n^2 - \lambda_n}$.*

*On the other hand, when $0 \leq \lambda_i < 1$, from Lemma 13 we have*

$$y_{k+1,i} \sin \theta_i = y_{1,i} \lambda_i^{\frac{k}{2}} \sin[(t+1)\theta_i] + \lambda_i \sum_{s=1}^{k} \beta_{s,i} \lambda_i^{\frac{k-s}{2}} \sin[(k+1-s)\theta_i]$$

$$+ \sum_{s=1}^{k} r_{s,i} \lambda_i^{\frac{k-s}{2}} \sin[(k+1-s)\theta_i]$$

*And we have*

$$\|y_{k+1,i}\|^2 \sin^2 \theta_i \leq 3 \|y_{1,i}\|^2 \lambda_i^k \sin^2[(t+1)\theta_i] + 3\lambda_i^2 \left( \sum_{s=1}^{k} \beta_{s,i} \lambda_i^{\frac{k-s}{2}} \sin[(k+1-s)\theta_i] \right)^2$$

$$+ 3 \left( \sum_{s=1}^{k} r_{s,i} \lambda_i^{\frac{k-s}{2}} \sin[(k+1-s)\theta_i] \right)^2$$

$$\leq 3 \|y_{1,i}\|^2 \lambda_i^k + 3\lambda_i^2 \left( \sum_{s=1}^{k} \beta_{s,i} \lambda_i^{\frac{k-s}{2}} \right)^2 + 3 \left( \sum_{s=1}^{k} r_{s,i} \lambda_i^{\frac{k-s}{2}} \right)^2$$

*Summing from $k = 0$ to $K - 1$, we have*

$$\sum_{k=0}^{K-1} \|y_{k+1,i}\|^2 \sin^2 \theta_i = \sum_{k=1}^{K} \|y_{k,i}\|^2 \sin^2 \theta_i$$

$$\leq 3 \|y_{1,i}\|^2 \sum_{k=0}^{K-1} \lambda_i^t + 3\lambda_i^2 \sum_{k=1}^{K-1} \left( \sum_{s=1}^{k} \|\beta_{s,i}\| \lambda_i^{\frac{t-s}{2}} \right)^2 + 3 \sum_{k=1}^{K-1} \left( \sum_{s=1}^{k} r_{s,i} \lambda_i^{\frac{k-s}{2}} \right)^2$$

$$\leq \frac{3 \|y_{1,i}\|^2}{1 - \lambda_i} + \frac{3\lambda_i^2}{(1 - \sqrt{\lambda_i})^2} \sum_{k=1}^{K-1} \|\beta_{k,i}\|^2 + \frac{3}{(1 - \sqrt{\lambda_i})^2} \sum_{k=1}^{K-1} \|r_{k,i}\|^2$$

*Since $\sin^2 \theta_i = 1 - \lambda_i$, we have*

$$\sum_{k=1}^{K} \|y_{k,i}\|^2 \leq \frac{3 \|y_{1,i}\|^2}{(1 - \lambda_i)^2} + \frac{3\lambda_i^2}{(1 - \sqrt{\lambda_i})^2 (1 - \lambda_i)} \sum_{k=1}^{K-1} \|\beta_{k,i}\|^2 + \frac{3}{(1 - \sqrt{\lambda_i})^2 (1 - \lambda_i)} \sum_{k=1}^{K-1} \|r_{k,i}\|^2$$

$$\leq \frac{3 \|y_{1,i}\|^2}{(1 - \lambda_2)^2} + \frac{3\lambda_2^2}{(1 - \sqrt{\lambda_2})^2 (1 - \lambda_2)} \sum_{k=1}^{K-1} \|\beta_{k,i}\|^2 + \frac{3}{(1 - \sqrt{\lambda_2})^2 (1 - \lambda_2)} \sum_{k=1}^{K-1} \|r_{k,i}\|^2$$

*So there exists $C_1, C_2, C_3$*

$$C_1 = \max\left\{\frac{3}{1-|v_n|^2}, \frac{3}{(1-\lambda_2)^2}\right\}$$

$$C_2 = \max\left\{\frac{3\lambda_n^2}{(1-|v_n|)^2}, \frac{3\lambda_2^2}{(1-\sqrt{\lambda_2})^2(1-\lambda_2)}\right\}$$

$$C_3 = \max\left\{\frac{3}{(1-|v_n|)^2}, \frac{3}{(1-\sqrt{\lambda_2})^2(1-\lambda_2)}\right\}$$

$$\sum_{k=1}^{K}\|y_{k,i}\|^2 \leq C_1\|y_{1,i}\|^2 + C_2\sum_{k=1}^{K-1}\|\beta_{k,i}\|^2 + C_3\sum_{k=1}^{K-1}\|r_{k,i}\|^2$$

*By taking expectation we have*

$$\sum_{k=1}^{K}\mathbb{E}\|y_{k,i}\|^2 \leq C_1\mathbb{E}\|y_{1,i}\|^2 + C_2\sum_{k=1}^{K-1}\mathbb{E}\|\beta_{k,i}\|^2 + C_3\sum_{k=1}^{K-1}\mathbb{E}\|r_{k,i}\|^2$$

*We next analyze $\beta_{k,i}$:*

$$\sum_{i=2}^{n}\mathbb{E}\|\beta_{k,i}\|^2$$

$$=\alpha^2\sum_{i=2}^{n}\mathbb{E}\|h_{k,i} - h_{k-1,i}\|^2$$

$$=\alpha^2\sum_{i=2}^{n}\mathbb{E}\left\|\widetilde{G}_k Pe_i - \widetilde{G}_{k-1}Pe_i\right\|^2$$

$$\leq\alpha^2\sum_{i=1}^{n}\mathbb{E}\left\|\widetilde{G}_k Pe_i - \widetilde{G}_{k-1}Pe_i\right\|^2$$

$$\leq\alpha^2\mathbb{E}\left\|\widetilde{G}_k P - \widetilde{G}_{k-1}P\right\|_F^2$$

$$\overset{Lemma\ 14}{\leq}\alpha^2\mathbb{E}\left\|\widetilde{G}_k - \widetilde{G}_{k-1}\right\|_F^2$$

$$=\alpha^2\sum_{i=1}^{n}\mathbb{E}\left\|\widetilde{G}_k e_i - \widetilde{G}_{k-1}e_i\right\|^2$$

$$\leq3\alpha^2\sum_{i=1}^{n}\mathbb{E}\left\|\widetilde{G}_k e_i - G_k e_i\right\|^2 + 3\alpha^2\sum_{i=1}^{n}\mathbb{E}\left\|\widetilde{G}_{k-1}e_i - G_{k-1}e_i\right\|^2$$

$$+3\alpha^2\sum_{i=1}^{n}\mathbb{E}\|G_k e_i - G_{k-1}e_i\|^2$$

$$\leq6\alpha^2n\sigma^2 + 3\alpha^2\sum_{i=1}^{n}\mathbb{E}\|G_k e_i - G_{k-1}e_i\|^2$$

$$\leq6\alpha^2n\sigma^2 + 3\alpha^2 L^2\sum_{i=1}^{n}\mathbb{E}\|x_{k,i} - x_{k-1,i}\|^2$$

$$\leq6\alpha^2n\sigma^2 + 3\alpha^2 L^2\sum_{i=1}^{n}\mathbb{E}\left\|Y_k P^\top e_i - Y_{k-1}P^\top e_i\right\|^2$$

$$\leq6\alpha^2n\sigma^2 + 3\alpha^2 L^2\mathbb{E}\left\|Y_k P^\top - Y_{k-1}P^\top\right\|_F^2$$

$$\overset{Lemma\ 14}{\leq}6\alpha^2n\sigma^2 + 3\alpha^2 L^2\mathbb{E}\|Y_k - Y_{k-1}\|_F^2$$

$$\leq 6\alpha^2 n\sigma^2 + 3\alpha^2 L^2 \sum_{i=1}^{n} \mathbb{E} \left\| y_{k,i} - y_{k-1,i} \right\|^2$$

*And Putting it back, we have*

$$\sum_{i=2}^{n} \sum_{k=1}^{K} \mathbb{E} \left\| y_{k,i} \right\|^2$$

$$\leq C_1 \mathbb{E} \left\| Y_1 \right\|_F^2 + C_2 \sum_{i=2}^{n} \sum_{k=1}^{K-1} \mathbb{E} \left\| \beta_{k,i} \right\|^2 + C_3 \sum_{k=1}^{K-1} \sum_{i=2}^{n} \mathbb{E} \left\| r_{k,i} \right\|^2$$

$$\leq C_1 \mathbb{E} \left\| Y_1 \right\|_F^2 + C_2 \sum_{k=1}^{K-1} \left( 6\alpha^2 n\sigma^2 + 3\alpha^2 L^2 \sum_{i=1}^{n} \mathbb{E} \left\| y_{k,i} - y_{k-1,i} \right\|^2 \right) + C_3 \sum_{k=1}^{K-1} \sum_{i=2}^{n} \mathbb{E} \left\| r_{k,i} \right\|^2$$

$$\overset{Lemma\ 14}{\leq} C_1 \mathbb{E} \left\| Y_1 \right\|_F^2 + 6C_2\alpha^2 n\sigma^2 K + 3C_2\alpha^2 L^2 \sum_{k=1}^{K-1} \sum_{i=1}^{n} \mathbb{E} \left\| y_{k,i} - y_{k-1,i} \right\|^2 + C_3 \sum_{k=1}^{K-1} \mathbb{E} \left\| \Omega_k \right\|_F^2$$

*Since*

$$\mathbb{E} \left\| y_{k,1} - y_{k-1,1} \right\|^2 = \mathbb{E} \left\| X_k P e_1 - X_{k-1} P e_1 \right\|^2 = \mathbb{E} \left\| X_k v_1 - X_{k-1} v_1 \right\|^2$$

$$= \mathbb{E} \left\| X_k \frac{1}{\sqrt{n}} \mathbb{1}_n - X_{k-1} \frac{1}{\sqrt{n}} \mathbb{1}_n \right\|^2 = n\mathbb{E} \left\| \overline{X}_k - \overline{X}_{k-1} \right\|^2 = n\alpha^2 \mathbb{E} \left\| \widetilde{\overline{G}}_k \right\|^2$$

$$\leq n\alpha^2 \mathbb{E} \left\| \widetilde{\overline{G}}_k - \overline{G}_k \right\|^2 + n\alpha^2 \mathbb{E} \left\| \overline{G}_k \right\|^2 \leq n\alpha^2 \frac{\sigma^2}{n} + n\alpha^2 \mathbb{E} \left\| \overline{G}_k \right\|^2$$

$$= \alpha^2 \sigma^2 + n\alpha^2 \mathbb{E} \left\| \overline{G}_k \right\|^2$$

*Putting it back, and we obtain*

$$\sum_{i=2}^{n} \sum_{k=1}^{K} \mathbb{E} \left\| y_{k,i} \right\|^2$$

$$\leq C_1 \mathbb{E} \left\| Y_1 \right\|_F^2 + 6C_2\alpha^2 n\sigma^2 K + 3C_2\alpha^4 \sigma^2 L^2 K + 3C_2\alpha^4 nL^2 \sum_{k=1}^{K-1} \mathbb{E} \left\| \overline{G}_k \right\|^2$$

$$+ 3C_2\alpha^2 L^2 \sum_{k=1}^{K-1} \sum_{i=2}^{n} \mathbb{E} \left\| y_{k,i} - y_{k-1,i} \right\|^2 + C_3 \sum_{k=1}^{K-1} \mathbb{E} \left\| \Omega_k \right\|_F^2$$

$$\leq C_1 \mathbb{E} \left\| Y_1 \right\|_F^2 + 6C_2\alpha^2 n\sigma^2 K + 3C_2\alpha^4 \sigma^2 L^2 K + 3C_2\alpha^4 nL^2 \sum_{k=1}^{K-1} \mathbb{E} \left\| \overline{G}_k \right\|^2$$

$$+ 6C_2\alpha^2 L^2 \sum_{k=1}^{K-1} \sum_{i=2}^{n} \mathbb{E} \left( \left\| y_{k,i} \right\|^2 + \left\| y_{k-1,i} \right\|^2 \right) + C_3 \sum_{k=1}^{K-1} \mathbb{E} \left\| \Omega_k \right\|_F^2$$

$$\leq C_1 \mathbb{E} \left\| Y_1 \right\|_F^2 + 6C_2\alpha^2 n\sigma^2 K + 3C_2\alpha^4 \sigma^2 L^2 K + 3C_2\alpha^4 nL^2 \sum_{k=1}^{K-1} \mathbb{E} \left\| \overline{G}_k \right\|^2$$

$$+ 12C_2\alpha^2 L^2 \sum_{k=1}^{K-1} \sum_{i=2}^{n} \mathbb{E} \left\| y_{k,i} \right\|^2 + C_3 \sum_{k=1}^{K-1} \mathbb{E} \left\| \Omega_k \right\|_F^2$$

*Rearrange the terms, we get*

$$(1 - 12C_2\alpha^2 L^2) \sum_{i=2}^{n} \sum_{k=1}^{K} \mathbb{E} \left\| y_{k,i} \right\|^2$$

$$\leq C_1 \mathbb{E} \left\| Y_1 \right\|_F^2 + 6C_2 \alpha^2 n \sigma^2 K + 3C_2 \alpha^4 \sigma^2 L^2 K + 3C_2 \alpha^4 n L^2 \sum_{k=1}^{K-1} \mathbb{E} \left\| \overline{G}_k \right\|^2 + C_3 \sum_{k=1}^{K-1} \mathbb{E} \left\| \Omega_k \right\|_F^2$$

$$\leq C_1 \mathbb{E} \left\| X_1 \right\|_F^2 + 6C_2 \alpha^2 n \sigma^2 K + 3C_2 \alpha^4 \sigma^2 L^2 K + 3C_2 \alpha^4 n L^2 \sum_{k=1}^{K-1} \mathbb{E} \left\| \overline{G}_k \right\|^2 + C_3 \sum_{k=1}^{K-1} \mathbb{E} \left\| \Omega_k \right\|_F^2$$

*Considering*

$$\begin{aligned}
\mathbb{E} \left\| X_1 \right\|_F^2 &= \alpha^2 \mathbb{E} \left\| \widetilde{G}_0 \right\|_F^2 \\
&= \alpha^2 \sum_{i=1}^{n} \mathbb{E} \left\| \widetilde{G}_{0,i} - G_{0,i} + G_{0,i} - \nabla f(0) + \nabla f(0) \right\|^2 \\
&\leq 3\alpha^2 \sum_{i=1}^{n} \mathbb{E} \left\| \widetilde{G}_{0,i} - G_{0,i} \right\|^2 + 3\alpha^2 \sum_{i=1}^{n} \mathbb{E} \left\| G_{0,i} - \nabla f(0) \right\|^2 + 3\alpha^2 \sum_{i=1}^{n} \mathbb{E} \left\| \nabla f(0) \right\|^2 \\
&\leq 3\alpha^2 n \sigma^2 + 3\alpha^2 n \varsigma_0^2 + 3\alpha^2 n \mathbb{E} \left\| \nabla f(0) \right\|
\end{aligned}$$

*We finally get*

$$\begin{aligned}
&(1 - 12C_2 \alpha^2 L^2) \sum_{i=2}^{n} \sum_{k=1}^{K} \mathbb{E} \left\| y_{k,i} \right\|^2 \\
=&(1 - 12C_2 \alpha^2 L^2) \sum_{i=1}^{n} \sum_{k=1}^{K} \mathbb{E} \left\| \overline{X}_k - x_{k,i} \right\|^2 \\
\leq& 3C_1 \alpha^2 n \sigma^2 + 3C_1 \alpha^2 n \varsigma_0^2 + 3C_1 \alpha^2 n \mathbb{E} \left\| \nabla f(0) \right\| + 6C_2 \alpha^2 n \sigma^2 K + 3C_2 \alpha^4 \sigma^2 L^2 K \\
&+ 3C_2 \alpha^4 n L^2 \sum_{k=1}^{K-1} \mathbb{E} \left\| \overline{G}_k \right\|^2 + C_3 \sum_{k=1}^{K-1} \mathbb{E} \left\| \Omega_k \right\|_F^2
\end{aligned}$$

*That completes the proof.*

**Lemma 13** *Given $\rho \in \left( -\frac{1}{3}, 0 \right) \cup (0, 1)$, for any two sequence $\{a_t\}_{t=1}^{\infty}$, $\{b_t\}_{t=1}^{\infty}$ and $\{c_t\}_{t=1}^{\infty}$ that satisfying*

$$\begin{aligned}
a_0 &= b_0 = 0, \\
a_{t+1} &= \rho \left( 2a_t - a_{t-1} \right) + b_t - b_{t-1} + c_t, \forall t \geq 1
\end{aligned}$$

*we have*

$$a_{t+1} = a_1 \left( \frac{u^{t+1} - v^{t+1}}{u - v} \right) + \sum_{s=1}^{t} (b_s - b_{s-1} + c_s) \left( \frac{u^{t-s+1} - v^{t-s+1}}{u - v} \right), \forall t \geq 0$$

*where*

$$u = \rho + \sqrt{\rho^2 - \rho}, v = \rho - \sqrt{\rho^2 - \rho}$$

*Moreover, if $0 < \rho < 1$, we have*

$$a_{t+1} = a_1 \rho^{\frac{t}{2}} \frac{\sin[(t+1)\theta]}{\sin \theta} + \sum_{s=1}^{t} (b_s - b_{s-1} + c_s) \rho^{\frac{t-s}{2}} \frac{\sin[(t-s+1)\theta]}{\sin \theta}$$

*where*

$$\theta = \arccos \left( \sqrt{\rho} \right)$$

**Proof** *when $t \geq 1$, we have*

$$a_{t+1} = 2\rho a_t - \rho a_{t-1} + b_t - b_{t-1} + c_t$$

*since,*

$$u = \rho + \sqrt{\rho^2 - \rho}, v = \rho - \sqrt{\rho^2 - \rho}$$

*we obtain*

$$a_{t+1} - ua_t = (a_t - ua_{t-1})v + b_t - b_{t-1} + c_t$$

*Recursively we have*

$$\begin{aligned}
a_{t+1} - ua_t &= (a_t - ua_{t-1})v + b_t - b_{t-1} + c_t \\
&= (a_{t-1} - ua_{t-2})v^2 + (b_{t-1} - b_{t-2} + c_{t-1})v + b_t - b_{t-1} + c_t \\
&= (a_1 - ua_0)v^t + \sum_{s=1}^{t}(b_s - b_{s-1} + c_s)v^{t-s} \\
&= a_1 v^t + \sum_{s=1}^{t}(b_s - b_{s-1} + c_s)v^{t-s}
\end{aligned}$$

*Dividing both sides by $u^{t+1}$, we have*

$$\begin{aligned}
\frac{a_{t+1}}{u^{t+1}} &= \frac{a_t}{u^t} + u^{-(t+1)}\left(a_1 v^t + \sum_{s=1}^{t}(b_s - b_{s-1} + c_s)v^{t-s}\right) \\
&= \frac{a_{t-1}}{u^{t-1}} + u^{-t}\left(a_1 v^{t-1} + \sum_{s=1}^{t-1}(b_s - b_{s-1} + c_s)v^{t-1-s}\right) \\
&\quad + u^{-(t+1)}\left(a_1 v^t + \sum_{s=1}^{t}(b_s - b_{s-1} + c_s)v^{t-s}\right) \\
&= \frac{a_1}{u} + \sum_{k=1}^{t}u^{-k-1}\left(a_1 v^k + \sum_{s=1}^{k}(b_s - b_{s-1} + c_s)v^{k-s}\right)
\end{aligned}$$

*Multiplying both sides by $u^{t+1}$*

$$\begin{aligned}
a_{t+1} &= a_1 u^t + \sum_{k=1}^{t}u^{t-k}\left(a_1 v^k + \sum_{s=1}^{k}(b_s - b_{s-1} + c_s)v^{t-s}\right) \\
&= a_1 u^t\left(1 + \sum_{k=1}^{t}\left(\frac{v}{u}\right)^k\right) + u^t\sum_{k=1}^{t}\sum_{s=1}^{k}(b_s - b_{s-1} + c_s)v^{-s}\left(\frac{v}{u}\right)^k \\
&= a_1 u^t\sum_{k=0}^{t}\left(\frac{v}{u}\right)^k + u^t\sum_{s=1}^{t}\sum_{k=s}^{t}(b_s - b_{s-1} + c_s)v^{-s}\left(\frac{v}{u}\right)^k \\
&= a_1 u^t\left(\frac{1 - \left(\frac{v}{u}\right)^{t+1}}{1 - \frac{v}{u}}\right) + u^t\sum_{s=1}^{t}(b_s - b_{s-1} + c_s)v^{-s}\left(\frac{v}{u}\right)^s\frac{1 - \left(\frac{v}{u}\right)^{t-s-1}}{1 - \frac{v}{u}} \\
&= a_1\left(\frac{u^{t+1} - v^{t+1}}{u - v}\right) + \sum_{s=1}^{t}(b_s - b_{s-1} + c_s)\frac{u^{t-s+1} - v^{t-s+1}}{u - v}
\end{aligned}$$

*Note that when $0 < \rho < 1$, both $u$ and $v$ are complex numbers, we have*

$$u = \sqrt{\rho}e^{i\theta}, v = \sqrt{\rho}e^{-i\theta}$$

*where $\theta = \arccos\sqrt{\rho}$. And under this context, we have*

$$a_{t+1} = a_1 \rho^{\frac{t}{2}}\frac{\sin[(t+1)\theta]}{\sin\theta} + \sum_{s=1}^{t}(b_s - b_{s-1} + c_s)\rho^{\frac{t-s}{2}}\frac{\sin[(t-s+1)\theta]}{\sin\theta}$$

*That completes the proof.*

**Lemma 14** *For any matrix $X \in \mathbb{R}^{N\times n}$, we have*

$$\sum_{i=2}^{n}\|Xv_i\|^2 \leq \sum_{i=1}^{n}\|Xv_i\|^2 = \|X\|_F^2$$

$$\sum_{i=1}^{n}\|XP^\top e_i\|^2 = \|XP^\top\|_F^2 = \|X\|_F^2$$

**Proof**

$$\sum_{i=2}^{n} \|X_t v_i\|^2 \leq \sum_{i=1}^{n} \|X_t v_i\|^2 = \|X_t P\|_F^2 = Tr(X_t PP^\top X_t^\top) = Tr(X_t X_t^\top) = \|X_t\|_F^2$$

*And similarly,*

$$\sum_{i=1}^{n} \left\|X P^\top e_i\right\|^2 = \left\|X P^\top\right\|_F^2 = Tr(X_t P^\top P X_t^\top) = Tr(X_t X_t^\top) = \|X_t\|_F^2$$

*That completes the proof.*

**Lemma 15** *If we run Algorithm 2 for $K$ iterations the following inequality holds:*

$$\left(1 - \frac{3C_1 \alpha^2 L^2}{C_4}\right) \mathbb{E}\|\nabla f(0)\| + \left(1 - \alpha L - 3\frac{C_2}{C_4} \alpha^4 L^4\right) \frac{1}{K} \sum_{k=1}^{K-1} \mathbb{E}\left\|\overline{G}_k\right\|^2$$

$$+ \frac{1}{K} \sum_{k=0}^{K-1} \mathbb{E}\left\|\nabla f(\overline{X}_k)\right\|^2$$

$$\leq \frac{2(f(0) - f^*)}{\alpha K} + \frac{\alpha L}{n} \sigma^2 + \frac{3C_1 \alpha^2 L^2(\sigma^2 + \varsigma_0^2)}{C_4 K} + 6\frac{C_2}{C_4} \alpha^2 \sigma^2 L^2 + 3\frac{C_2}{nC_4} \alpha^4 \sigma^2 L^4$$

$$+ \frac{C_3 L^2}{C_4} \left(\frac{3D_1 n + 4}{3D_2 n}\right)^2 \alpha^2 G_\infty^2 d$$

*where*

$$C_1 = \max\left\{\frac{3}{1 - |v_n|^2}, \frac{3}{(1 - \lambda_2)^2}\right\}$$

$$C_2 = \max\left\{\frac{3\lambda_n^2}{(1 - |v_n|)^2}, \frac{3\lambda_2^2}{(1 - \sqrt{\lambda_2})^2(1 - \lambda_2)}\right\}$$

$$C_3 = \max\left\{\frac{3}{(1 - |v_n|)^2}, \frac{3}{(1 - \sqrt{\lambda_2})^2(1 - \lambda_2)}\right\}$$

$$C_4 = 1 - 12C_2 \alpha^2 L^2$$

$$\Omega_k e_i = \sum_{j=1}^{n} \left((q_{k,j} - x_{k+\frac{1}{2},j}) - (q_{k,i} - x_{k+\frac{1}{2},i})\right) W_{ji}$$

**Proof** *Since*

$$\overline{X}_{k+1} = (2X_k - X_{k-1} - \alpha\widetilde{G}_k + \alpha\widetilde{G}_{k-1})W\frac{\mathbb{1}_n}{n} + (Q_k - X_{k+\frac{1}{2}})(W - I)\frac{\mathbb{1}_n}{n}$$

$$= 2\overline{X}_k - \overline{X}_{k-1} - \alpha\overline{\widetilde{G}}_k + \alpha\overline{\widetilde{G}}_{k-1}$$

*and we have*

$$\overline{X}_{k+1} - \overline{X}_k = \overline{X}_k - \overline{X}_{k-1} - \alpha\overline{\widetilde{G}}_k + \alpha\overline{\widetilde{G}}_{k-1}$$

$$= \overline{X}_1 - \overline{X}_0 - \alpha\sum_{t=1}^{k}(\overline{\widetilde{G}}_t - \overline{\widetilde{G}}_{t-1})$$

$$= -\alpha\overline{\widetilde{G}}_k$$

*As a result, we can reuse Lemma 8 from D-PSGD, thus we have*

$$\frac{1 - \alpha L}{K} \sum_{k=0}^{K-1} \mathbb{E}\left\|\overline{G}_k\right\|^2 + \frac{1}{K} \sum_{k=0}^{K-1} \mathbb{E}\left\|\nabla f(\overline{X}_k)\right\|^2$$

$$\leq \frac{2(f(0) - f^*)}{\alpha K} + \frac{\alpha L}{n} \sigma^2 + \frac{L^2}{nK} \sum_{k=0}^{K-1} \sum_{i=1}^{n} \mathbb{E}\left\|\overline{X}_k - x_{k,i}\right\|^2$$

*From Lemma 12 we obatin*

$$
\frac{1-\alpha L}{K}\sum_{k=0}^{K-1}\mathbb{E}\left\|\overline{G}_k\right\|^2 + \frac{1}{K}\sum_{k=0}^{K-1}\mathbb{E}\left\|\nabla f(\overline{X}_k)\right\|^2
$$

$$
\leq \frac{2(f(0)-f^*)}{\alpha K} + \frac{\alpha L}{n}\sigma^2 + \frac{3C_1\alpha^2 L^2(\sigma^2 + \varsigma_0^2 + \mathbb{E}\left\|\nabla f(0)\right\|)}{C_4 K} + 6\frac{C_2}{C_4}\alpha^2\sigma^2 L^2 + 3\frac{C_2}{nC_4}\alpha^4\sigma^2 L^4
$$

$$
+3\frac{C_2}{C_4}\alpha^4 L^4 \frac{1}{K}\sum_{k=1}^{K-1}\mathbb{E}\left\|\overline{G}_k\right\|^2 + \frac{C_3 L^2}{C_4 nK}\sum_{k=1}^{K-1}\mathbb{E}\left\|\Omega_k\right\|_F^2
$$

*Rearrange the terms, we get*

$$
\left(1 - \frac{3C_1\alpha^2 L^2}{C_4}\right)\mathbb{E}\left\|\nabla f(0)\right\| + \left(1 - \alpha L - 3\frac{C_2}{C_4}\alpha^4 L^4\right)\frac{1}{K}\sum_{k=1}^{K-1}\mathbb{E}\left\|\overline{G}_k\right\|^2
$$

$$
+ \frac{1}{K}\sum_{k=0}^{K-1}\mathbb{E}\left\|\nabla f(\overline{X}_k)\right\|^2
$$

$$
\leq \frac{2(f(0)-f^*)}{\alpha K} + \frac{\alpha L}{n}\sigma^2 + \frac{3C_1\alpha^2 L^2(\sigma^2 + \varsigma_0^2)}{C_4 K} + 6\frac{C_2}{C_4}\alpha^2\sigma^2 L^2 + 3\frac{C_2}{nC_4}\alpha^4\sigma^2 L^4
$$

$$
+ \frac{C_3 L^2}{C_4 nK}\sum_{k=1}^{K-1}\mathbb{E}\left\|\Omega_k\right\|_F^2
$$

$$
\overset{\text{Lemma 16}}{\leq} \frac{2(f(0)-f^*)}{\alpha K} + \frac{\alpha L}{n}\sigma^2 + \frac{3C_1\alpha^2 L^2(\sigma^2 + \varsigma_0^2)}{C_4 K} + 6\frac{C_2}{C_4}\alpha^2\sigma^2 L^2 + 3\frac{C_2}{nC_4}\alpha^4\sigma^2 L^4
$$

$$
+ \frac{C_3 L^2}{C_4}\left(\frac{3D_1 n + 4}{3D_2 n}\right)^2\alpha^2 G_\infty^2 d
$$

*That completes the proof.*

**Lemma 16**

$$
\sum_{k=0}^{K-1}\mathbb{E}\left\|\Omega_k\right\|_F^2 \leq \left(\frac{3D_1 n + 4}{3D_2 n}\right)^2\alpha^2 G_\infty^2 dnK
$$

**Proof** *Similar to the case in D-PSGD, we have*

$$
\sum_{k=0}^{K-1}\mathbb{E}\left\|\Omega_k\right\|_F^2 = \sum_{k=0}^{K-1}\sum_{i=1}^{n}\mathbb{E}\left\|\sum_{j=1}^{n}\left((q_{k,j} - x_{k+\frac{1}{2},j}) - (q_{k,i} - x_{k+\frac{1}{2},i})\right)W_{ji}\right\|^2
$$

$$
\overset{\text{Lemma 3}}{\leq} 4\sum_{k=0}^{K-1}\sum_{i=1}^{n}\delta^2\theta^2 d \leq \left(\frac{3D_1 n + 4}{3D_2 n}\right)^2\alpha^2 G_\infty^2 dnK
$$

*That completes the proof.*

# H  MONIQUA ON AD-PSGD (PROOF TO THEOREM 4)

## H.1  ALGORITHM

---
**Algorithm 3** Moniqua with Asynchronous Communication
---
**Input:** initial point $x_{0,i} = x_0$, step size $\alpha$, the discrepency bound $\theta$, number of iterations $K$, quantization function $\mathcal{Q}$, initial random seed
 1: **for** $k = 0, 1, 2, \cdots, K - 1$ **do**
 2:     worker $i_k$ is updating the gradient while during this iteration the global communication behaviour is written in the form of $W_k$.
 3:     Compute a local stochastic gradient with model delayed by $\tau_k$: $\widetilde{g}_{k-\tau_k, i_k}$
 4:     Compute modulo-ed model: $q_{k,i_k} \leftarrow \theta \cdot \mathcal{Q}_\delta \left( \frac{x_{k,i_k}}{\theta} \mod 1 \right)$ (element-wise division and mod)
 5:     Randomly select one of the neighbors $j_k$ and average local weights with remote weights while subtracting the biased term: $x_{k+\frac{1}{2}, i_k} \leftarrow x_{k,i_k} + \frac{1}{2} q_{k,j_k} - \frac{1}{2} q_{k,i_k}$
 6:     Update the local weight with local gradient: $x_{k+1, i_k} \leftarrow x_{k,i_k} - \alpha \widetilde{g}_{k-\tau_k, i_k}$
 7: **end for**
**Output:** $\overline{X}_K = \frac{1}{n} \sum_{i=1}^{n} x_{K,i}$

---

## H.2  DEFINITION AND NOTATION

In the original analysis of AD-PSGD, to better capture the nature of workers computing at different speed, the objective function is expressed as

$$f(x) = \sum_{i=1}^{n} p_i f_i(x)$$

where $p_i$ is a parameter denoting the speed of $i$-th worker gradient updates. In the rest of the proof, we denote $p = \max_i \{p_i\}$

For simplicity, we also define the following terms

$$\nabla F(X_k) = n \left[ p_1 g_{k,1}, \cdots, p_n g_{k,n} \right] \in \mathbb{R}^{d \times n}$$
$$\nabla \widetilde{F}(X_k) = n \left[ p_1 \widetilde{g}_{k,1}, \cdots, p_n \widetilde{g}_{k,n} \right] \in \mathbb{R}^{d \times n}$$
$$\widetilde{G}_k = \left[ \cdots, \widetilde{g}_{k,i_k}, \cdots \right]$$
$$G_k = \left[ \cdots, g_{k,i_k}, \cdots \right]$$
$$\Lambda_a^b = \frac{\mathbb{1}_n \mathbb{1}_n^\top}{n} - \prod_{q=a}^{b} W_q$$

## H.3  ASSUMPTION

We makes the following assumptions:

1. **Lipschitzian Gradient**: All the function $f_i$ have L-Lipschitzian gradients.
2. **Communication Matrix** [12]: The communication matrix $W_k$ is doubly stochastic for any $k \geq 0$ and for any $b \geq a \geq 0$, there exists $t_{\mathrm{mix}}$ such that

$$\left\| \prod_{q=a}^{b} W_q \left( I - \frac{\mathbb{1}_n \mathbb{1}_n^\top}{n} \right) \right\|_1 \leq 2 \cdot 2^{-\left\lfloor \frac{b-a+1}{t_{\mathrm{mix}}} \right\rfloor}$$

3. **Bounded Variance**:

$$\mathbb{E}_{\xi_i \sim \mathcal{D}_i} \left\| \nabla \widetilde{f}_i(x_i; \xi_i) - \nabla f_i(x) \right\|^2 \leq \sigma^2, \forall i$$

---
[12]Please refer to Section E for more details

$$\mathbb{E}_{i \sim \{1, \cdots, n\}} \left\| \nabla f_i(x) - \nabla f(x) \right\|^2 \leq \varsigma^2, \forall i$$

where $\nabla \widetilde{f}_i(x; \xi_i)$ denotes gradient sample on worker $i$ computed via data sample $\xi_i$.

4. **Bounded Staleness**: There exists $T$ such that $\tau_k \leq T, \forall k$

5. **Gradient magnitude**: The norm of a sampled gradient is bounded by $\|\widetilde{g}_{k,i}\|_\infty \leq G_\infty$ for some constant $G_\infty$.

## H.4 Proof to Theorem 4

**Proof** *We start from*

$$\frac{1}{K} \sum_{k=0}^{K-1} \mathbb{E} \left\| \nabla f(\overline{X}_k) \right\|^2 + \left( 1 - \frac{2\alpha L}{n} \right) \frac{1}{K} \sum_{k=0}^{K-1} \mathbb{E} \left\| \nabla \overline{F}(X_{k-\tau_k}) \right\|^2$$

$$\overset{Lemma\ 20}{\leq} \frac{2n(f(0) - f^*)}{\alpha K} + \frac{(\sigma^2 + 6\varsigma^2)\alpha L}{n} + \left( 2L^2 + \frac{12\alpha L^3}{n} \right) \frac{1}{K} \sum_{k=0}^{K-1} \sum_{i=1}^{n} p_i \mathbb{E} \left\| X_{k-\tau_k} \left( \frac{\mathbb{1}_n}{n} - e_i \right) \right\|^2$$

$$+ \frac{2L^2}{K} \sum_{k=0}^{K-1} \mathbb{E} \left\| \frac{(X_k - X_{k-\tau_k})\mathbb{1}_n}{n} \right\|^2$$

$$\overset{Lemma\ 21}{\leq} \frac{2n(f(0) - f^*)}{\alpha K} + \frac{(\sigma^2 + 6\varsigma^2)\alpha L}{n} + \frac{2\alpha^2 T^2 (\sigma^2 + 6\varsigma^2) L^2}{n^2} + \frac{4\alpha^2 T^2 L^2}{n^2 K} \sum_{k=0}^{K-1} \mathbb{E} \left\| \sum_{i=1}^{n} p_i g_{k-\tau_k, i} \right\|^2$$

$$+ \left( 2L^2 + \frac{12\alpha L^3}{n} + \frac{24 L^4 \alpha^2 T^2}{n^2} \right) \frac{1}{K} \sum_{k=0}^{K-1} \sum_{i=1}^{n} p_i \mathbb{E} \left\| X_{k-\tau_k} \left( \frac{\mathbb{1}_n}{n} - e_i \right) \right\|^2$$

$$\overset{Lemma\ 19}{\leq} \frac{2n(f(0) - f^*)}{\alpha K} + \frac{(\sigma^2 + 6\varsigma^2)\alpha L}{n} + \frac{2\alpha^2 T^2 (\sigma^2 + 6\varsigma^2) L^2}{n^2} + \frac{4\alpha^2 T^2 L^2}{n^2 K} \sum_{k=0}^{K-1} \mathbb{E} \left\| \sum_{i=1}^{n} p_i g_{k-\tau_k, i} \right\|^2$$

$$+ \frac{128\alpha^2 t_{\mathrm{mix}}^2 L^2}{A_1} \left( (\sigma^2 + 6\varsigma^2) p + \frac{2p}{K} \sum_{k=0}^{K-1} \mathbb{E} \left\| \sum_{i=1}^{n} p_i g_{k-\tau_k, i} \right\|^2 + G_\infty^2 d \right)$$

*where $A_1 = 1 - 192 p \alpha^2 t_{\mathrm{mix}}^2 L^2$ as defined in Lemma 19.*

*Rearrange the terms, we get*

$$\frac{1}{K} \sum_{k=0}^{K-1} \mathbb{E} \left\| \nabla f(\overline{X}_k) \right\|^2 \leq \frac{2n(f(0) - f^*)}{\alpha K} + \frac{(\sigma^2 + 6\varsigma^2)\alpha L}{n} + \frac{2\alpha^2 T^2 (\sigma^2 + 6\varsigma^2) L^2}{n^2}$$

$$+ \frac{128 p \alpha^2 t_{\mathrm{mix}}^2 L^2}{A_1} (\sigma^2 + 6\varsigma^2) + \frac{128 \alpha^2 t_{\mathrm{mix}}^2 L^2}{A_1} G_\infty^2 d$$

*By setting $\alpha = \frac{n}{2L + \sqrt{K(\sigma^2 + 6\varsigma^2)}}$*

$$\frac{1}{K} \sum_{k=0}^{K-1} \mathbb{E} \left\| \nabla f(\overline{X}_k) \right\|^2 \lesssim \frac{1}{K} + \frac{\sqrt{\sigma^2 + 6\varsigma^2}}{\sqrt{K}} + \frac{p t_{\mathrm{mix}}^2 (\sigma^2 + 6\varsigma^2) n^2}{(\sigma^2 + 6\varsigma^2) K + 4L^2} + \frac{n^2 t_{\mathrm{mix}}^2 G_\infty^2 d}{(\sigma^2 + 6\varsigma^2) K + 4L^2}$$

$$\lesssim \frac{1}{K} + \frac{\sqrt{\sigma^2 + 6\varsigma^2}}{\sqrt{K}} + \frac{(\sigma^2 + 6\varsigma^2) t_{\mathrm{mix}}^2 n^2}{(\sigma^2 + 6\varsigma^2) K + 1} + \frac{n^2 t_{\mathrm{mix}}^2 G_\infty^2 d}{(\sigma^2 + 6\varsigma^2) K + 1}$$

## H.5 Lemma for Moniqua on AD-PSGD

**Lemma 17**

$$\mathbb{E} \left\| \widetilde{G}_{k-\tau_k} \frac{\mathbb{1}_n}{n} \right\|^2 \leq \frac{\sigma^2}{n^2} + \frac{1}{n^2} \sum_{i=1}^{n} p_i \mathbb{E} \left\| g_{k-\tau_k, i} \right\|^2, \forall k \geq 0.$$

**Proof**

$$\mathbb{E}\left\|\widetilde{G}_{k-\tau_k}\frac{\mathbb{1}_n}{n}\right\|^2 \leq \sum_{i=1}^n p_i \mathbb{E}\left\|\frac{\widetilde{g}_{k-\tau_k,i}}{n}\right\|^2$$

$$= \sum_{i=1}^n p_i \mathbb{E}\left\|\frac{\widetilde{g}_{k-\tau_k,i} - g_{k-\tau_k,i}}{n}\right\|^2 + \sum_{i=1}^n p_i \mathbb{E}\left\|\frac{g_{k-\tau_k,i}}{n}\right\|^2$$

$$\leq \frac{\sigma^2}{n^2} + \frac{1}{n^2}\sum_{i=1}^n p_i \mathbb{E}\left\|g_{k-\tau_k,i}\right\|^2$$

**Lemma 18**

$$\sum_{i=1}^n p_i \mathbb{E}\left\|g_{k-\tau_k,i}\right\|^2 \leq 12L^2\sum_{i=1}^n p_i \mathbb{E}\left\|X_{k-\tau_k}\left(\frac{\mathbb{1}_n}{n} - e_i\right)\right\|^2 + 6\varsigma^2 + 2\mathbb{E}\left\|\sum_{i=1}^n p_i g_{k-\tau_k,i}\right\|^2, \forall k \geq 0.$$

**Proof**

$$\sum_{i=1}^n p_i \mathbb{E}\left\|g_{k-\tau_k,i}\right\|^2 = \sum_{i=1}^n p_i \mathbb{E}\left\|g_{k-\tau_k,i} - \sum_{i=1}^n p_i g_{k-\tau_k,i} + \sum_{i=1}^n p_i g_{k-\tau_k,i}\right\|^2$$

$$\leq 2\sum_{i=1}^n p_i \mathbb{E}\left\|g_{k-\tau_k,i} - \sum_{i=1}^n p_i g_{k-\tau_k,i}\right\|^2 + 2\sum_{i=1}^n p_i \mathbb{E}\left\|\sum_{i=1}^n p_i g_{k-\tau_k,i}\right\|^2$$

$$= 2\sum_{i=1}^n p_i \mathbb{E}\left\|g_{k-\tau_k,i} - \sum_{i=1}^n p_i g_{k-\tau_k,i}\right\|^2 + 2\mathbb{E}\left\|\sum_{i=1}^n p_i g_{k-\tau_k,i}\right\|^2$$

*And*

$$\sum_{i=1}^n p_i \mathbb{E}\left\|g_{k-\tau_k,i} - \sum_{i=1}^n p_i g_{k-\tau_k,i}\right\|^2$$

$$\leq 3\sum_{i=1}^n p_i \mathbb{E}\left\|g_{k-\tau_k,i} - \nabla f_i(\overline{X}_{k-\tau_k})\right\|^2 + 3\sum_{i=1}^n p_i \mathbb{E}\left\|\nabla f_i(\overline{X}_{k-\tau_k}) - \sum_{j=1}^n p_j \nabla f_j(\overline{X}_{k-\tau_k})\right\|^2$$

$$+ 3\sum_{i=1}^n p_i \mathbb{E}\left\|\sum_{i=1}^n p_i g_{k-\tau_k,i} - \sum_{j=1}^n p_j \nabla f_j(\overline{X}_{k-\tau_k})\right\|^2$$

$$\leq 3L^2\sum_{i=1}^n p_i \mathbb{E}\left\|x_{k-\tau_k,i} - \overline{X}_{k-\tau_k}\right\|^2 + 3\sum_{i=1}^n p_i \mathbb{E}\left\|\nabla f_i(\overline{X}_{k-\tau_k}) - \sum_{j=1}^n p_j \nabla f_j(\overline{X}_{k-\tau_k})\right\|^2$$

$$+ 3\mathbb{E}\left\|\sum_{i=1}^n p_i g_{k-\tau_k,i} - \sum_{j=1}^n p_j \nabla f_j(\overline{X}_{k-\tau_k})\right\|^2$$

$$\leq 3L^2\sum_{i=1}^n p_i \mathbb{E}\left\|X_{k-\tau_k}\left(\frac{\mathbb{1}_n}{n} - e_i\right)\right\|^2 + 3\sum_{i=1}^n p_i \mathbb{E}\left\|\nabla f_i(\overline{X}_{k-\tau_k}) - \nabla f(\overline{X}_{k-\tau_k})\right\|^2$$

$$+ 3\sum_{j=1}^n p_j \mathbb{E}\left\|g_{k-\tau_k,j} - \nabla f_j(\overline{X}_{k-\tau_k})\right\|^2$$

$$\leq 6L^2\sum_{i=1}^n p_i \mathbb{E}\left\|X_{k-\tau_k}\left(\frac{\mathbb{1}_n}{n} - e_i\right)\right\|^2 + 3\varsigma^2$$

*That completes the proof.*

**Lemma 19** *Let* $A_1 = 1 - 192p\alpha^2 t_{\mathrm{mix}}^2 L^2$,

$$\sum_{k=0}^{K-1} \sum_{i=1}^{n} p_i \mathbb{E} \left\| X_{k-\tau_k} \left( \frac{\mathbb{1}_n}{n} - e_i \right) \right\|^2$$

$$\leq \frac{32\alpha^2 t_{\mathrm{mix}}^2}{A_1} \left( (\sigma^2 + 6\varsigma^2)pK + 2p \sum_{k=0}^{K-1} \mathbb{E} \left\| \sum_{i=1}^{n} p_i g_{k-\tau_k,i} \right\|^2 + G_\infty^2 dK \right)$$

**Proof**

$$\sum_{i=1}^{n} p_i \mathbb{E} \left\| X_k \left( \frac{\mathbb{1}_n}{n} - e_i \right) \right\|^2$$

$$= \sum_{i=1}^{n} p_i \mathbb{E} \left\| \left( X_{k-1} W_{k-1} - \alpha \widetilde{G}_{k-1-\tau_{k-1}} + \Omega_{k-1} \right) \left( \frac{\mathbb{1}_n}{n} - e_i \right) \right\|^2$$

$$\overset{X_0=0}{=} \sum_{i=1}^{n} p_i \mathbb{E} \left\| \sum_{t=0}^{k-1} \left( -\alpha \widetilde{G}_{t-\tau_t} + \Omega_t \right) \Lambda_{t+1}^{k-1} e_i \right\|^2$$

$$\leq 2 \sum_{i=1}^{n} p_i \mathbb{E} \left\| \sum_{t=0}^{k-1} \alpha \widetilde{G}_{t-\tau_t} \Lambda_{t+1}^{k-1} e_i \right\|^2 + 2 \sum_{i=1}^{n} p_i \mathbb{E} \left\| \sum_{t=0}^{k-1} \Omega_t \Lambda_{t+1}^{k-1} e_i \right\|^2$$

*Now for the first term, we have*

$$2 \sum_{i=1}^{n} p_i \mathbb{E} \left\| \sum_{t=0}^{k-1} \alpha \widetilde{G}_{t-\tau_t} \Lambda_{t+1}^{k-1} e_i \right\|^2 \leq 2p\alpha^2 \mathbb{E} \left\| \sum_{t=0}^{k-1} \widetilde{G}_{t-\tau_t} \Lambda_{t+1}^{k-1} \right\|_F^2$$

$$\leq 2p\alpha^2 \mathbb{E} \left( \sum_{t=0}^{k-1} \left\| \widetilde{G}_{t-\tau_t} \right\|_F \left\| \Lambda_{t+1}^{k-1} \right\| \right)^2$$

$$\leq 2p\alpha^2 \mathbb{E} \left( \sum_{t=0}^{k-1} \left\| \widetilde{G}_{t-\tau_t} \right\|_F \left\| \Lambda_{t+1}^{k-1} \right\|_1 \right)^2$$

$$\leq 8p\alpha^2 \mathbb{E} \left( \sum_{t=0}^{k-1} \left\| \widetilde{G}_{t-\tau_t} \right\|_F 2^{-\left\lfloor \frac{k-t-1}{t_{\mathrm{mix}}} \right\rfloor} \right)^2$$

*Now we replace $k$ with $k - \tau_k$, that is*

$$\sum_{i=1}^{n} p_i \mathbb{E} \left\| X_{k-\tau_k} \left( \frac{\mathbb{1}_n}{n} - e_i \right) \right\|^2$$

$$\leq 8p\alpha^2 \mathbb{E} \left( \sum_{t=0}^{k-\tau_k-1} \left\| \widetilde{G}_{t-\tau_t} \right\|_F 2^{-\left\lfloor \frac{k-\tau_k-t-1}{t_{\mathrm{mix}}} \right\rfloor} \right)^2 + 2 \sum_{i=1}^{n} p_i \mathbb{E} \left\| \sum_{t=0}^{k-\tau_k-1} \Omega_t \Lambda_{t+1}^{k-\tau_k-1} e_i \right\|^2$$

*Summing from $k = 0$ to $K - 1$ on both sides, we obtain*

$$\sum_{k=0}^{K-1} \sum_{i=1}^{n} p_i \mathbb{E} \left\| X_{k-\tau_k} \left( \frac{\mathbb{1}_n}{n} - e_i \right) \right\|^2$$

$$\leq 8p\alpha^2 \sum_{k=0}^{K-1} \mathbb{E} \left( \sum_{t=0}^{k-\tau_k-1} \left\| \widetilde{G}_{t-\tau_t} \right\|_F 2^{-\left\lfloor \frac{k-\tau_k-t-1}{t_{\mathrm{mix}}} \right\rfloor} \right)^2$$

$$+ 2 \sum_{i=1}^{n} p_i \sum_{k=0}^{K-1} \mathbb{E} \left\| \sum_{t=0}^{k-\tau_k-1} \Omega_t \Lambda_{t+1}^{k-\tau_k-1} e_i \right\|^2$$

$$\leq 8p\alpha^2 \sum_{k=0}^{K-1} \mathbb{E} \left( \sum_{t=0}^{k-\tau_k-1} \left\| \widetilde{G}_{t-\tau_t} \right\|_F 2^{-\left\lfloor \frac{k-\tau_k-t-1}{t_{\mathrm{mix}}} \right\rfloor} \right)^2$$

$$+ 2 \sum_{i=1}^{n} p_i \sum_{k=0}^{K-1} \mathbb{E} \left( \sum_{t=0}^{k-\tau_k-1} \left\| \Omega_t \right\|_{1,2} \left\| \Lambda_{t+1}^{k-\tau_k-1} \right\|_1 \left\| e_i \right\|_1 \right)^2$$

$$\leq 8p\alpha^2 \sum_{k=0}^{K-1} \mathbb{E} \left( \sum_{t=0}^{k-\tau_k-1} \left\| \widetilde{G}_{t-\tau_t} \right\|_F 2^{-\left\lfloor \frac{k-\tau_k-t-1}{t_{\mathrm{mix}}} \right\rfloor} \right)^2$$

$$+ 8 \sum_{i=1}^{n} p_i \sum_{k=0}^{K-1} \mathbb{E} \left( \sum_{t=0}^{k-\tau_k-1} \left\| \Omega_t \right\|_{1,2} 2^{-\left\lfloor \frac{k-\tau_k-t-1}{t_{\mathrm{mix}}} \right\rfloor} \right)^2$$

$$\overset{\textit{Lemma 22}}{\leq} 8p\alpha^2 \sum_{k=0}^{K-1} \mathbb{E} \left( \sum_{t=0}^{k-\tau_k-1} \left\| \widetilde{G}_{t-\tau_t} \right\|_F 2^{-\left\lfloor \frac{k-\tau_k-t-1}{t_{\mathrm{mix}}} \right\rfloor} \right)^2 + 32 t_{\mathrm{mix}}^2 \sum_{i=1}^{n} p_i \sum_{k=0}^{K-1} \mathbb{E} \left\| \Omega_k \right\|_{1,2}^2$$

$$\leq 8p\alpha^2 \sum_{k=0}^{K-1} \mathbb{E} \left( \sum_{t=0}^{k-\tau_k-1} \left\| \widetilde{G}_{t-\tau_t} \right\|_F 2^{-\left\lfloor \frac{k-\tau_k-t-1}{t_{\mathrm{mix}}} \right\rfloor} \right)^2 + 128 \delta^2 \theta^2 d t_{\mathrm{mix}}^2 K$$

$$\overset{\textit{Lemma 22}}{\leq} 32 p \alpha^2 t_{\mathrm{mix}}^2 \sum_{k=0}^{K-1} \mathbb{E} \left\| \widetilde{G}_{k-\tau_k} \right\|_F^2 + 128 \delta^2 \theta^2 d t_{\mathrm{mix}}^2 K$$

*Note that for the first term, we have*

$$\sum_{k=0}^{K-1} \mathbb{E} \left\| \widetilde{G}_{k-\tau_k} \right\|_F^2$$

$$= \sum_{k=0}^{K-1} \mathbb{E} \left\| \widetilde{g}_{k-\tau_k, i_k} \right\|^2$$

$$= \sum_{k=0}^{K-1} \mathbb{E} \left\| \widetilde{g}_{k-\tau_k, i_k} - g_{k-\tau_k, i_k} \right\|^2 + \sum_{k=0}^{K-1} \mathbb{E} \left\| g_{k-\tau_k, i_k} \right\|^2$$

$$\leq \sigma^2 K + \sum_{k=0}^{K-1} \sum_{i=1}^{n} p_i \mathbb{E} \left\| g_{t-\tau_t, i} \right\|^2$$

$$\leq (\sigma^2 + 6\varsigma^2) K + 12 L^2 \sum_{k=0}^{K-1} \sum_{i=1}^{n} p_i \mathbb{E} \left\| X_{k-\tau_k} \left( \frac{\mathbb{1}_n}{n} - e_i \right) \right\|^2 + 2 \sum_{k=0}^{K-1} \mathbb{E} \left\| \sum_{i=1}^{n} p_i g_{k-\tau_k, i} \right\|^2$$

*Putting these two terms back, we obtain*

$$\sum_{k=0}^{K-1} \sum_{i=1}^{n} p_i \mathbb{E} \left\| X_{k-\tau_k} \left( \frac{\mathbb{1}_n}{n} - e_i \right) \right\|^2$$

$$\leq 32 p \alpha^2 t_{\mathrm{mix}}^2 \left( (\sigma^2 + 6\varsigma^2) K + 12 L^2 \sum_{k=0}^{K-1} \sum_{i=1}^{n} p_i \mathbb{E} \left\| X_{k-\tau_k} \left( \frac{\mathbb{1}_n}{n} - e_i \right) \right\|^2 + 2 \sum_{k=0}^{K-1} \mathbb{E} \left\| \sum_{i=1}^{n} p_i g_{k-\tau_k, i} \right\|^2 \right)$$

$$+ 128 \delta^2 \theta^2 d t_{\mathrm{mix}}^2 K$$

*Rearrange the terms, we obtain*

$$\left( 1 - 192 p \alpha^2 t_{\mathrm{mix}}^2 L^2 \right) \sum_{k=0}^{K-1} \sum_{i=1}^{n} p_i \mathbb{E} \left\| X_{k-\tau_k} \left( \frac{\mathbb{1}_n}{n} - e_i \right) \right\|^2$$

$$\leq 32 p \alpha^2 t_{\mathrm{mix}}^2 \left( (\sigma^2 + 6\varsigma^2) K + 2 \sum_{k=0}^{K-1} \mathbb{E} \left\| \sum_{i=1}^{n} p_i g_{k-\tau_k, i} \right\|^2 \right) + 128 \delta^2 \theta^2 t_{\mathrm{mix}}^2 K$$

$$\overset{Lemma\ 23}{\leq} 32\alpha^2 t_{\mathrm{mix}}^2 \left( (\sigma^2 + 6\varsigma^2)pK + 2p \sum_{k=0}^{K-1} \mathbb{E} \left\| \sum_{i=1}^{n} p_i g_{k-\tau_k, i} \right\|^2 + G_\infty^2 dK \right)$$

*Let $A_1 = 1 - 192p\alpha^2 t_{\mathrm{mix}}^2 L^2$, we obtain*

$$\sum_{k=0}^{K-1} \sum_{i=1}^{n} p_i \mathbb{E} \left\| X_{k-\tau_k} \left( \frac{\mathbb{1}_n}{n} - e_i \right) \right\|^2$$

$$\leq \frac{32\alpha^2 t_{\mathrm{mix}}^2}{A_1} \left( (\sigma^2 + 6\varsigma^2)pK + 2p \sum_{k=0}^{K-1} \mathbb{E} \left\| \sum_{i=1}^{n} p_i g_{k-\tau_k, i} \right\|^2 + G_\infty^2 dK \right)$$

**Lemma 20**

$$\frac{1}{K} \sum_{k=0}^{K-1} \mathbb{E} \left\| \nabla f(\overline{X}_k) \right\|^2 + \left( 1 - \frac{2\alpha L}{n} \right) \frac{1}{K} \sum_{k=0}^{K-1} \mathbb{E} \left\| \nabla \overline{F}(X_{k-\tau_k}) \right\|^2$$

$$\leq \frac{2n(f(0) - f^*)}{\alpha K} + \frac{2L^2}{K} \sum_{k=0}^{K-1} \mathbb{E} \left\| \frac{(X_k - X_{k-\tau_k})\mathbb{1}_n}{n} \right\|^2$$

$$+ \left( 2L^2 + \frac{12\alpha L^3}{n} \right) \frac{1}{K} \sum_{k=0}^{K-1} \sum_{i=1}^{n} p_i \mathbb{E} \left\| X_{k-\tau_k} \left( \frac{\mathbb{1}_n}{n} - e_i \right) \right\|^2 + \frac{(\sigma^2 + 6\varsigma^2)\alpha L}{n}$$

**Proof** *We start from $f(\overline{X}_{k+1})$ Since*

$$\overline{X}_{k+1} = X_k W_k \frac{\mathbb{1}_n}{n} + (Q_k - X_k)(W_k - I)\frac{\mathbb{1}_n}{n} - \alpha \overline{\widetilde{G}}_{k-\tau_k} = \overline{X}_k - \alpha \overline{\widetilde{G}}_{k-\tau_k}$$

*Then from Taylor Expansion, we have*

$$\mathbb{E} f(\overline{X}_{k+1})$$

$$= \mathbb{E} f\left( \overline{X}_k - \alpha \overline{\widetilde{G}}_{k-\tau_k} \right)$$

$$\leq \mathbb{E} f(\overline{X}_k) - \alpha \mathbb{E} \langle \nabla f(\overline{X}_k), \overline{\widetilde{G}}_{k-\tau_k} \rangle + \frac{\alpha^2 L}{2} \mathbb{E} \left\| \overline{\widetilde{G}}_{k-\tau_k} \right\|^2$$

$$= \mathbb{E} f(\overline{X}_k) - \alpha \mathbb{E} \langle \nabla f(\overline{X}_k), \overline{G}_{k-\tau_k} \rangle - \alpha \mathbb{E} \langle \nabla f(\overline{X}_k), \overline{\widetilde{G}}_{k-\tau_k} - \overline{G}_{k-\tau_k} \rangle + \frac{\alpha^2 L}{2} \mathbb{E} \left\| \overline{\widetilde{G}}_{k-\tau_k} \right\|^2$$

$$= \mathbb{E} f(\overline{X}_k) - \frac{\alpha}{n} \mathbb{E} \langle \nabla f(\overline{X}_k), \nabla \overline{F}(X_{k-\tau_k}) \rangle + \frac{\alpha^2 L}{2} \mathbb{E} \left\| \frac{\widetilde{g}_{k-\tau_k, i_k}}{n} \right\|^2$$

$$\leq \mathbb{E} f(\overline{X}_k) - \frac{\alpha}{n} \mathbb{E} \langle \nabla f(\overline{X}_k), \nabla \overline{F}(X_{k-\tau_k}) \rangle$$

$$+ \frac{\alpha^2 L}{2} \sum_{i=1}^{n} p_i \mathbb{E} \left\| \frac{\widetilde{g}_{k-\tau_k, i_k} - g_{k-\tau_k, i_k}}{n} \right\|^2 + \frac{\alpha^2 L}{2} \sum_{i=1}^{n} p_i \mathbb{E} \left\| \frac{g_{k-\tau_k, i}}{n} \right\|^2$$

$$\leq \mathbb{E} f(\overline{X}_k) - \frac{\alpha}{n} \mathbb{E} \langle \nabla f(\overline{X}_k), \nabla \overline{F}(X_{k-\tau_k}) \rangle + \frac{\alpha^2 L \sigma^2}{2n^2} + \frac{\alpha^2 L}{2n^2} \sum_{i=1}^{n} p_i \mathbb{E} \| g_{k-\tau_k, i} \|^2$$

$$= \mathbb{E} f(\overline{X}_k) + \frac{\alpha}{2n} \mathbb{E} \left\| \nabla f(\overline{X}_k) - \nabla \overline{F}(X_{k-\tau_k}) \right\|^2 - \frac{\alpha}{2n} \mathbb{E} \left\| \nabla f(\overline{X}_k) \right\|^2 - \frac{\alpha}{2n} \mathbb{E} \left\| \nabla \overline{F}(X_{k-\tau_k}) \right\|^2$$

$$+ \frac{\alpha^2 L \sigma^2}{2n^2} + \frac{\alpha^2 L}{2n^2} \sum_{i=1}^{n} p_i \mathbb{E} \| g_{k-\tau_k, i} \|^2$$

*Rearrange these terms, we can get*

$$\frac{\alpha}{2n} \mathbb{E} \left\| \nabla f(\overline{X}_k) \right\|^2 + \frac{\alpha}{2n} \mathbb{E} \left\| \nabla \overline{F}(X_{k-\tau_k}) \right\|^2$$

$$\leq \mathbb{E} f(\overline{X}_k) - \mathbb{E} f(\overline{X}_{k+1}) + \frac{\alpha}{2n} \mathbb{E} \left\| \nabla f(\overline{X}_k) - \nabla \overline{F}(X_{k-\tau_k}) \right\|^2$$

$$+ \frac{\alpha^2 L \sigma^2}{2n^2} + \frac{\alpha^2 L}{2n^2} \sum_{i=1}^{n} p_i \mathbb{E} \|g_{k-\tau_k,i}\|^2$$

*Summing over $k = 0$ to $K - 1$ on both sides, we can get*

$$\frac{1}{K} \sum_{k=0}^{K-1} \mathbb{E} \left\| \nabla f(\overline{X}_k) \right\|^2 + \frac{1}{K} \sum_{k=0}^{K-1} \mathbb{E} \left\| \nabla \overline{F}(X_{k-\tau_k}) \right\|^2$$

$$\leq \frac{2n(f(0) - f^*)}{\alpha K} + \frac{1}{K} \sum_{k=0}^{K-1} \mathbb{E} \left\| \nabla f(\overline{X}_k) - \nabla \overline{F}(X_{k-\tau_k}) \right\|^2 + \frac{\alpha L \sigma^2}{n} + \frac{\alpha L}{nK} \sum_{k=0}^{K-1} \sum_{i=1}^{n} p_i \mathbb{E} \|g_{k-\tau_k,i}\|^2$$

*For $\sum_{k=0}^{K-1} \mathbb{E} \left\| \nabla f(\overline{X}_k) - \nabla \overline{F}(X_{k-\tau_k}) \right\|^2$, we have*

$$\sum_{k=0}^{K-1} \mathbb{E} \left\| \nabla f(\overline{X}_k) - \nabla \overline{F}(X_{k-\tau_k}) \right\|^2$$

$$\leq 2 \sum_{k=0}^{K-1} \mathbb{E} \left\| \nabla f(\overline{X}_k) - \nabla f(\overline{X}_{k-\tau_k}) \right\|^2 + 2 \sum_{k=0}^{K-1} \mathbb{E} \left\| \nabla f(\overline{X}_{k-\tau_k}) - \nabla \overline{F}(X_{k-\tau_k}) \right\|^2$$

$$= 2 \sum_{k=0}^{K-1} \mathbb{E} \left\| \nabla f(\overline{X}_k) - \nabla f(\overline{X}_{k-\tau_k}) \right\|^2 + 2 \sum_{k=0}^{K-1} \mathbb{E} \left\| \sum_{i=1}^{n} p_i \left( \nabla f_i(\overline{X}_{k-\tau_k}) - g_{k-\tau_k,i} \right) \right\|^2$$

$$\leq 2 \sum_{k=0}^{K-1} \mathbb{E} \left\| \nabla f(\overline{X}_k) - \nabla f(\overline{X}_{k-\tau_k}) \right\|^2 + 2 \sum_{k=0}^{K-1} \mathbb{E} \sum_{i=1}^{n} p_i \left\| \nabla f_i(\overline{X}_{k-\tau_k}) - g_{k-\tau_k,i} \right\|^2$$

$$\leq 2L^2 \sum_{k=0}^{K-1} \mathbb{E} \left\| \frac{(X_k - X_{k-\tau_k}) \mathbb{1}_n}{n} \right\|^2 + 2L^2 \sum_{k=0}^{K-1} \sum_{i=1}^{n} p_i \mathbb{E} \left\| X_{k-\tau_k} \left( \frac{\mathbb{1}_n}{n} - e_i \right) \right\|^2$$

*Putting it back, we have*

$$\frac{1}{K} \sum_{k=0}^{K-1} \mathbb{E} \left\| \nabla f(\overline{X}_k) \right\|^2 + \frac{1}{K} \sum_{k=0}^{K-1} \mathbb{E} \left\| \nabla \overline{F}(X_{k-\tau_k}) \right\|^2$$

$$\leq \frac{2n(f(0) - f^*)}{\alpha K} + \frac{2L^2}{K} \sum_{k=0}^{K-1} \mathbb{E} \left\| \frac{(X_k - X_{k-\tau_k}) \mathbb{1}_n}{n} \right\|^2$$

$$+ \frac{2L^2}{K} \sum_{k=0}^{K-1} \sum_{i=1}^{n} p_i \mathbb{E} \left\| X_{k-\tau_k} \left( \frac{\mathbb{1}_n}{n} - e_i \right) \right\|^2 + \frac{\alpha L \sigma^2}{n} + \frac{\alpha L}{nK} \sum_{k=0}^{K-1} \sum_{i=1}^{n} p_i \mathbb{E} \|g_{k-\tau_k,i}\|^2$$

$$\overset{Lemma\ 18}{\leq} \frac{2n(f(0) - f^*)}{\alpha K} + \frac{2L^2}{K} \sum_{k=0}^{K-1} \mathbb{E} \left\| \frac{(X_k - X_{k-\tau_k}) \mathbb{1}_n}{n} \right\|^2$$

$$+ \frac{2L^2}{K} \sum_{k=0}^{K-1} \sum_{i=1}^{n} p_i \mathbb{E} \left\| X_{k-\tau_k} \left( \frac{\mathbb{1}_n}{n} - e_i \right) \right\|^2 + \frac{\alpha L \sigma^2}{n}$$

$$+ \frac{\alpha L}{nK} \sum_{k=0}^{K-1} \left( 12L^2 \sum_{i=1}^{n} p_i \mathbb{E} \left\| X_{k-\tau_k} \left( \frac{\mathbb{1}_n}{n} - e_i \right) \right\|^2 + 6\varsigma^2 + 2\mathbb{E} \left\| \sum_{i=1}^{n} p_i g_{k-\tau_k,i} \right\|^2 \right)$$

$$= \frac{2n(f(0) - f^*)}{\alpha K} + \frac{2L^2}{K} \sum_{k=0}^{K-1} \mathbb{E} \left\| \frac{(X_k - X_{k-\tau_k}) \mathbb{1}_n}{n} \right\|^2$$

$$+ \left( 2L^2 + \frac{12\alpha L^3}{n} \right) \frac{1}{K} \sum_{k=0}^{K-1} \sum_{i=1}^{n} p_i \mathbb{E} \left\| X_{k-\tau_k} \left( \frac{\mathbb{1}_n}{n} - e_i \right) \right\|^2$$

$$+ \frac{(\sigma^2 + 6\varsigma^2)\alpha L}{n} + \frac{2\alpha L}{nK} \sum_{k=0}^{K-1} \mathbb{E} \left\| \sum_{i=1}^{n} p_i g_{k-\tau_k, i} \right\|^2$$

*Note that*

$$\mathbb{E} \left\| \sum_{i=1}^{n} p_i g_{k-\tau_k, i} \right\|^2 = \mathbb{E} \left\| \nabla \overline{F}(X_{k-\tau_k}) \right\|^2$$

*Moving it to the left side, we finally get*

$$\frac{1}{K} \sum_{k=0}^{K-1} \mathbb{E} \left\| \nabla f(\overline{X}_k) \right\|^2 + \left( 1 - \frac{2\alpha L}{n} \right) \frac{1}{K} \sum_{k=0}^{K-1} \mathbb{E} \left\| \nabla \overline{F}(X_{k-\tau_k}) \right\|^2$$

$$\leq \frac{2n(f(0) - f^*)}{\alpha K} + \frac{2L^2}{K} \sum_{k=0}^{K-1} \mathbb{E} \left\| \frac{(X_k - X_{k-\tau_k}) \mathbb{1}_n}{n} \right\|^2$$

$$+ \left( 2L^2 + \frac{12\alpha L^3}{n} \right) \frac{1}{K} \sum_{k=0}^{K-1} \sum_{i=1}^{n} p_i \mathbb{E} \left\| X_{k-\tau_k} \left( \frac{\mathbb{1}_n}{n} - e_i \right) \right\|^2 + \frac{(\sigma^2 + 6\varsigma^2)\alpha L}{n}$$

*That completes the proof.*

**Lemma 21** *For all $k \geq 0$, we have*

$$\frac{2L^2}{K} \sum_{k=0}^{K-1} \mathbb{E} \left\| (X_k - X_{k-\tau_k}) \frac{\mathbb{1}_n}{n} \right\|^2$$

$$\leq \frac{2\alpha^2 T^2 (\sigma^2 + 6\varsigma^2) L^2}{n^2} + \frac{24 L^4 \alpha^2 T^2}{n^2 K} \sum_{k=0}^{K-1} \sum_{i=1}^{n} p_i \mathbb{E} \left\| X_{k-\tau_k} \left( \frac{\mathbb{1}_n}{n} - e_i \right) \right\|^2$$

$$+ \frac{4\alpha^2 T^2 L^2}{n^2 K} \sum_{k=0}^{K-1} \mathbb{E} \left\| \sum_{i=1}^{n} p_i g_{k-\tau_k, i} \right\|^2$$

**Proof** *From Lemma 20, we know the fact*

$$\overline{X}_{k+1} = X_k W_k \frac{\mathbb{1}_n}{n} + (Q_k - X_k)(W_k - I) \frac{\mathbb{1}_n}{n} - \alpha \overline{\widetilde{G}}_{k-\tau_k} = \overline{X}_k - \alpha \overline{\widetilde{G}}_{k-\tau_k}$$

*As a result*

$$\sum_{k=0}^{K-1} \mathbb{E} \left\| (X_k - X_{k-\tau_k}) \frac{\mathbb{1}_n}{n} \right\|^2$$

$$= \sum_{k=0}^{K-1} \mathbb{E} \left\| \sum_{t=1}^{\tau_k} \alpha \widetilde{G}_{k-t} \frac{\mathbb{1}_n}{n} \right\|^2$$

$$\leq \alpha^2 \sum_{k=0}^{K-1} \tau_k \sum_{t=1}^{\tau_k} \mathbb{E} \left\| \widetilde{G}_{k-t} \frac{\mathbb{1}_n}{n} \right\|^2$$

$$\leq \alpha^2 \sum_{k=0}^{K-1} \tau_k \sum_{t=1}^{\tau_k} \left( \frac{\sigma^2}{n^2} + \frac{1}{n^2} \sum_{i=1}^{n} p_i \mathbb{E} \left\| g_{k-t, i} \right\|^2 \right)$$

$$\leq \frac{\alpha^2 T^2 \sigma^2 K}{n^2} + \frac{\alpha^2 T}{n^2} \sum_{k=0}^{K-1} \sum_{t=1}^{\tau_k} \sum_{i=1}^{n} p_i \mathbb{E} \left\| g_{k-t, i} \right\|^2$$

$$\leq \frac{\alpha^2 T^2 \sigma^2 K}{n^2} + \frac{\alpha^2 T}{n^2} \sum_{k=0}^{K-1} \sum_{t=1}^{\tau_k} \left( 12 L^2 \sum_{i=1}^{n} p_i \mathbb{E} \left\| X_{k-t} \left( \frac{\mathbb{1}_n}{n} - e_i \right) \right\|^2 + 6\varsigma^2 + 2\mathbb{E} \left\| \sum_{i=1}^{n} p_i g_{k-t, i} \right\|^2 \right)$$

$$\leq \frac{\alpha^2 T^2 \sigma^2 K}{n^2} + \frac{\alpha^2 T^2}{n^2} \sum_{k=0}^{K-1} \left( 12 L^2 \sum_{i=1}^{n} p_i \mathbb{E} \left\| X_{k-\tau_k} \left( \frac{\mathbb{1}_n}{n} - e_i \right) \right\|^2 + 6\varsigma^2 + 2\mathbb{E} \left\| \sum_{i=1}^{n} p_i g_{k-\tau_k, i} \right\|^2 \right)$$

$$= \frac{\alpha^2 T^2 (\sigma^2 + 6\varsigma^2) K}{n^2} + \frac{12 L^2 \alpha^2 T^2}{n^2} \sum_{k=0}^{K-1} \sum_{i=1}^{n} p_i \mathbb{E} \left\| X_{k-\tau_k} \left( \frac{\mathbb{1}_n}{n} - e_i \right) \right\|^2$$

$$+ \frac{2\alpha^2 T^2}{n^2} \sum_{k=0}^{K-1} \mathbb{E} \left\| \sum_{i=1}^{n} p_i g_{k-\tau_k, i} \right\|^2$$

*And we get*

$$\frac{2L^2}{K} \sum_{k=0}^{K-1} \mathbb{E} \left\| (X_k - X_{k-\tau_k}) \frac{\mathbb{1}_n}{n} \right\|^2$$

$$\leq \frac{2\alpha^2 T^2 (\sigma^2 + 6\varsigma^2) L^2}{n^2} + \frac{24 L^4 \alpha^2 T^2}{n^2 K} \sum_{k=0}^{K-1} \sum_{i=1}^{n} p_i \mathbb{E} \left\| X_{k-\tau_k} \left( \frac{\mathbb{1}_n}{n} - e_i \right) \right\|^2$$

$$+ \frac{4\alpha^2 T^2 L^2}{n^2 K} \sum_{k=0}^{K-1} \mathbb{E} \left\| \sum_{i=1}^{n} p_i g_{k-\tau_k, i} \right\|^2$$

*That completes the proof.*

**Lemma 22** *Given non-negative sequences $\{a_t\}_{t=1}^{\infty}$, $\{b_t\}_{t=1}^{\infty}$ and $\{\tau_t\}_{t=1}^{\infty}$ and a positive number $T$ that satisfying*

$$a_t = \sum_{s=1}^{t-\tau_t} \rho^{\lfloor \frac{t-\tau_t - s}{T} \rfloor} b_s$$

*with $0 \leq \rho < 1$, we have*

$$S_k = \sum_{t=1}^{k} a_t \leq \frac{(2-\rho)T}{1-\rho} \sum_{s=1}^{k} b_s$$

$$D_k = \sum_{t=1}^{k} a_t^2 \leq \frac{(2-\rho)T^2}{(1-\rho)^2} \sum_{s=1}^{k} b_s^2$$

**Proof**

$$S_k = \sum_{t=1}^{k} a_t = \sum_{t=1}^{k} \sum_{s=1}^{t-\tau_t} \rho^{\lfloor \frac{t-\tau_t - s}{T} \rfloor} b_s \leq \sum_{t=1}^{k} \sum_{s=1}^{t} \rho^{\max(\lfloor \frac{t-\tau_t - s}{T} \rfloor, 0)} b_s = \sum_{s=1}^{k} \sum_{t=s}^{k} \rho^{\max(\lfloor \frac{t-\tau_t - s}{T} \rfloor, 0)} b_s$$

$$= \sum_{s=1}^{k} \sum_{t=0}^{k-\tau_k - s} \rho^{\lfloor \frac{t}{T} \rfloor} b_s + \sum_{s=1}^{k} \sum_{t=1}^{\tau_k} \rho^0 b_s \leq \sum_{s=1}^{k} \left( \sum_{t=0}^{T-1} \sum_{m=0}^{\infty} \rho^m \right) b_s + \tau_k \sum_{s=1}^{k} b_s \leq \left( T + \frac{T}{1-\rho} \right) \sum_{s=1}^{k} b_s$$

$$D_k = \sum_{t=1}^{k} a_t^2 = \sum_{t=1}^{k} \sum_{s=1}^{t-\tau_t} \rho^{\lfloor \frac{t-\tau_t - s}{T} \rfloor} b_s \sum_{r=1}^{t-\tau_t} \rho^{\lfloor \frac{t-\tau_t - r}{T} \rfloor} b_r = \sum_{t=1}^{k} \sum_{s=1}^{t-\tau_t} \sum_{r=1}^{t-\tau_t} \rho^{\lfloor \frac{t-\tau_t - s}{T} \rfloor + \lfloor \frac{t-\tau_t - r}{T} \rfloor} b_s b_r$$

$$\leq \sum_{t=1}^{k} \sum_{s=1}^{t-\tau_t} \sum_{r=1}^{t-\tau_t} \rho^{\lfloor \frac{t-\tau_t - s}{T} \rfloor + \lfloor \frac{t-\tau_t - r}{T} \rfloor} \frac{b_s^2 + b_r^2}{2} = \sum_{t=1}^{k} \sum_{s=1}^{t-\tau_t} \sum_{r=1}^{t-\tau_t} \rho^{\lfloor \frac{t-\tau_t - s}{T} \rfloor + \lfloor \frac{t-\tau_t - r}{T} \rfloor} b_s^2$$

$$\leq \sum_{t=1}^{k} \sum_{s=1}^{t-\tau_t} b_s^2 \rho^{\lfloor \frac{t-\tau_t - s}{T} \rfloor} \sum_{r=1}^{t-\tau_t} \rho^{\lfloor \frac{t-\tau_t - r}{T} \rfloor} \leq \sum_{t=1}^{k} \sum_{s=1}^{t-\tau_t} b_s^2 \rho^{\lfloor \frac{t-\tau_t - s}{T} \rfloor} \sum_{r=0}^{T-1} \sum_{m=0}^{\infty} \rho^m$$

$$cs6 \leq \frac{T}{1-\rho} \sum_{t=1}^{k} \sum_{s=1}^{t-\tau_t} \rho^{\lfloor \frac{t-\tau_t - s}{T} \rfloor} b_s^2 \overset{Using S_k}{\leq} \frac{(2-\rho)T^2}{(1-\rho)^2} \sum_{s=1}^{k} b_s^2$$

**Lemma 23** *for $\forall i, j$ and $\forall k \geq 0$, we have*

$$\| X_k (e_i - e_j) \|_\infty \leq \theta = 16 t_{\mathrm{mix}} \alpha G_\infty$$

**Proof** *Similar to Section F and Section G, we use mathmatical induction to prove this.*

*I. First, for $k = 0$, we have*

$$\|X_k(e_i - e_j)\|_\infty = 0 \le \theta = 16 t_{\mathrm{mix}} \alpha G_\infty$$

*II. Suppose for $k \ge 0$, we have $\|X_t(e_i - e_j)\|_\infty \le \theta$, $\forall t \le k$, then we have*

$$\|X_{k+1}(e_i - e_j)\|_\infty$$

$$\le \left\| X_{k+1}\left(\frac{\mathbb{1}_n}{n} - e_i\right) \right\|_\infty + \left\| X_{k+1}\left(\frac{\mathbb{1}_n}{n} - e_j\right) \right\|_\infty$$

$$\le \left\| X_{k+1}\left(I - \frac{\mathbb{1}_n \mathbb{1}_n^\top}{n}\right) \right\|_{1,\infty} \|e_i\|_1 + \left\| X_{k+1}\left(I - \frac{\mathbb{1}_n \mathbb{1}_n^\top}{n}\right) \right\|_{1,\infty} \|e_j\|_1$$

$$= 2 \left\| X_{k+1}\left(I - \frac{\mathbb{1}_n \mathbb{1}_n^\top}{n}\right) \right\|_{1,\infty}$$

$$\le 2 \left\| \left(X_k W_k - \alpha \widetilde{G}_{k-\tau_k} + \Omega_k\right)\left(\frac{\mathbb{1}_n}{n} - e_i\right) \right\|_{1,\infty}$$

$$= 2 \left\| \sum_{t=0}^k \left(-\alpha \widetilde{G}_{t-\tau_t} + \Omega_t\right)\left(\prod_{q=t+1}^k W_q - \frac{\mathbb{1}_n \mathbb{1}_n^\top}{n}\right) \right\|_{1,\infty}$$

$$\le 2 \sum_{t=0}^k \left\| \left(-\alpha \widetilde{G}_{t-\tau_t} + \Omega_t\right)\left(\prod_{q=t+1}^k W_q - \frac{\mathbb{1}_n \mathbb{1}_n^\top}{n}\right) \right\|_{1,\infty}$$

$$\le 2 \sum_{t=0}^k \left\| -\alpha \widetilde{G}_{t-\tau_t} + \Omega_t \right\|_{1,\infty} \left\| \prod_{q=t+1}^k W_q - \frac{\mathbb{1}_n \mathbb{1}_n^\top}{n} \right\|_1$$

$$\le 4(\alpha G_\infty + 2\delta\theta) \sum_{t=0}^k 2^{-\lfloor (k-t)/t_{\mathrm{mix}} \rfloor}$$

$$\le 4(\alpha G_\infty + 2\delta\theta) \sum_{t=0}^{t_{\mathrm{mix}}-1} \sum_{r=0}^{\infty} 2^{-r}$$

$$\le 8(\alpha G_\infty + 2\delta\theta) t_{\mathrm{mix}}$$

*Put in $\delta = \frac{1}{32 t_{\mathrm{mix}}}$, we obtain*

$$\|X_{k+1}(e_i - e_j)\|_2 \le 8(\alpha G_\infty + 2\delta\theta) t_{\mathrm{mix}} = 8 t_{\mathrm{mix}} \alpha G_\infty + 8 t_{\mathrm{mix}} \alpha G_\infty = 16 t_{\mathrm{mix}} \alpha G_\infty$$

*Combining I and II and we complete the proof.*

