# OpenReview forum: "Moniqua: Modulo Quantized Communication in Decentralized SGD"
_ICLR.cc/2020/Conference — Reject_

### Official Review · AnonReviewer1 · 2019-10-23
**Official Blind Review #1**

**Rating:** 3

**Review:**

This paper studies an important problem in the decentralized optimization, i.e., communication compression. Unlike gradient compression, the model compression in decentralized optimization is more challenging because the model parameters will not vanish like gradient. To solve this problem, the authors proposed Moniqua, where only lower-order bits of model are communicated since their higher-order bits are getting close. A hyperparameter $\theta$ is used as a “criterion” to separate the higher-order bits and lower-order bits of the model parameter via a modular arithmetic. The authors also apply Moniqua on D^2 and AD-PSGD.

There are several major concerns:
1.	It is not clear how the performance is measured in this paper. If the authors simply take the average of training loss over all nodes, this is not a good evaluation unless consensus is achieved. Unless clarifying this, it is hard to judge the performance.
2.	It is better to provide evaluation on the consensus error which meansures the model consistency.
3.	The modular hyperparameter $\theta$ is not easy to choose and seems cannot help achieve consensus. On the one hand, the theory suggests its value to be proportional to the gradient magnitude bound, which could be very large in practice. But if $\theta$ is large, we cannot achieve even approximate consensus because of the quantization error. On the other hand, a small $\theta$ cannot ensure sufficient model average since higher-order bits are ignored by the modular arithmetic but actually they should be communicated. Either way, it doesn’t seem to be good for achieving consensus.

Overall, the idea of Moniqua is interesting and the authors provide useful extensions based on it. But the evaluation is not convincing and whether the proposed Moniqua can achieve consensus is unclear.




**Experience Assessment:**

I have published one or two papers in this area.

**Review Assessment: Checking Correctness Of Derivations And Theory:**

I assessed the sensibility of the derivations and theory.

**Review Assessment: Checking Correctness Of Experiments:**

I carefully checked the experiments.

**Review Assessment: Thoroughness In Paper Reading:**

I read the paper at least twice and used my best judgement in assessing the paper.

---

> ### Author Response · Authors · 2019-11-14
> **Thank you very much for the detailed reviews and comments!**
>
> Thank you very much for your reviews and comments! Below we address your each concern individually:
>
> ==========Response to Concern 1.==========
> Thanks for sharing this insightful concern! To clarify, all the performances are measured on the averaged model, which is a model that takes the average over all the model parameters on all the workers. We are measuring the averaged model since this is the standard output of decentralized SGD algorithms, including D-PSGD, D^2, AD-PSGD and Moniqua (Please refer to the output of Algorithm 1 on page 4). In the experiments, we use an independent process in the backend to compute the averaged model and measure its training and test accuracy/loss by a forward pass over the DNN model on the training and test set.
>
> ==========Response to Concern 2.==========
> In theory, we prove all the workers will reach consensus in Lemma 6 on page 23 for Moniqua (Lemma 12 and 19 for the two extensions.)
>
> In validation, you are correct that results on the consensus error should also be included. We provide extra experimental results in Appendix C.5 of the revised paper. These experiments show that: 1) The average distance among workers are decreasing to zero even with aggressive quantization (2 and 3 bits) as the setting in the original paper, where constant step size is used. 2) With decreasing step size, the average distance among workers are also decreasing to zero. We validate this on both ResNet110 and ResNet18.
>
> ==========Response to Concern 3.==========
> First, we’d like to point out that $\theta$ is proportional to the infinity norm of gradient, which is generally independent of the model size. And we have a theoretical result that guarantees consensus among workers (Lemma 6 on page 23 for Moniqua, Lemma 12 and 19 for the two extensions).
>
> Second, in practice we could start choosing or tuning $\theta$ from a small value, even if the value we end up choosing is smaller than the true $\theta$, this also works. To reason about this, consider the toy example with two workers as shown in the “Introduction”: with true $\theta$, $m_1=m_1+\frac{1}{2}(m_2-m_1)$. If $\theta$ is smaller than the true $\theta$, then the equation becomes $m_1=m_1+\frac{\gamma}{2}(m_2-m_1)$ with some factor  $\gamma$. Note that this new expression is equivalent to previous work of using deteriorating factor or extrapolation  in communication (Reisizadeh et al., 2018;Koloskova et al., 2019), which also leads to convergence as long as  $\gamma$ is not extremely small. From a theoretical perspective, this means we are using a communication matrix with slightly small spectral gap as defined in Assumption 2 on page 4.
>
> There are also several effective ways of choosing $\theta$ in practice,  please refer to “Response to Question 10” to Reviewer 3.
>
> Thanks again for your insightful suggestions and concerns! If you think our clarification and the extra results address your concerns, we would be appreciated if you could reconsider your score for this paper. On the other hand, if you believe there are still significant flaws or questions, please let us know and we would be happy to improve it!

---

### Official Review · AnonReviewer3 · 2019-10-23
**Official Blind Review #3**

**Rating:** 3

**Review:**

This paper considers training of DL models in a decentralized setting. Previous work (Tang et al 2018b, Koloskova et al 2019, Tang et al 2019) introduced communication compression in order to reduce communication cost in decentralized SGD. This paper shows that the naive way to apply quantization to compress communication can fail (but, none of the previous works used this naive way). The paper proposes a novel decentralized SGD algorithm called MONIQUA which communicates only with compressed vectors. The paper theoretically proves that MONIQUA asymptotically converges with the same speed as D-PSGD (Decentralized SGD with full communications) for non-convex functions.
The main benefit of the proposed algorithm is that in contrast to the baselines, MONIQUA doesn’t require any additional memory and computation overhead connected to that.
The paper also allow to combine MONIQUA with asynchronous communications (AD-PSGD) and decentralized data (D^2), which is novel, previous baseline didn’t allow asynchronous communications. Paper experimentally validates MONIQUA and show that it converges faster than the baselines on early optimization stage, but omit comparison of final results.

My score is weak reject. The method is interesting and elegant, doesn’t require additional memory, theoretically convergent with a good speed and allows asynchronous communication. However, it looks a bit incremental. In contrast to the baselines, MONIQUA does not support arbitrary communication compression. Experimental comparison is shown only for the beginning of the optimization where the algorithm doesn’t achieve state of the art accuracy.

Major concerns:

1. Superior experimental results are shown only for (i) the constant stepsize schedule and (ii) only in the beginning of the optimization. The algorithm is unable to achieve s.o.t.a. test accuracy and has severe accuracy drop of 10-20%. It is unclear if MONIQUA is able to close the accuracy gap.
To get good test accuracy the exponentially decaying learning rate schedules are usually used when training ResNet on Cifar10. However, that is unclear if this learning rate schedule could be supported by the algorithm. I see that theoretically the same result would be not possible. Once the learning rate dropped, ||x_i - x_j|| has to be smaller than the new theta, which is not true.

2. In contrast to the baselines, MONIQUA doesn’t support arbitrary quantization. The quantization level is fixed. For example, for the ring 8 case (which was used in experiments), the number of bits required to converge is at least 5 (I calculated it using expression for B on page 5). This also questions if MONIQUA will be able to converge to good accuracy in extreme bit-budget.


Other concerns that should be addressed:

3. The spectral gap is usually not a constant: for example for the ring topology it decreases as 1/n^2. That should be clarified in the convergence rate and the number of bits required to communicate.

4. In the experiments, to ensure fair comparison, the tuning details of the averaging rate for DeepSqueeze and Choco baselines, and theta parameter for MONIQUA were not given.

5. Reproducibility of the experiments: Many experimental details are omitted:
   - What are the values of hyperparameters used in the plots?
   - How did you model asynchrony in experiments?
   - How did you choose the parameter \theta in experiments?

6. In (1), why x is \in [-1, 1]^d? What if this assumption doesn’t hold in (4)? The Theorem 1 is not clear then.

7. Why taking mod \theta is required? Cannot you assume that ||Q(x) - x|| <= theta delta? The whole analysis will stay true.

8. In Asynchronous communication section, why \tau_k are the same for all the workers? Cannot different workers have different delays?

9. Why in experiments D-PSGD doesn’t converge when the data are decentralized? This is not consistent with the theory, D-PSGD should converge just slower.

Questions:
10. Is there a way to compute theta it in practice beforehand?

Minor comments:
- x mod y is not defined when x is a float number.
- Intuition behind Moniqua: do you assume that worker 1 has only one neighbour (worker 2)? does m_1 is one dimensional or high dimensional?
- Algorithm 1, the notation \bar X_k might be confusing as in the proof big letters correspond to matrixes. Consider to change on \bar x_k.
- why some equations are numbered and some are not?
- In Asynchronous communication section: what are the assumptions on W_k?
- it would be helpful for the clarity to remind what is the stochastic rounding quantization in configuration of experiments section.
- which are the other algorithms you talk about in footnote 6?
- In wall-clock time evaluation section: how floored outputs could be represented as 16-bit integers? aren’t the vectors which you quantize always smaller than one?
- wall-clock time evaluation section: the statement that “algorithms diverge” might be confusing.
- Aggressive quantization add  ”.”
- why the number of epochs differ in different experiments? Fig. 3(a) and Fig. 4(a) ?


**Experience Assessment:**

I have published one or two papers in this area.

**Review Assessment: Checking Correctness Of Derivations And Theory:**

I assessed the sensibility of the derivations and theory.

**Review Assessment: Checking Correctness Of Experiments:**

I assessed the sensibility of the experiments.

**Review Assessment: Thoroughness In Paper Reading:**

I read the paper at least twice and used my best judgement in assessing the paper.

---

> ### Author Response · Authors · 2019-11-14
> **Thank you very much for your detailed comments and suggestions! (Part 1)**
>
> Thank you very much for your suggestions and comments! Below we address your each concern individually:
>
> ==========Response to Concern 1. ==========
> (i) About constant learning rate.
> In theory, non-constant step size is still an open question in the domain of decentralized SGD. All the baselines as well as variants of decentralized SGD including D-PSGD, D^2 and AD-PSGD analyze theoretical convergence rates with constant step size (Lian et al., 2017a;Tang et al., 2018a;Lian et al., 2017b;Tang et al. 2018b;Koloskova et al. 2019). Since Moniqua is based on plain D-PSGD, in theory we follow the constant step size scheme of D-PSGD (and D^2 and AD-PSGD) while leaving the theory of non-constant step size as future work. We want to emphasize that the goal of Moniqua is to provide a technique for compressed communication that can be applied to a wide variety of decentralized learning algorithms while achieving the same theoretical convergence rate: modifying or advancing the full-precision baselines is beyond the scope of what we are trying to do.
>
> In the spirit of being consistent with theory, we include experimental results based on constant step size. However, you are correct that empirical results of non-constant step size should also be included. We provide extra experimental results on decreasing step size in Appendix C.5 of the revised paper. This experimental results show that with decreasing step size, Moniqua is able to achieve state-of-the-art test accuracy and more robust to low-bits budget compared to baselines.
>
> (ii) About accuracy gap and s.t.o.a test accuracy.
> Thanks for pointing this out!
> First, we want to clarify that in the section of “Aggressive Quantization” where you observe the accuracy gap, we aim at showing the robustness of each algorithm, that is, their ability to converge under extreme cases. In these very extreme cases (2 or 3 bits), no algorithms can achieve state-of-the-art test accuracy. Of all the algorithms tested, Moniqua is the one that is most close to full-precision algorithm.
>
> Second, you are absolutely right that investigating how many bits allows Monqua to achieve s.t.o.a. accuracy is also important. we discuss the number of bits needed by Moniqua to guarantee convergence in section “Efficient Moniqua” and section C.1 of the original paper as well as appended experimental results in section C.5 of the revised paper. The extra experimental results show that when training ResNet110 on CIFAR10, Moniqua needs 6bits to achieve state-of-the-art test accuracy (while baselines suffer from significant accuracy gap). On the other hand, when training ResNet18 on CIFAR10, Moniqua needs 4bits to achieve state-of-the-art test accuracy (while baselines suffer from accuracy gap). By comparison, baselines need more bits to achieve the same accuracy. We also discuss methods of using even fewer bits below 6 bits on ResNet110 in section “Efficient Moniqua”.
>
> ==========Response to Concern 2. ==========
> Thank you for sharing this concern!
> First, please note that the expression on page 5 is the upper bound for the number of bits needed. That being said, for the ring 8 case the number of bits required to converge is at most 5.
>
> Second, in practice the number of bits needed largely depends on the model, dataset and task. Baselines are showing results based on smaller models like ResNet20 (Tang et al., 2018a;b) and logistic regression (Koloskova et al. 2019), and even with these simpler models, no baseline can close accuracy gap below 4 bits (according to their experimental results). In contrast, we are testing Resnet110, and we show in the extra experimental results (section C.5) and “Efficient Moniqua” that Moniqua needs 5-6 bits in average (ring network) for ResNet110 to achieve s.t.o.a accuracy, which is a lower bit-level than baselines.
> We also found the number of bits required largely depends on the structure of model as well. For example, VGG-type models are more robust than ResNet-type models since they have more fully-connected layers (details in section C.1, this finding is supported by previous work (Grubic et al., 2018)). We found 3-4 bits (ring network) in average is sufficient for VGG16 to achieve s.t.o.a accuracy. However, these investigations are beyond the main gist of Moniqua, thus we don’t discuss that in detail in the paper.
>
> ==========Response to Concern 3.==========
> Thank you for pointing this out. It is correct that for some graphs, the spectral gap is not a constant. However, the expression for spectral gap largely depends on the structure and size of the graph. In the presentation of asymptotic convergence rate, we maintain the consistency with related and previous works in decentralized SGD to treat it as a constant and point that out at the bottom of Theorem 2 (Lian et al., 2017a;Tang et al., 2018a;Lian et al., 2017b;Tang et al. 2018b;Koloskova et al. 2019). And for number of bits, we include spectral gap in expression on page 5.

---

> > ### Author Response · Authors · 2019-11-14
> > **Thank you very much for your detailed comments and suggestions! (Part 2)**
> >
> > ==========Response to Concern 4.==========
> > Please note that the averaging rate is not a hyperparameter. As stated in the pseudo code of each algorithm (including DeepSqueeze, ChocoSGD and Moniqua), average step is conducted once per iteration (this is also how baselines implement their algorithms). And none of baselines as well as Moniqua is listing it as a hyperparameter. On the other hand, we agree with you that communication frequency is also an interesting topic, which we will leave as future work.
> > And for the $\theta$, we include that in the last line of the first paragraph under “Wall-clock Time Evaluation”. And $\theta$ in other experiments are given in “Response to Concern 5”.
> >
> > ==========Response to Concern 5.==========
> > -We list part of the hyperparameters in the paragraph ”Configuration”. Here we list more details: for “Wall-clock Time Evaluation”, we use step size=0.05 and $\theta$=3.0. In the “Aggressive Quantization”, we use step size=0.01 and $\theta$=1.0 for 2 bits and $\theta$=1.5 for 3bits.
> > -Asynchronous communication (details on page 39) is where worker communicate by randomly choose one neighbor to average model parameters. We do not model asynchrony but run the algorithm in an actual distributed system (Google Cloud). We believe it’s only in the actual system that allows us to obtain more accurate performance.
> > -Since this concern is overlapping with Question 10, please refer to Response to Question 10 for details.
> >
> > ==========Response to Concern 6.==========
> > This assumption does NOT restrict x to be in [-1,1]. On the contrary, it says the quantizers we study has the following property:
> > 1). If it quantizes an $x \in [-1,1]^d$, then we assume $\|Q(x)-x\|_\infty \leq \delta$;
> > 2). if it quantizes an $x$ outside $[-1,1]^d$, no assumption is needed.
> > Thus we are only making assumptions on the quantizer with respect to a small region. Compared to previous work  (Koloskova et al., 2019; Tang et al., 2018c; 2019) where they assume quantizer is unbiased and has bounded error for all $x \in \mathbb{R}^d$, our assumption is much weaker. And this assumption works with large number of quantizers as we discussed in the “Quantizer” section.
> >
> > ==========Response to Concern 7.==========
> > This is a good question! Note that in Theorem 1, $\theta\delta$ is proportional to the step size while the step size is proportional to $1/\sqrt{K}$, where $K$ is the number of total iterations. That means, if we use a quantizer with assumption $\|Q(x) - x\| \leq \theta\delta$, we will have to increase the precision to reduce $\|Q(x) - x\|$ as $K$ increases, thus at the cost of more bits. So can we use the fixed number of bits to do this? Yes and this is where Moniqua “kicks in”. By Modulo operation, we could achieve higher precision without using more bits. Part of the intuition is introduced in the “Introduction”, here we provide another toy example: Consider using 2 bits to quantize a scalar in [0,3] (without the loss of generality we consider linear quantization), then we get quantized points 0,1,2,3 and the quantization error bound is 1. After using Modulo operation, we rescale the quantization space to a smaller region, e.g. [0,1], then with the same 2 bits quantizing we get 0, 0.333, 0.667, 1 and the quantization error bound becomes 0.333. The intuition part in the Introduction shows that with this rescale, we can still get the model difference as desired.
> >
> > ==========Response to Concern 8.==========
> > As we point out in the asynchronous section, “an iteration represents a single gradient update on one randomly-chosen worker”, that being said, $\tau_k$ refers to the delay of one worker that’s doing the k-th update. Note that this is a standard formulation in previous analysis of asynchronous algorithms (especially in the domain of parallel SGD) (Lian et al., 2017b).
> >
> > ==========Response to Concern 9.==========
> > Note that when data is decentralized, D-PSGD does not have the guarantee to converge (for detailed explanation and analysis, please refer to the previous work in (Tang et al., 2018a)). Intuitively in deep learning, when running D-PSGD on decentralized data, the worker could fall into different local optimas. Note that the results shown in the paper, where D-PSGD is not converging, is aligned with experimental results in previous work (Tang et al., 2018a).
> >
> > ==========Response to Question 10.==========
> > This is a good question! In practice, we recommend two methods for computing $\theta$: 1) we could first run D-PSGD for a few iterations or epochs, and checkpointed the largest value of the infinity norm of gradients, then use our expression shown in Theorem 2 to calculate $\theta$. Since workers are getting close as training proceeds (proof in Lemma 6 on page 23), $\theta$ in the first few iterations can be approximately used as a bound for Modulo operation. 2) another way is to treat $\theta$ as a hyperparameter as step size and use tuning methods such as grid search or random search.

---

> > > ### Author Response · Authors · 2019-11-14
> > > **Thank you very much for your detailed comments and suggestions! (Part 3)**
> > >
> > > ==========Response to minor comments==========
> > > Thank you very much for sharing these detailed minor comments. Since some of the comments are regarding implementations, we provide response to them below:
> > > 1. About the definition of modulo when x is a float number: In computing, the modulo operation returns the remainder of numerator/denominator. And the return type can be specified. For example, in C++11, such modulo operation on float number is defined as “fmod”.
> > > 2. Intuition behind Moniqua: This is a toy example for better illustrating the idea of Moniqua. To simplify things, we consider two workers in the system and model is one dimensional. And we believe it is trivial to extend it to high dimensional since coordinates do not have dependency when applied Modulo operation.
> > > 3. We provide assumption for the two extensions on page 27 and page 39-40. For asynchronous communication specifically, please refer to section E for some preliminary illustrations.
> > > 4. Other algorithm refers to (Reisizadeh et al.), which does not converge in the domain of non-convex DNN problem.
> > > 5. About floored outputs in 16-bit integers: Note that since all the workers are using the same delta, we compute the delta on the receiver side. For example, if a sender needs to send 0.98 and delta is 0.01, then it sends 98 and the receiver recovers it by times 98 with delta and get 0.98.
> > > 6. Your other comments are regarding writing and organization of the paper. We sincerely thank you for detailed suggestions and we will fix them in the final version of the paper.
> > >
> > > Thanks again for your detailed reviews and suggestions! If you think our clarification and the extra results address your concerns, we would be appreciated if you could reconsider your score for this paper. On the other hand, if you believe there are still significant flaws or questions, please let us know and we would be happy to improve it!

---

> > > > ### Comment · AnonReviewer3 · 2019-11-14
> > > > **Some concerns**
> > > >
> > > > I have checked only parts of the authors comments so far, and this is not my full response. However, as the rebuttal phase is ending soon, I put some of my concerns that the authors still have time to reply.
> > > >
> > > > 1. “In these very extreme cases (2 or 3 bits), no algorithms can achieve state-of-the-art test accuracy” -> I noticed that [1] shows that with sign compression (around 1 bit per coordinate) SOTA accuracy can still be achieved on the Cifar10 dataset (the same dataset as used here). Therefore, I still believe that the experimental comparison is not convincing enough. The goal of using compressed communications is to speed up the convergence while not compromising the accuracy too much (10-20% is a lot). If with compressed communications the algorithm cannot achieve SOTA accuracy, then in practice it makes sense to wait a bit longer and use full precision training instead.
> > > >
> > > > [1] Decentralized Deep Learning with Arbitrary Communication Compression
> > > > A. Koloskova, T. Lin, S. U. Stich, M. Jaggi (https://arxiv.org/abs/1907.09356)
> > > >
> > > > 4. I think there is a misunderstanding: by the averaging rate I meant the \gamma parameter in Choco-SGD and \eta in DeepSqueeze --- these are hyperparameters. Without stating hyperparameter tuning details (of averaging learning rate for the baselines / theta for the Moniqua; and of SGD learning rate), the results are not reproducible and also are questionable.
> > > >
> > > > 5. when reporting hyperparameters, please report hyperparameters of the baselines too.

---

> > > > > ### Author Response · Authors · 2019-11-14
> > > > > **Thanks for the quick reply!**
> > > > >
> > > > > Thank you very much for such quick reply!
> > > > >
> > > > > 1. Thanks for sharing this paper! (we refer it as [1])
> > > > > First, we think it is an inspiring work! However, we find some of the results shown in [1] contradict several previous results in [2] and [3] ([1][2][3] are running exactly the same task, ResNet20 on CIFAR10 with ring 8 network).  Specifically, [2] shows DCD-PSGD cannot converge at 4 bits and ECD-PSGD can with unbiased quantizer while [1] shows exactly the opposite result. And [3] shows that [1] diverges with 2 bits while [1] shows they can achieve s.t.o.a accuracy with fewer than 1 bit. It's probably because [1] is adopting uncommon step size (1.60, as shown on page 19 where normally ResNet is tuned with step size <1), and that is beyond our tuning scope.
> > > > > We notice that results in [1] is reported at 300 epoch while full-precision algorithm could achieve s.t.o.a accuracy much earlier (aound 230 epochs). It is not clear whether the more iterations will compensate the benefit from communication acceleration (since they do not include wall clock time comparison).
> > > > >
> > > > > Second, Moniqua can achieve s.t.o.a accuracy, but not in these extreme cases (1-2 bits). We agree with you that ultimate accuracy is also important. However in practice, different from centralized SGD,  extreme low bits-budget such as 1 or 2 bit communication is rarely used in decentralized training since it could cause consensus problem and make workers fall into different local optima, especially with when data is locally gathered. Moniqua is able to achieve s.t.o.a accuracy with slightly more bits on ResNet110 (<6 bits) and VGG16 (<4bits) (as discussed in the rest of the rebuttal.)
> > > > >
> > > > > Third, as we point out in "Response to Concern 2", the number of bits needed largely depends on the model, dataset and task. Note that we are testing ResNet110, which has 6.3X parameters than ResNet20. And  we are using stochastic rounding, which is a much simpler quantizer compared to those used in [1]. (We will try our best to add more results using advanced quantizers from [1] before rebuttal ends, but we think that issue is orthogonal to Moniqua.)
> > > > >
> > > > > Fourth, different from Moniqua with zero additional memory overhead, [1] needs at least 3X additional memory (in the worst case $n^2$ where n is number of workers). This is not very practical in many cases. In practice, unlike datacenter networks, decentralized SGD is often used in cases where workers have limited memory, computing resources and bandwidth. Applicable scenarios include ad hoc networks, mobile networks and Internet of Things (IoT). In these cases, the most serious issue is how to utilize limited resources to get close to the state-of-the-art accuracy as much as possible. Previous works on compressed communication for decentralized learning mitigate the bandwidth problem by trading it off for  more memory and computation [1][2][3], which does not fully solve the problem since memory and computation are also typically limited. Motivated by this, Moniqua is a additional-memory-and-computation-free solution that enables decentralized SGD to use less bandwidth, thus achieving acceleration in the limited-bandwidth setting with no extra overheads.
> > > > >
> > > > > [2]Communication Compression for Decentralized Training.
> > > > > Hanlin Tang, Shaoduo Gan, Ce Zhang, Tong Zhang and Ji Liu.(https://arxiv.org/pdf/1803.06443.pdf)
> > > > >
> > > > > [3] DeepSqueeze: Decentralized Meets Error-Compensated Compression.
> > > > > Hanlin Tang, Xiangru Lian, Shuang Qiu, Lei Yuan, Ce Zhang, Tong Zhang and Ji Liu.
> > > > > (https://arxiv.org/pdf/1907.07346.pdf)
> > > > >
> > > > >
> > > > > 4&5: Sorry for the misunderstanding, we list the hyperparameters for baselines here:
> > > > > In the "Wall-clock Time Evaluation":
> > > > > ChocoSGD:(step size=0.05, $\gamma=0.75$);
> > > > > DeepSqueeze: (step size=0.1, $\eta=0.8$);
> > > > > DCD-PSGD: (step size=0.05);
> > > > > ECD-PSGD: (step size=0.1)
> > > > >
> > > > > In the "Aggressive Quantization":
> > > > > For 2 bits experiment, (step size=0.01, $\gamma=0.25$) for ChocoSGD and (step size=0.01, $\eta=0.40$) for DeepSqueeze; (step size=0.01) for DCD-PSGD; (step size=0.01) for ECD-PSGD.
> > > > > For 3 bits experiment: (step size=0.01, $\gamma=0.35$) for ChocoSGD and (step size=0.05, $\eta=0.60$) for DeepSqueeze; (step size=0.01) for DCD-PSGD; (step size=0.01) for ECD-PSGD.
> > > > >
> > > > > Thanks again for the quick reply. Please let us know if you have more concerns!

---

> > > > > > ### Comment · AnonReviewer3 · 2019-11-15
> > > > > > **Reply**
> > > > > >
> > > > > > Thanks for your reply, however, I don’t agree with the points you made. I have the following questions:
> > > > > >
> > > > > > (i) I also notice that in your new experiments with ResNet110 and ResNet18 (Fig. 7). you still have 6-10% accuracy gap (which was not stated). For training ResNet18 on Cifar10 depending on the width, the accuracy it can achieve is between 91 and 95 %. This should be stated clearly in the paper with the accuracy gap value.
> > > > > >
> > > > > > (ii) [1] reaches s.t.o.a accuracy for their model. So it looks strange not including in your tuning range stepsizes larger than 1, where the baselines ([1]) provided evidence for using these stepsizes. As it stands now one would conclude that your tuning is done in an unfair way.
> > > > > >
> > > > > > Suggestion:
> > > > > > Another reasons for this large accuracy gap you have might be that you use an uncommon training procedure (decreasing the stepsize by factor of 10 every 30 epoch). Even d-psgd baseline could not achieve a good accuracy. I suggest you to try another training procedure. I think the training procedure of [1] is more standard.

---

> > > > > > > ### Author Response · Authors · 2019-11-15
> > > > > > > **Thank you very much for sharing these concerns!**
> > > > > > >
> > > > > > > Thank you for your reply!
> > > > > > >
> > > > > > > (i) You are right that we should highlight this in that section! Thank you for pointing this out and we will update this! However, we would like to point out that, in the results of ResNet18 in Figure 7, the accuracy gap you mentioned is from full-precision D-PSGD baseline (Moniqua doesn't suffer gap compared to D-PSGD). This "gap" is aligned with the result from D-PSGD paper [4], where they also get test accuracy below 90%.
> > > > > > >
> > > > > > > The gap in D-PSGD can be caused by many reasons. Since the rebuttal time is limited and ResNet18 is a newly added experiment, we do omit batch norm in all of the baselines as well as Moniqua in training ResNet18, which could be a reason. (But for ResNet110 in figure 7 where we use batch norm, that converges to the s.t.o.a accuracy.)
> > > > > > >
> > > > > > > Since Moniqua is a work that aims at achieving same rate as full-precision D-PSGD, we think it's beyond our scope to improve baseline D-PSGD's accuracy on a particular task.
> > > > > > >
> > > > > > > [4] Can Decentralized Algorithms Outperform Centralized Algorithms? A Case Study for Decentralized Parallel Stochastic Gradient Descent.
> > > > > > > Xiangru Lian, Ce Zhang, Huan Zhang, Cho-Jui Hsieh, Wei Zhang and Ji Liu.
> > > > > > > (https://arxiv.org/pdf/1705.09056.pdf)
> > > > > > >
> > > > > > > (ii) The step size scheme we use is from [5]. We actually did try step size larger than 1 in the experiments on ChocoSGD. However, we found that this aggressive step size causes severe instability of algorithm and is easy to diverge (large step size also contradicts theory shown in [1]). In practice, we haven't seen any algorithm using step size >1 (or >2). We understand this might be a new direction to explore and using large step size can be something significant in this domain. But since [1] does not specify any open source code, we felt hard to reproduce their results.
> > > > > > >
> > > > > > > [5] Deep Residual Learning for Image Recognition
> > > > > > > Kaiming He, Xiangyu Zhang, Shaoqing Ren and Jian Sun.
> > > > > > > (https://arxiv.org/pdf/1512.03385.pdf)
> > > > > > >
> > > > > > > Thanks again for your concerns! Please let us know if there are other questions!

---

### Official Review · AnonReviewer2 · 2019-10-30
**Official Blind Review #2**

**Rating:** 8

**Review:**

This work is about the use of quantized communication in decentralized stochastic gradient descent. The general advantage of using quantized communication is that is has the potential to reduce the amount of data exchanged, thereby leading to faster convergence -- and the main focus of this work is to include ideas that stem from quantized communication into decentralized training approaches. Competing methods of this kind suffer from potentially severe problems like memory overhead, limited applicability to convex problems, or the need for nonlinear quatizers.

The proposed Moniqua algorithm is an attempt to overcome these shortcomings. A central technical contribution is the analysis of direct quantization strategies in decentralized SGD in Theorem 1, which basically states that local models can fail to converge even for simple objective functions. The motivation for the Moniqua algorithm -- which is designed to solve this problem of local models -- is rather intuitive, and the algorithm can be implemented quite easily in practice. Some theoretic insight is provided, such as the asymptotic convergence rate (which is the same as in D-PSGD) and the loglog(n)-bound on the number of bits per parameter communicated as a function of the number of parallel workers.

Moniqua is empirically evaluated on a large number of different network configurations, and it seems that it indeed converges faster than other related algorithms, while being highly robust to aggressive quantization and strict bit-budgets. In summary, this work addresses a highly relevant problem and it nicely combines formal asymptotic analysis with experimental validation. I think, this work has indeed a great potential to advance this field of research.

**Experience Assessment:**

I have read many papers in this area.

**Review Assessment: Checking Correctness Of Derivations And Theory:**

I assessed the sensibility of the derivations and theory.

**Review Assessment: Checking Correctness Of Experiments:**

I assessed the sensibility of the experiments.

**Review Assessment: Thoroughness In Paper Reading:**

I read the paper at least twice and used my best judgement in assessing the paper.

---

> ### Author Response · Authors · 2019-11-14
> **Thank you very much for your reviews and comments!**
>
> Thank you very much for your reviews and comments! Your comments on  our paper is a really good summary that captures the main gist and contributions of Moniqua. We will be excited to see how Moniqua could advance the domain of decentralized optimization!

---

### Official Review · AnonReviewer4 · 2019-11-25
**Official Blind Review #4**

**Rating:** 3

**Review:**

This papers proposed an interesting idea for distributed decentralized training with quantized communication. The authors show that naively compressing the exchanged model can fail to converge, and introduce to compress the model difference with modulo operation. The idea is further applied to decentralized data and asynchronous optimization settings. Convergence analyses are provided for non-convex problems.

Though I found this paper interesting and novel, I have several concerns:

1. There is no convincing explanation why modulo operation makes the algorithm better. In particular, the equation $((m_2)_j - (m_1)_j) \text{mod} \theta = (m_2)_j - (m_1)_j$ does not hold. For example, let $(m_2)_j = 5$, $(m_1)_j = 3$ and $\theta=2$, then $((m_2)_j - (m_1)_j) \text{mod} \theta = (5 - 3) \text{mod} 2 = 0$ and $(m_2)_j - (m_1)_j = 5- 3 = 2$.

2. The proposed algorithm requires knowledge of $\theta$, which depends on the upper bound of gradient. If $\theta$ is underestimated, the algorithm will suffer from large errors. For instance, suppose that true $\theta$ is $5$ while we use $4.5$, then an identity quantizer cannot even recover the original value, e.g., $(5/4.5) \text{mod} 1*4.5 = 0.5 << 5$. On the other hand, if $\theta$ is overestimated, the convergence is dramatically slowed down. I checked authors' response to Reviewer 3 regarding this issue. However, they are not efficient and can still provide wrong estimates.

3. The upper bound of staleness is missing in the main text and convergence bound. I found that bounded delay is assumed in the supplementary. However, the constant for bounded delay is missing in Theorem 4.

4. The experiments are not convincing. In particular, Figure 2 shows that all the quantization methods perform very bad with low bits format. The centralized method DoubleSqueeze uses error feedback and supports arbitrary compression. Based on my experience, even with the 1-bit quantizer introduced in (Karimireddy et al., 2019), DoubleSqueeze converges as fast as full precision SGD. Also, it seems that final test accuracy of D-PSGD on ResNet110 is just 80+, which is much lower than its official 93.6% test accuracy. This indicates that experiments are not appropriately done.

5. CIFAR-10 is a very small scale dataset. For distributed training, it is more convincing to conduct experiments on larger datasets such as ImageNet.

They are a lot of typos. The authors need to double check.

Karimireddy et al., Error Feedback Fixes SignSGD and other Gradient Compression Schemes. ICML 2019.


**Experience Assessment:**

I have published one or two papers in this area.

**Review Assessment: Checking Correctness Of Derivations And Theory:**

I assessed the sensibility of the derivations and theory.

**Review Assessment: Checking Correctness Of Experiments:**

I carefully checked the experiments.

**Review Assessment: Thoroughness In Paper Reading:**

I read the paper thoroughly.

---

### Author Response · Authors · 2019-11-14
**We thank all the reviewers for their time and suggestions!**

Dear Reviewers,

Thank you very much for all the concerns and reviews.

We submitted a revised version, where we fix some wording and provide some extra experimental results to address your concerns and questions. Here is a list of experimental results (details are in section C.5 of the revision paper, page 15-16):

1. We provide results on Moniqua with decreasing step size on different models.
2. We provide the final test accuracy on Moniqua and other baselines under different bit-level.
3. We provide consensus error results when using Moniqua both with constant and decreasing step size.

Please let us know if you have any other questions. Thank you again for your reviews!

---

### Decision · Program_Chairs · 2019-12-19

**Decision:**

Reject

**Comment:**

This papers proposed an interesting idea for distributed decentralized training with quantized communication. The method is interesting and elegant. However, it is incremental, does not support arbitrary communication compression, and does not have a convincing explanation why modulo operation makes the algorithm better. The experiments are not convincing. Comparison is shown only for the beginning of the optimization where the algorithm does not achieve state of the art accuracy. Moreover, the modular hyperparameter is not easy to choose and seems cannot help achieve consensus.